# Glycerol-weighted chemical exchange saturation transfer nanoprobes allow $^{19}F/^{1}H$ dual-modality magnetic resonance imaging-guided cancer radiotherapy

Rong A [1,2,3], Haoyu Wang[1,2,3], Chaoqun Nie [2,3], Zhaoguo Han[1,2,3], Meifang Zhou[1,2], Olagbaju Oluwatosin Atinuke[1,2], Kaiqi Wang[1,2], Xiance Wang[1,2], Shuang Liu[1,2], Jingshi Zhao[2], Wenju Qiao[1,2], Xiaohong Sun[1,2], Lina Wu[1,2] & Xilin Sun [1,2] ✉

Recently, radiotherapy (RT) has entered a new realm of precision cancer therapy with the introduction of magnetic resonance (MR) imaging guided radiotherapy systems into the clinic. Nonetheless, identifying an optimized radiotherapy time window (ORTW) is still critical for the best therapeutic efficacy of RT. Here we describe pH and $O_2$ dual-sensitive, per-fluorooctylbromide (PFOB)-based and glycerol-weighted chemical exchange saturation transfer (CEST) nano-molecular imaging probes (Gly-PFOBs) with dual fluorine and hydrogen proton based CEST MR imaging properties ($^{19}F/^{1}H$-CEST). Oxygenated Gly-PFOBs ameliorate tumor hypoxia and improve $O_2$-dependent radiotherapy. Moreover, the pH and $O_2$ dual-sensitive properties of Gly-PFOBs could be quantitatively, spatially, and temporally monitored by $^{19}F/^{1}H$-CEST imaging to optimize ORTW. In this study, we describe the CEST signal characteristics exhibited by the glycerol components of Gly-PFOBs. The pH and $O_2$ dual-sensitive Gly-PFOBs with $^{19}F/^{1}H$-CEST MR dual-modality imaging properties, with superior therapeutic efficacy and biosafety, are employed for sensitive imaging-guided lung cancer RT, illustrating the potential of multi-functional imaging to noninvasively monitor and enhance RT-integrated effectiveness.

Radiotherapy (RT) is a primary modality in cancer treatment, with more than half of all cancer patients receiving RT for curative or palliative reasons. It is noteworthy that radiotherapy has the potential to elicit irreversible DNA damage, a phenomenon that exhibits a direct correlation with the degree of tumor oxygenation[1–4]. However, the tumor-hypoxic environment characterized by disordered vasculature and rapid proliferation of tumors is involved in the repair of radiation-mediated DNA damage, ultimately resulting in the failure of tumor eradication[2,5]. Therefore, in attempting to achieve better outcomes, the RT encounters several hurdles, such as radiation resistance from the hypoxic microenvironment and excessive radiation to overcome hypoxia causing damage to adjacent healthy tissues[6–8]. Moreover, tumor hypoxia is often associated with the excessive accumulation of $H^+$ ions in the tumor microenvironment (TME), in turn, acidic TME

[1]Department of Nuclear Medicine, the Fourth Hospital of Harbin Medical University, Heilongjiang Province, China. [2]NHC Key Laboratory of Molecular Probe and Targeted Diagnosis and Therapy, Molecular Imaging Research Center (MIRC) of Harbin Medical University, Heilongjiang Province, China. [3]These authors contributed equally: Rong A, Haoyu Wang, Chaoqun Nie, Zhaoguo Han. ✉e-mail: sunxl@ems.hrbmu.edu.cn

further facilitates the development of hypoxic tumor regions[9,10]. This vicious cycle eventually leads to the exacerbation of tumor hypoxia, RT resistance, and even worsening therapeutic efficacy.

Clinically, RT sensitizers, such as oxygen[11], hyperbaric oxygen[12,13], and oxygen mimetics, have been used to overcome tumor hypoxia. Also, proton pump inhibitors, such as esomeprazole (EMSO)[14,15] and lansoprazole (LAN)[16,17], have been proven to be modulators of the tumor acidic microenvironment. With the introduction of the most advanced MRI-guided radiotherapy system in the clinic, magnetic resonance-guided precision radiotherapy overcoming the disadvantages of CT and PET-guided radiotherapy has entered a new stage[18,19], which pushes the deep explorations and development of the multifunctional nanoprobes to solve the still existing limitations in cancer RT. Another critical issue that needs to be addressed while administering oxygen-based RT sensitizers is the optimal timing for RT. In other words, to obtain ideal synergistic therapeutic effects, it is essential to figure out whether the oxygen is effectively supplied to the tumor hypoxia microenvironment, what is the real-time status of tumor hypoxia and acidic microenvironment, and which time is optimal for the implementation of radiotherapy[2,20,21]. Further clinical challenges include quantitative determination and visualization of TME dynamic hypoxic and acidic changes.

The rapid development of emerging advanced nanomaterials and nanobiotechnology platforms has provided opportunities to overcome radiation resistance and visualize the optimal time window for radiation therapy[22–25]. In this context, perfluorinated compounds (PFCs) are inert organic compounds with excellent biocompatibility that were previously used in the clinic as "artificial blood" to improve tissue oxygenation due to their high affinity for $O_2$ via van der Waals interaction[26]. Furthermore, the fluorine-rich characteristics of PFCs facilitate the utilization of in vivo $^{19}$F-MR imaging[27,28]. PFCs are situated within the phospholipid membrane at the water interface, while the nano-emulsions of PFCs exhibit a substantial surface area and facilitate convenient modifications[29]. Consequently, PFC nano-emulsions hold promise for being harnessed as a versatile magnetic resonance (MR) contrast agent. PFOB used in this study stands out among PFCs for medical use as it, non-toxic, high stability, inertness, possesses the unique property of being both hydrophobic and lipophobic, and has an acceptable excretion profile[30].

Chemical exchange saturation transfer (CEST) is a relatively new and promising magnetic resonance molecular imaging approach that measures proton exchange between the exchangeable solute protons and the much larger pool of bulk water protons[31–35]. The magnetization of exchangeable solute protons is detected through spin saturation and transferred into water via exchange, and the exchange rates of these exchangeable protons often depend on pH, CEST has shown excellent performance for pH assessment in the extracellular TME with unique, accurate, and enhanced sensitivity[36–38].

In this work, we propose a systematic methodology that can simultaneously address the above-mentioned to achieve optimal RT therapeutic outcomes. Specifically, we disclose one kind of glycerol-weighted and PFOB-based CEST nano-molecular imaging probes, Gly-PFOBs, as a precision image-guided RT-sensitized strategy. Gly-PFOBs represent an example of PFCs with chemical exchange saturation transfer properties for pH and oxygen visualization that can also serve as radiosensitizers. Using this strategy, (i) the $O_2$ and pH dual sensitivities can dynamically reflect the changes in tumor hypoxia and acidic microenvironment, (ii) high oxygen-carrying characteristic can effectively improve tumor hypoxia and exert maximum synergistic radiotherapeutic effects, (iii) the real-time and non-invasive $^{19}$F/$^{1}$H-CEST dual-modality MR imaging provides an optimized radiotherapy time window (ORTW) for precision radiotherapy (Fig. 1).

## Results

### Characterization of $^{19}$F/$^{1}$H-CEST dual-imaging modality Gly-PFOBs

TEM results of 623 mM Gly-PFOBs (glycerol-based) and control probes without CEST signals (PFOBs) displayed spherical or quasi-spherical morphology with a uniform distribution (Fig. 2a). Elemental mapping analysis for Gly-PFOBs also confirmed the compositions of the N, F, P, and O elements (Fig. 2b). As shown by DLS, the hydrodynamic particle size of Gly-PFOBs and PFOBs were about 168.2 nm and 171.8 nm, respectively, and both had a similar zeta potential around −50 mV at 37 °C. The UV-Vis spectra of lissamine rhodamine B sulfonyl labeled Gly-PFOBs and PFOBs presented the same absorption peak at 570 nm, while glycerol showed no absorption peak. Besides, there were no obvious changes in the diameter and zeta potential of Gly-PFOBs and PFOBs at 4 °C, 25 °C, and 37 °C within 45 days, indicating excellent stability of Gly-PFOBs (Fig. 2c and Supplementary Fig. 1a–d).

To evaluate the glycerol-retaining ability of Gly-PFOBs, we applied free glycerol assay kit for the glycerol content test. The results showed that the content of glycerol in the dialysate slightly increased over time. At the 6 h of dialysis, only 1.04% (32.3 μmol/3115 μmol) and 1.33% (41.43 μmol/3115 μmol) of glycerol from Gly-PFOBs free into the external environment (Supplementary Table 1, Supplementary Fig. 2a). To further verify the above-measured results, the Gas Chromatography-Mass Spectrometry (GC-MS) was used to quantify glycerol concentration in Gly-PFOBs dialyzed against ultra-pure water or saline for different lengths of time. Compared with Gly-PFOBs without dialysis, the glycerol concentration in Gly-PFOBs was only slightly decreased after dialyzed against ultra-pure water or saline (Supplementary Table 2, Supplementary Fig. 2b). These results were consistent with the results of the Free Glycerol Assay Kit.

As is evident from Fig. 2d and Supplementary Fig. 3, the CEST signal at 0.68 ppm frequency shift could be detected at lower saturation powers, while the signal intensity was maximized at relatively higher saturation powers. The signal intensity correspondingly increased as the glycerol concentration increased, but at 425 mM and 623 mM concentrations, the Gly-PFOBs were moderately differentiated because of limited free water molecule content in probe solutions. Furthermore, the $^{1}$H-CEST imaging analysis results presented in Supplementary Fig. 4a, b illustrated that the CEST signal intensity increased with increasing pulse durations and glycerol concentrations in Gly-PFOBs, while the CEST signal differences between 3 s and 5 s pulse durations were not significant with higher glycerol concentrations (425 mM, and 623 mM). In addition, different glycerol aqueous solution concentrations were used and scanned with the same magnetic resonance sequence parameters to verify whether the Gly-PFOBs signal was derived from glycerol. The $^{1}$H-CEST imaging results displayed in Fig. 2e indicated that the CEST signal of Gly-PFOBs was derived from glycerol. In consideration of the potential obscuring of CEST effects in vivo experiments caused by the broadening of the direct saturation line width at higher saturation powers, we opted to utilize lower saturation powers (0.8 μT or 1.0 μT) with a pulse duration of 5 s in subsequent experiments. Also, when the $^{19}$F/$^{1}$H-CEST dual-imaging capability of Gly-PFOBs was evaluated, Gly-PFOBs provided a stronger CEST signal at higher concentrations. Expectedly, the glycerol modification dose did not affect the $^{19}$F-MR signal intensity (Fig. 2f and Supplementary Fig. 5a).

### In vitro pH and $O_2$ dual sensitivities and hypoxia alleviation by oxygenated Gly-PFOBs

Since pH often directly affects the exchange rates of exchangeable protons[31–35], we examined the CEST properties of Gly-PFOBs at different pH values. Figure 3a, b shows a highly pH-dependent CEST effect of Gly-PFOBs with a significantly decreased MTR$_{asym}$ at increased pH levels from 5.9 to 8.0 at the saturation power of 0.6 μT, 0.8 μT, 1.0 μT, 1.2 μT and 1.8 μT. Furthermore, the phantom images in Fig. 3c

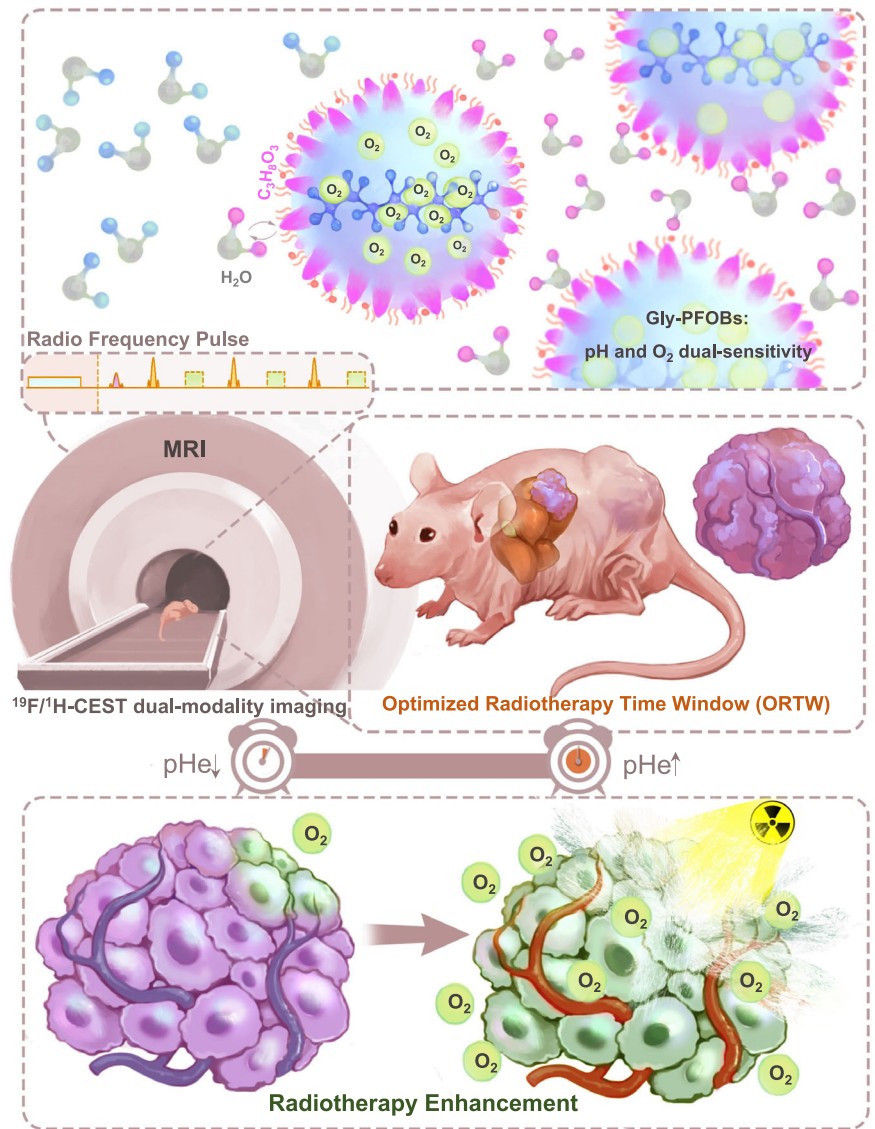

**Fig. 1 | Schematic illustration of $^{19}$F/$^{1}$H-CEST dual-imaging modality Gly-PFOBs.** The precision imaging-guided RT-sensitized strategy utilizes glycerol-weighted and PFOB-based CEST nano-molecular imaging probes, known as Gly-PFOBs. In this strategy, (i) Gly-PFOBs with $O_2$ and pH dual sensitive properties dynamically reflect the changes in tumor hypoxia and acidic microenvironment, (ii) the high oxygen-carrying characteristic of Gly-PFOBs effectively improves tumor hypoxia and exerts maximum synergistic radiotherapeutic effects, and (iii) the real-time and non-invasive $^{19}$F/$^{1}$H-CEST dual-modality MR imaging enables the identification of an optimized radiotherapy time window (ORTW) for precision radiotherapy.

demonstrate the feasibility of using $^{19}$F/$^{1}$H-CEST MR to image and assess the pH level (1.0 μT, 5 s). Consistent with the Z-spectra, Gly-PFOBs at acidic pH presented strong CEST signal intensity. In addition, different pH values of Gly-PFOBs solutions did not interfere with the $^{19}$F-MR signals due to the hydrogen proton exchange between glycerol and surrounding physiological water with no effect on the relaxation properties of fluorine atoms (Fig. 3c and Supplementary Fig. 5b).

The pH sensitivity of Gly-PFOBs at the cellular level was evaluated by incubating NCI-H460 lung cancer cells with the proton pump inhibitor ESOM to manipulate the extracellular pH. ESOM is known for its efficacy and tolerability[39] and lowers intracellular pH (pHi) while elevating the extracellular pH (pHe). A significant statistical difference was observed between the Gly-PFOBs signal intensity of the ESOM-treated and saline-treated groups, while no signal difference was observed in the PFOBs group ($P < 0.05$) (Fig.3d). When cell supernatants were used for pH measurements to confirm $^{19}$F/$^{1}$H-CEST MR imaging results, the pH was higher (pH $6.97 ± 0.01$) in the ESOM-treated group than the saline group (pH $6.82 ± 0.02$). These results

were consistent with the $^{19}$F/$^{1}$H-CEST MR imaging results ($P < 0.001$, Fig. 3e), and illustrated that Gly-PFOBs could reveal the minimal pH fluctuations (ΔpH = 0.15 units) in vitro.

PFOB has large solubilization capacity for oxygen among PFC compounds via the van der Waals interaction. PFOB, especially 20% v/v, saturated with oxygen remarkably increased the oxygen content in deoxygenated water, indicating its efficient oxygen loading and gradual release in an oxygen-deficient environment (Fig. 3f). As calculated from oxygen concentration curve, the oxygen loading capacity of Gly-PFOBs was $15 ± 0.1$ mg/L (Fig. 3g). Further, we investigated whether oxygen could affect the CEST signal intensity of PFOB (20%, v/v)-based Gly-PFOBs. The CEST signal gradually enhanced with the release of oxygen, and the signal difference (ΔST%) in the oxygen-releasing group was about 2.5 times higher than in the argon (Ar)-releasing group (Fig. 3h, i). In addition, based on the $^{19}$F $T_1$ values of Gly-PFOBs with different oxygen partial pressures, the real-time oxygen partial pressures were calculated, and the CEST signals at corresponding oxygen partial pressures were obtained (Supplementary Fig. 6a, b).

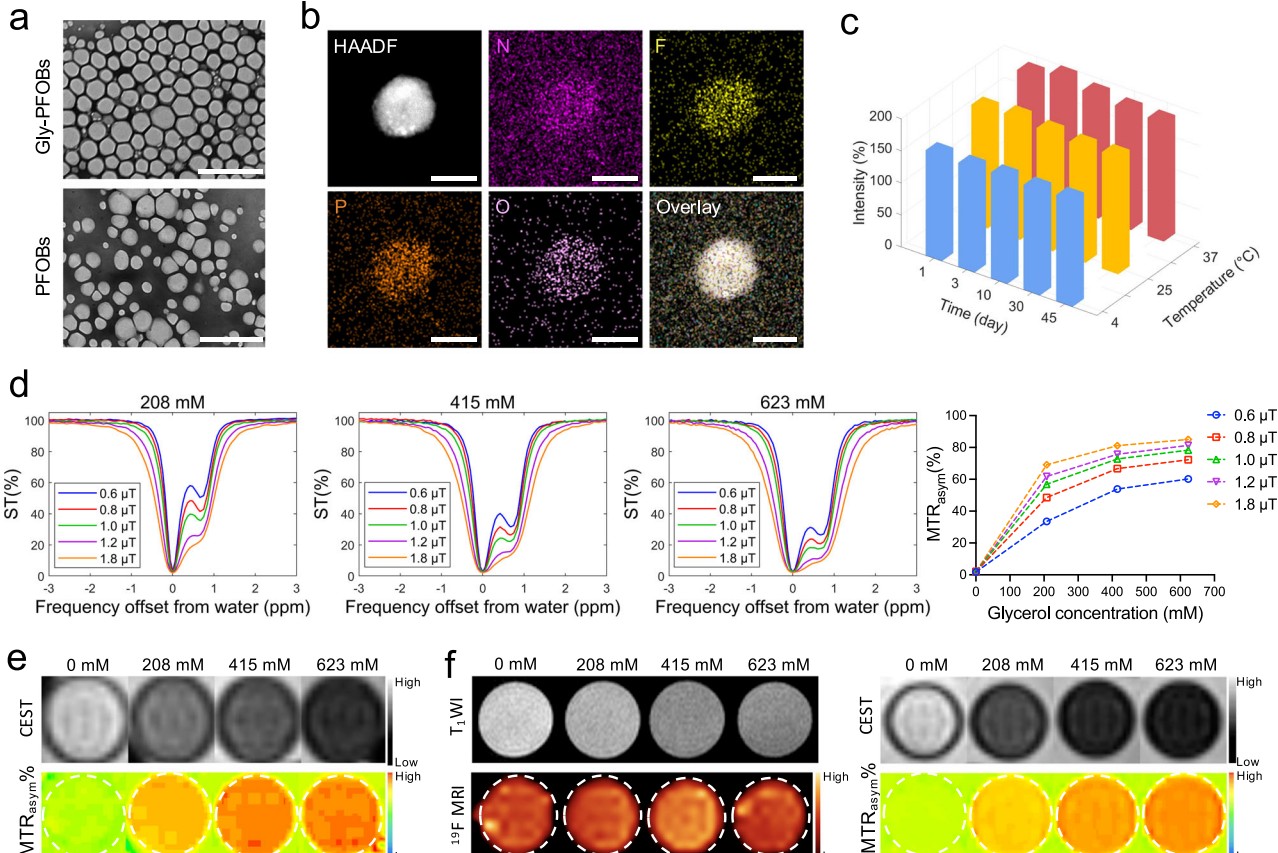

**Fig. 2 | Characterization of $^{19}$F/$^1$H-CEST dual-imaging modality Gly-PFOBs.**
**a** Morphologies of Gly-PFOBs and control probes without CEST signals (PFOBs) were determined by TEM. Scale bars, 1 μm. Repeated three times independently with similar results. **b** Elemental mapping analysis of Gly-PFOBs, repeated one time. Scale bars, 50 nm. **c** Measurement of changes in the intensity-averaged mean particle size (hydrodynamic particle diameter) of Gly-PFOBs, $n = 5$ independent measurements. Three-dimensional (3D) graphs were generated using Matlab based on the average values obtained. **d** CEST signal properties of Gly-PFOBs with different glycerol concentrations (0 mM, 208 mM, 415 mM, and 623 mM) at different saturation pulse powers (0.6 μT, 0.8 μT, 1.0 μT, 1.2 μT, and 1.8 μT). Repeated three times independently with similar results. **e** $^1$H-CEST MR imaging results of aqueous glycerol solutions with different concentrations. Repeated 3 times independently with similar results. **f** T$_1$WI, $^{19}$F/$^1$H-CEST MR imaging results of Gly-PFOBs with different glycerol concentrations. Repeated 3 times independently with similar results. CEST chemical exchange saturation transfer, MTR$_{asym}$ magnetization transfer ratio asymmetry, ST% saturation frequency. Source data are provided as a Source data file.

These results indicated that oxygen could affect the CEST signal intensities of Gly-PFOBs that probes with higher oxygen contents present lower CEST signal intensity.

Overall, the excellent pH and oxygen sensitivity CEST effects of Gly-PFOBs make them a perfect candidate for tumor imaging applications. Further, we investigated the capability of oxygenated Gly-PFOBs to reverse cell hypoxia. As shown in Fig. 3j, k, compared with Gly-PFOBs without oxygenation, hypoxic NCI-H460 cells exhibited a significant change in HIF-1α expression after incubating with oxygenated Gly-PFOBs ($P < 0.001$). Moreover, the fluorescence results of lissamine rhodamine B sulfonyl demonstrated an excellent cellular uptake of Gly-PFOBs. In addition, the hypoxic cell viability under 0, 2, 4, 6, or 8 Gy X-ray radiation in the presence of Gly-PFOBs was measured. Compared with the RT alone group, Gly-PFOBs (O$_2$) combined with X-ray treatment could further reduce cell viability, suggesting the promising potential of the Gly-PFOBs (O$_2$) probes as an X-ray sensitizer for hypoxic cancer cell killing ($P < 0.0001$) (Supplementary Fig. 7).

### Detection of in vivo pH changes
We established a warm partial liver ischemia model to simulate the changes in the acidic microenvironment caused by hypoxemia in tissues for evaluating the pH sensitivity of Gly-PFOBs in vivo. The $^{19}$F/$^1$H-CEST MR imaging results revealed that the fluorine signal

distribution was uniform in ischemic and non-ischemic livers, excluding the factors that caused the difference in CEST signal due to uneven distribution (Fig. 4a). In addition, statistical analysis of the results depicted higher CEST signal intensity in the ischemic than non-ischemic liver ($P < 0.01$) (Fig. 4b). The main reason for this signal discrepancy is believed to be the insufficient O$_2$ supply and increased glucose metabolism acidify the pH within ischemic liver tissues and the overproduction of lactate generated in the liver microenvironment[40,41]. The lactate imaging (CSI sequence) results showed a higher lactate level in the ischemic than non-ischemic liver (Fig. 4a). There were no significant morphological changes between the non-ischemic and ischemic liver. However, a higher pimonidazole immunofluorescence signal was observed in the ischemic liver, indicative of hypoxia (Fig. 4c, d and Supplementary Fig. 8a, b). Lissamine rhodamine B sulfonyl labeled Gly-PFOBs presents similar fluorescence intensity in the non-ischemic and ischemic liver, which fits with $^{19}$F-MR imaging results. The pH differences between ischemic and non-ischemic livers were confirmed by the pH-sensitive chemical optical microsensor, and the statistical difference of measurement results illustrated a lower pH level in the ischemic than non-ischemic liver tissue, supportive of the $^{19}$F/$^1$H-CEST dual-modality MR imaging results (ΔpH = 0.245 units, $P < 0.01$, Fig. 4e).

Subsequently, the antiacid therapy-monitoring property of Gly-PFOBs was assessed. The results demonstrated that the CEST signal

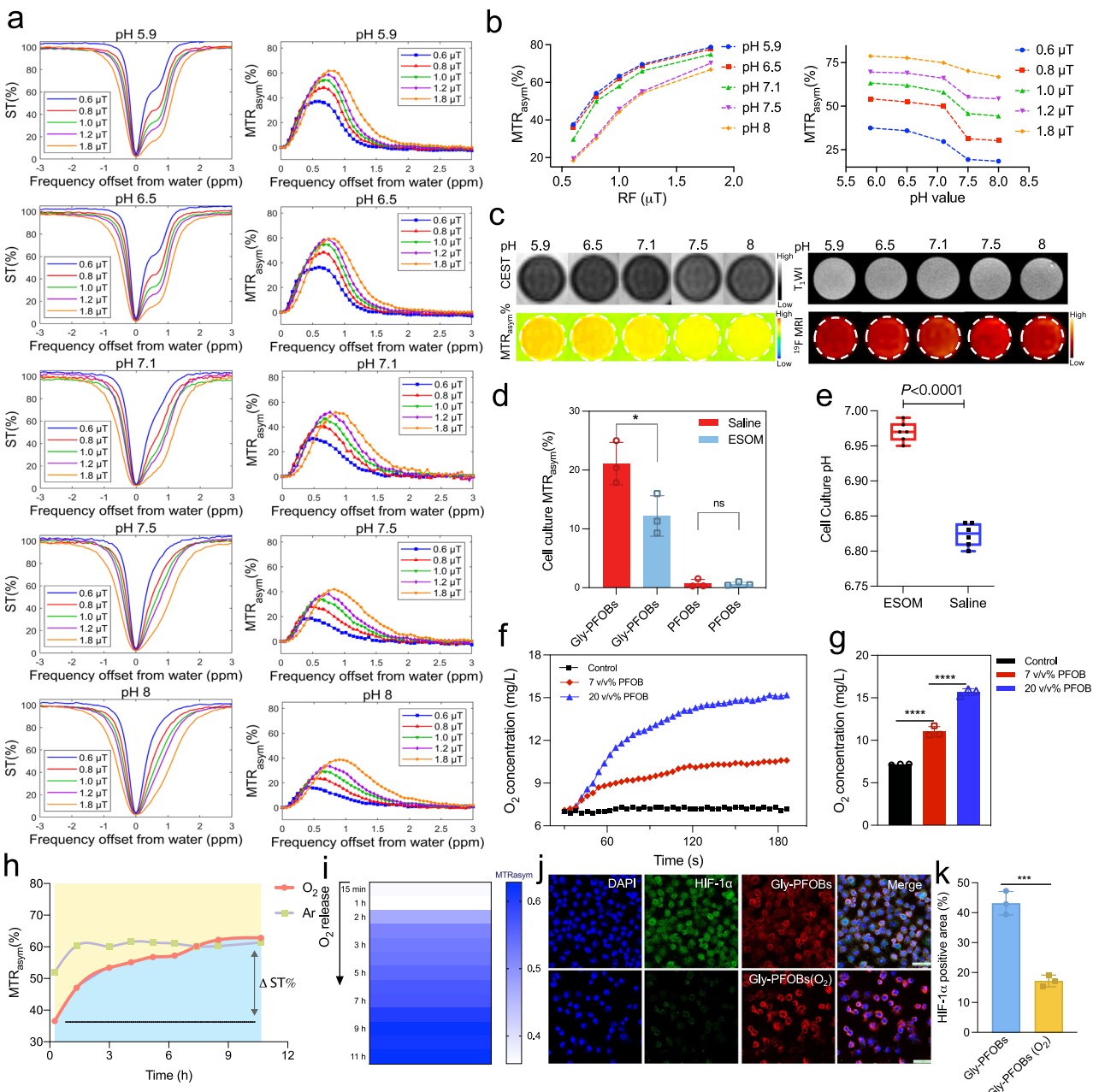

**Fig. 3 | In vitro pH and $O_2$ dual sensitivities and hypoxia alleviation by oxygenated Gly-PFOBs. a** Z-Spectra and $MTR_{asym}$ curve of the Gly-PFOBs at different pH values and saturation pulse powers (0.6 µT, 0.8 µT, 1.0 µT, 1.2 µT, and 1.8 µT, 5 s), and corresponding statistical analysis (**b**). **c** $^1$H-CEST MRI (1.0 µT, 5 s), $T_1$WI, and $^{19}$F-MRI results of Gly-PFOBs in different pH solutions. **a–c** Repeated 3 times independently with similar results. **d** Comparison of CEST signal intensity between Gly-PFOBs and PFOBs in the supernatant of NCI-H460 cells incubated with ESOM or saline, *$P < 0.05$ ($P = 0.0364$); n.s indicates no statistical significance ($P > 0.05$, $P = 0.9049$). Two-tailed, unpaired $t$ test. Data are presented as mean ± SD ($n = 3$ biologically independent samples). **e** The pH value of the supernatant of NCI-H460 cells incubated with ESOM or saline in pHmed for 2 h were measured by pH-sensitive optical microsensor, ****$P < 0.0001$. Two-tailed, unpaired $t$ test. Data are presented as mean ± SD ($n = 6$ biologically independent samples). In the boxplots, the center line, box limits, and whiskers denote the median, upper and lower quartiles, and 1.5 × interquartile range, respectively. **f** Measurement of oxygen loading and gradual release from PFOB saturated with oxygen in deoxygenated water and corresponding statistical analysis (**g**), ****$P < 0.0001$. One-way ANOVA with Sidak's multiple comparisons test. Data are presented as mean ± SD ($n = 3$ independent measurements). **h** CEST signal intensity changes of Gly-PFOBs with oxygen and argon (Ar) release over time. Data are presented as mean ± standard deviation (SD), $n = 3$ independent experiments. **i** CEST signal intensity changes of Gly-PFOBs with oxygen release. Repeated three times independently with similar results. **j** Immunofluorescence results of HIF-1α staining of NCI-H460 hypoxic cells co-incubated with Gly-PFOBs with or without oxygenation and corresponding statistical results (**k**). ***$P < 0.001$ ($P = 0.0005$). Two-tailed, unpaired $t$ test. Data are presented as mean ± SD ($n = 3$ biologically independent experiments). Scale bar: 50 µm. ESOM: esomeprazole. Source data are provided as a Source data file.

was decreased in the antiacid treatment group as the extracellular pH of the TME was manipulated by ESOM and increased after the antiacid therapy. In contrast, the PBS group presented a higher CEST signal due to the unmanipulated acidic TME. Also, the control probe PFOBs without the CEST signal failed to detect any signal difference (Fig. 4g).

Moreover, the glycerol also failed to visualize the antiacid therapy efficacy due to its fast metabolism property (Fig. 4h). The pH microsensor measurement results demonstrated significant differences in the tumor pH between ESOM and saline groups after three consecutive days of administration and were consistent with the MR imaging

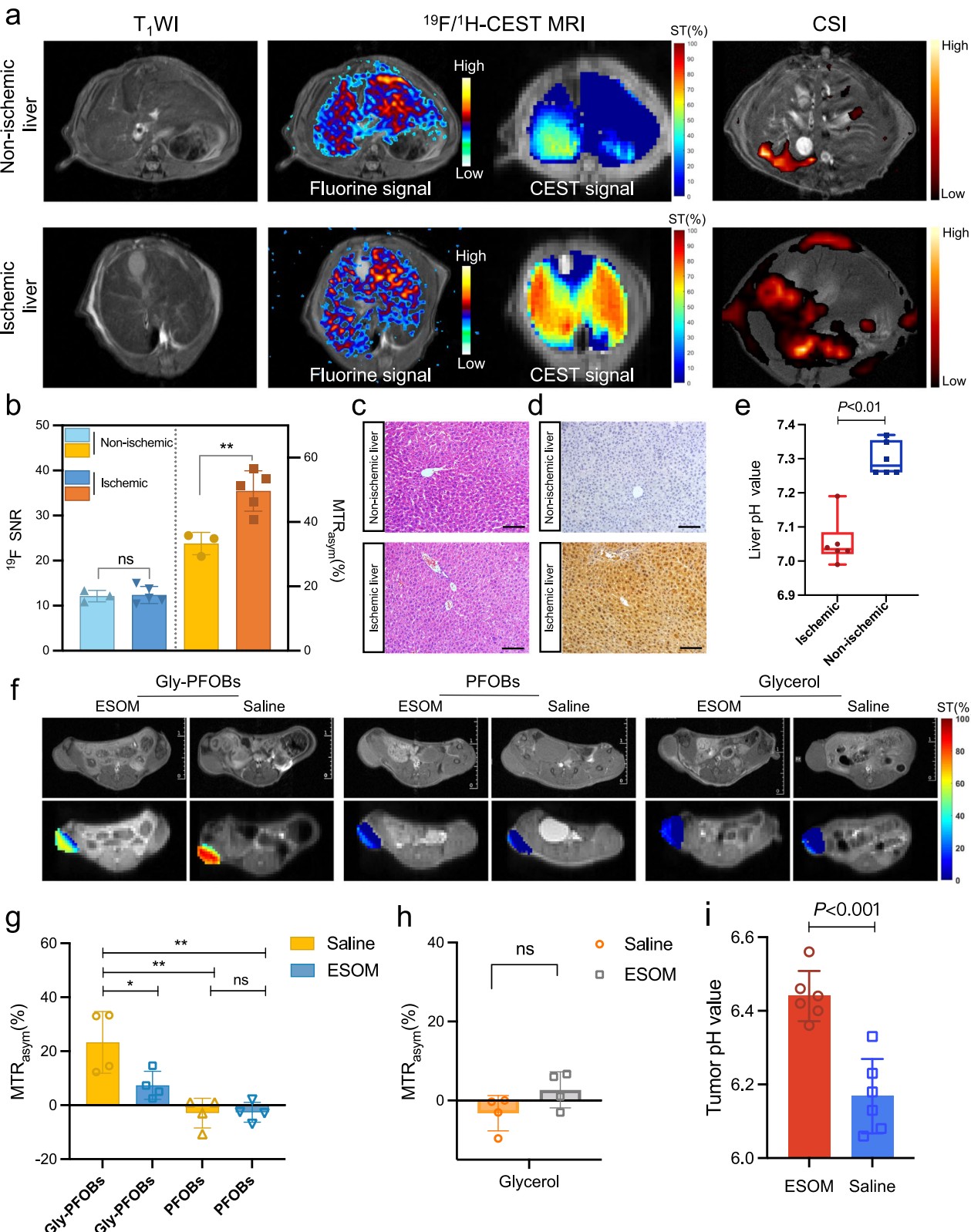

results ($\Delta$pH = 0.27 units, $P$ < 0.01, Fig. 4i). The average fluorescence intensity biodistribution percentage of the tumor and major organs 4 h post intravenous injection of Gly-PFOBs is shown in Supplementary Fig. 9a, b. The ex vivo fluorescence analysis revealed an accumulation of Gly-PFOBs in the tumor that provided adequate CEST signal required for exploring the acidic TME. The biodistribution of Gly-PFOBs was consistent with most conventional nanoprobes that were

mainly distributed in metabolic organs[42], such as the liver and spleen, and excreted through the biliary system.

To further explore the in vivo glycerol-retaining ability of Gly-PFOBs, NCI-H460 subcutaneous xenograft tumor tissues or BALB/c nude mice liver tissues were collected before and after intratumoral or intravenous injection of Gly-PFOBs at different time points (1 h, 2 h, 4 h, and 6 h), respectively. The tissue glycerol concentration was also

**Fig. 4 | Detection of in vivo pH changes. a** Representative $T_1$WI, $^{19}$F/$^{1}$H-CEST MRI, and CSI results following Gly-PFOBs injection. **b** Statistical results of $^{19}$F/$^{1}$H-CEST MR signal intensities. **P < 0.01(P = 0.0067), n. s., no significance. Two-tailed, unpaired $t$ test. Data are presented as mean ± SD ($n = 5$ mice were used for ischemic models; $n = 3$ mice used for Non-ischemic models as control). **c** H&E staining of the non-ischemic and ischemic liver, scale bar: 100 μm. **d** Pimonidazole hydrochloride immunohistochemical staining of the non-ischemic and ischemic liver, scale bar: 100 μm. **e** The pH values of the non-ischemic and ischemic liver were measured by pH-sensitive optical microsensor, **P < 0.01(P = 0.0022). Two-tailed Mann–Whitney $U$ test. Data are presented as mean ± SD ($n = 6$ biologically independent measurements). In the boxplots, the center line, box limits, and whiskers denote the median,

upper and lower quartiles, and 1.5 × interquartile range, respectively. **f** CEST MR images of Gly-PFOBs, PFOBs, and glycerol in monitoring antiacid therapy, and corresponding statistical analyses (**g**, **h**), *P < 0.05; **P < 0.01(*P = 0.0353, **P = 0.0011 Gly-PFOBs-saline vs. PFOBs-saline, **P = 0.0012 Gly-PFOBs-Saline vs. PFOBs-ESOM), One-way ANOVA, Tukey's multiple comparison test; n.s. indicates no statistical significance ($P > 0.05$, $P = 0.1136$). Two-tailed, unpaired $t$ test. Data are presented as mean ± SD ($n = 4$ mice). **i** pH determination in the TME of NCI-H460 subcutaneous tumor-bearing mice treated with ESOM or saline, ***P < 0.001(P = 0.0003). Data are presented as mean ± SD ($n = 6$ biologically independent measurements). CSI chemical shift imaging, SNR signal-to-noise ratio. Source data are provided as a Source data file.

determined by Free Glycerol Assay Kit. Initially, we ascertained the presence of endogenous glycerol levels in tumor and liver tissue, which were found to be 677.23 ± 21.94 nmol/g and 2429.15 ± 214.17 nmol/g, respectively. Following the exogenous introduction of Gly-PFOBs or free glycerol, the findings indicate that the exogenous free glycerol, being a hydrophilic substance, exhibits a tendency to rapidly clear from tissues irrespective of intratumoral or intravenous administration. Furthermore, it maintains a relatively low concentration comparable to the endogenous glycerol level at all subsequent time intervals (2 h, 4 h, and 6 h, except 1 h). Compared with free glycerol injection group, the clearance of glycerol happens more gradually in Gly-PFOBs intratumoral injection group. After 1 h of intratumoral injection of Gly-PFOBs, although the glycerol concentration decreased 17.67%, the concentration was still reached 18251 ± 475 nmol/g in NCI-H460 subcutaneous lung tumor tissue. This concentration was found to be 6.22 times higher than that observed in the free glycerol injection group (2932 ± 162 nmol/g). Similar findings were observed in liver tissue after intravenous injection of Gly-PFOBs (Supplementary Fig. 10a, b). These experimental results illustrated the glycerol-retaining ability of Gly-PFOBs.

Glycerol can exist in the body environment, involved in fat metabolism. The content of endogenous glycerol was also substantiated by the results of our experiments. Therefore, to further confirm our tested results, and to eliminate the influence of endogenous glycerol in the in vivo setting, we prepared $^{13}$C-Gly-PFOBs using $^{13}$C isotopically labeled glycerol and tested tissue glycerol content with carbon-13 nuclear magnetic resonance spectroscopy ($^{13}$C-NMR) and GC-MS. The injection method and animal model were same as above. The concentration of $^{13}$C-glycerol in NCI-H460 subcutaneous tumor tissues was 1360.9 ± 33.8 mg/L after 1 hour intratumoral injection of $^{13}$C-Gly-PFOBs, which was 7 times higher than that in free $^{13}$C-glycerol injection group (194.2 ± 10.94 mg/L). Subsequently, with the extension of time, there was a very slow decline of $^{13}$C-glycerol concentration in $^{13}$C-Gly-PFOBs injection group. Moreover, even after 6 hours of injection, the $^{13}$C-glycerol concentration in Gly-PFOBs injection group was still higher than that in the free $^{13}$C-glycerol group (555.79 ± 14.87 mg/L vs undetectable, 6 h). Similar observation was found in liver tissue after intravenous injection of $^{13}$C-Gly-PFOBs or free $^{13}$C-glycerol.

Even though the hepatic clearance of $^{13}$C-glycerol was observed to be more rapid following intravenous administration of $^{13}$C-Gly-PFOBs compared to subcutaneous tumors, the rate of $^{13}$C-glycerol clearance in the $^{13}$C-Gly-PFOBs injection group remained slower than that in the intravenous free $^{13}$C-glycerol injection group (Supplementary Fig. 10c–f). Taken together, above results have confirmed the glycerol-retaining ability of Gly-PFOBs and have suggested that glycerol is not that rapidly diffused and cleared from the Gly-PFOBs.

**ORTW and RT outcomes in the NSCLC xenograft tumor model**
We explored the potential utilities of dual pH- and oxygen-sensitive Gly-PFOBs in providing pivotal optimized radiotherapy time window (ORTW) information for RT. For this, the NCI-H460 lung cancer xenograft mouse models were subjected to $^{1}$H-CEST and BOLD fMRI before and after the intratumoral injection of fully oxygenated Gly-

PFOBs ($O_2$) or PFOBs. As shown in Fig. 5a–c, at initial 1 h, the CEST signal of the tumor region injected with Gly-PFOBs ($O_2$) increased from 0% to 25.45% (Gly-PFOBs ($O_2$) vs. PFOBs ($O_2$) at 1 h, $P < 0.0001$, at 30 min, $P < 0.05$), then the signal intensity decreased at 2 h from 25.45% to 16.27% (Gly-PFOBs ($O_2$) vs. PFOBs ($O_2$), $P < 0. 01$). Intriguingly, the CEST signal showed an uptrend at the 3 h time point from 16.27% to 25.28% (Gly-PFOBs ($O_2$) vs. PFOBs ($O_2$), $P < 0. 0001$). Compared with mice injected with Gly-PFOBs ($O_2$), there were no significant changes in corresponding measurements in mice injected with PFOBs ($O_2$) (Fig. 5b, c).

We reasoned that oxygen released from Gly-PFOBs ($O_2$) led to the initial CEST signal increase in 1 h as evidenced by Gly-PFOBs ($O_2$) phantom imaging studies. Since, oxygen could relieve tumor hypoxia, which would further induce the pH arises in the TME[10,43–45], consequently, the CEST signal of Gly-PFOBs decreased in the high pH environment, leading to the signal drop at the 2 h time point. Subsequently, the CEST signal of Gly-PFOBs was again increased at the 3 h time point. This could be due to the unsustainability of the oxygen supply of Gly-PFOBs ($O_2$) and the metabolic demand for oxygen exceeding its availability. Therefore, the acidic TME, caused by the accumulation of acidic metabolites produced by anaerobic glycolysis[10], could increase the CEST signal intensity derived from Gly-PFOBs.

The pH-sensitive optical microsensor was used to confirm the accuracy of Gly-PFOBs in tumor acidic microenvironment imaging. The results were consistent with the $^{1}$H-CEST imaging results, further confirming that the pH in the tumor region rose slowly and peaked at 2 h (ΔpH of pre and 2 h time points was 0.47 units, $P < 0.0001$) but then decreased at 3 h time point (ΔpH of pre and 3 h time points was 0.27 units, $P < 0.0001$, Fig. 5d).

We also performed the BOLD fMRI to confirm the accuracy of Gly-PFOBs in tumor hypoxic microenvironment imaging. The BOLD fMRI results displayed in Fig. 5a, b showed that the $T_2$*values of the tumor in Gly-PFOBs ($O_2$) and PFOBs($O_2$) groups initially increased (pre vs. 2 h, $P < 0.05$), followed by a downward trend after 2 h (Fig. 5e). Importantly, the corresponding pimonidazole hydrochloride immunofluorescence hypoxia staining was also consistent with the BOLD fMRI imaging results, where the lower fluorescence intensity of Gly-PFOBs ($O_2$) in tumor sections at 30 min and 1 h time points may be caused by the inhomogeneous dispersion of probes after intratumoral injection. (Supplementary Fig. 11a–c). These observations indicated that oxygen released from Gly-PFOBs ($O_2$) into the tumor could relieve hypoxia initially. However, oxygen was then dispersed through blood circulation, leading to a decreasing trend in $T_2$*values at the tumor site and suggesting that Gly-PFOBs ($O_2$) could relieve tumor hypoxia within specific time period.

Our data verified that oxygenated Gly-PFOBs ($O_2$) could effectively reveal the optimal time window for radiotherapy. Specifically, from the imaging analysis shown in Fig. 5c, we could identify the optimal and most sensitive time window for NCI-H460 solid tumor RT to be between 1 h–2 h after oxygenated Gly-PFOBs ($O_2$) injection, as the tumor oxygenation was significantly improved and hypoxia condition altered.

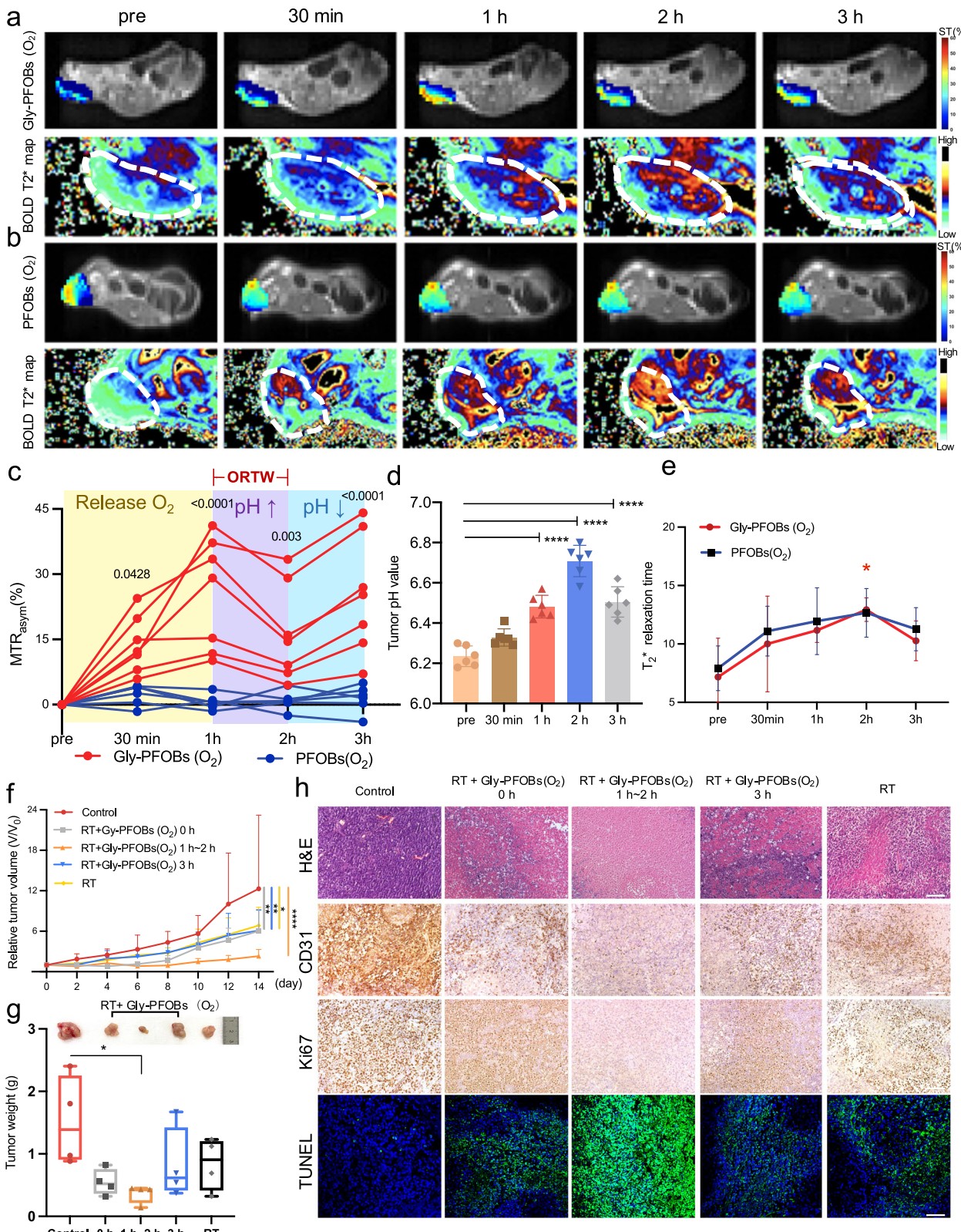

Based on the ORTW information obtained from Gly-PFOBs ($O_2$) dynamic CEST MR imaging, further in vitro and in vivo validation studies were conducted. As shown in Supplementary Fig. 12a, b, the NCI-H460 control group cells did not display any apparent apoptosis, while the RT alone group presented significant apoptosis. A high level of apoptosis was observed in the RT+ Gly-PFOBs 0 h and RT+ Gly-PFOBs 3 h NCI-H460 groups, which was even higher in the RT+Gly-PFOBs ($O_2$) 1 h-2 h group.

For animal experiments, we used five groups of NCI-H460 tumor-bearing BALB/c nude mice at different times after the probe injection: Group I: PBS control; Group II: RT+ Gly-PFOBs ($O_2$) 0 h; Group III: RT+ Gly-PFOBs ($O_2$)1 h-2 h; Group IV: RT+ Gly-PFOBs ($O_2$) 3 h; Group V: RT alone. As displayed in Fig. 5f, RT alone resulted in moderate tumor growth inhibition (44.72%, Group V versus I). Notably, Group II and Group IV effectively inhibited tumor growth by 51.1% (Group II versus

**Fig. 5 | ORTW and RT outcomes in NSCLC xenograft tumor model. a** CEST and BOLD MR imaging of BALB/c nude mice bearing NCI-H460 lung xenografts after intratumoral injection of oxygenated Gly-PFOBs(O₂) or PFOBs(O₂) (**b**) over time. **c** Dynamic CEST MR signal changes of tumor region after Gly-PFOBs (O₂) or PFOBs(O₂) injection. *$P < 0.05$ ($P = 0.0428$); **$P < 0.01$ ($P = 0.0030$); ****$P < 0.0001$, Gly-PFOBs (O₂) vs. PFOBs(O₂). Two-way ANOVA with Sidak's multiple comparisons test. Data are presented as mean ± SD, $n = 7$ mice for Gly-PFOBs (O₂) group and $n = 6$ mice for PFOBs(O₂) group. **d** Determination of the pH changes in the TME before and after Gly-PFOBs (O₂) probe injection after fully oxygenated. ****$P < 0.0001$. One-way ANOVA with Dunnett's multiple comparisons test. Data are presented as mean ± standard deviation (SD) ($n = 6$ biologically independent measurements). **e** Monitoring of tumor tissue oxygenation by calculating changes in $T_2$*values. *$P < 0.05$ ($P = 0.0291$). Two-way ANOVA with Sidak's multiple comparisons test.

Data are presented as mean ± SD ($n = 5$). **f, g** Statistical results of relative tumor volume and tumor weight (**e**) of each treatment group. *$P < 0.05$ ($P = 0.0252$, RT vs. control, 14d), **$P < 0.01$ ($P = 0.0060$, RT+Gly-PFOBs (O₂) 0 h *vs.* control, 14d), **$P < 0.01$ ($P = 0.0069$, RT+Gly-PFOBs (O₂) 3 h *vs.* control, 14d), ****$P < 0.0001$ (RT +Gly-PFOBs (O₂) 1-2 h *vs.* control, 14d). Two-way ANOVA, Tukey's multiple comparisons test. *$P = 0.0245$ for **g**, RT+Gly-PFOBs (O₂) 1-2 h *vs.* control, 14d, one-way ANOVA, Tukey's multiple comparisons test. Data are presented as mean ± standard deviation (SD) ($n = 4$ mice). In the boxplots, the center line, box limits, and whiskers denote the median, upper and lower quartiles, and 1.5 × interquartile range, respectively. **h** Representative H&E, CD31, Ki67, and TUNEL-related antigen staining in different treatment groups ($n = 4$), scale bar: 100 μm. ORTW optimized radiotherapy time window, BOLD blood oxygen level-dependent, RT radiotherapy. Source data are provided as a Source data file.

Group I, $P < 0.01$) and 50.04% (Group IV versus Group I, $P < 0.01$). The highest effect of 81.31% inhibition of tumor growth was found in the RT + Gly-PFOBs (O₂) 1 h-2 h Group III compared with Group I ($P < 0.0001$). The tumor weight measurement further proved that the antitumor efficacy was improved markedly in Group III (Group III versus Group I, $P < 0.05$) (Fig. 5g). No significant body weight fluctuation was observed in any of the groups (Supplementary Fig. 13a). The H&E, CD31, Ki67, and TUNEL-related antigen staining confirmed that RT+ Gly-PFOBs (O₂) 1 h-2 h (Group III) treatment produced the most significant inhibition of tumor cells among all groups, while RT+ Gly-PFOBs (O₂) 0 h and RT+ CESTs (O₂) 3 h (Group II and Group IV) induced moderate levels of tumor cell apoptosis and necrosis (Fig. 5h and Supplementary Fig. 13b–d). These results were in agreement with the tumor growth data. Thus, the synergistic RT-enhancing effect in Group III could be attributed to the optimized time window for RT as the oxygen release from the Gly-PFOBs (O₂) in that period (1 h-2 h) greatly alleviated tumor hypoxia and substantially enhanced radiosensitivity synergistically.

## ORTW and RT outcomes in SCLC metastasis in the liver

SCLC is an aggressive cancer characterized by a high risk of metastasis and is among the deadliest solid tumor malignancies due to its refractory nature to current treatments[46]. Based on the Gly-PFOBs results, we explored if this strategy could improve RT efficacy in SCLC. We established the H209 SCLC liver metastasis mouse model to test this and performed ¹⁹F/¹H-CEST imaging after Gly-PFOBs, Gly-PFOBs (O₂), or PFOBs injection. From the imaging results in Fig. 6a, several key observations could be summarized: First, the CEST signal intensity of Gly-PFOBs and Gly-PFOBs (O₂) probes in the liver reached its peak value at 1 h. Second, the signal intensity of Gly-PFOBs in H209 liver metastasis reached its peak value at 2 h 30 min followed by a gradually decreasing tendency due to the Gly-PFOBs metabolism. And third, the "M-type" CEST signal intensity curve illustrated that the CEST signal of Gly-PFOBs (O₂) in H209 liver metastasis increased slowly at the initial 1 h, possibly caused by the slow release of oxygen, weakening the CEST signal intensity of the probe. However, at 1 h 30 min, the signal intensity was close to that of the Gly-PFOBs without oxygen saturation, indicating the complete release of oxygen. Then the signal decrease at 2 h 30 min might be related to the improvement of tumor hypoxia and increased extracellular pH. Subsequently, the reappearance of hypoxia in tumor cells and the decrease in extracellular pH resulted in an increase in the signal at 3 h, followed by a downward trend for metabolic reasons. These data indicated that following Gly-PFOBs (O₂) injection, the ORTW for H209 liver metastasis was between 1 h 30 min to 2 h 30 min. Therefore, we chose 2 h 30 min as the best option for conducting RT for H209 liver metastasis. As expected, PFOBs did not present any signal trend in the liver or liver metastasis. BOLD fMRI was applied to examine hypoxia changes in the TME, and the gold standard pH-sensitive chemical optical microsensor was used to validate the accuracy of Gly-PFOBs in monitoring pH fluctuations. As shown in

Fig. 6b, the results were highly supportive of the dynamic imaging results.

In vivo ORTW-guided radiotherapy results showed that compared with the control, RT alone and RT+ Gly-PFOBs (O₂) 0 h resulted in moderate and similar therapeutic effects ($P < 0.01$ and $P < 0.001$, respectively). Significantly, the RT+Gly-PFOBs (O₂) 2 h 30 min group effectively inhibited metastasis of H209 SCLC to other regions of the liver than the RT+ Gly-PFOBs (O₂) 1 h group and demonstrated a significant difference when compared with RT alone group ($P < 0.01$) (Fig. 6c, d). The corresponding $T_1$ weighted image is shown in Supplementary Fig. 14). No significant body weight fluctuation was observed in any of the groups (Fig. 6e).

In summary, the antitumor and antimetastatic efficacy was maximized in the ORTW-guided RT+ Gly-PFOBs (O₂) 2 h 30 min group (Supplementary Fig. 15). H&E staining also confirmed that among all groups, the RT+ Gly-PFOBs (O₂) 2 h 30 min group produced the highest therapeutic effects against H209 SCLC liver metastasis (Fig. 6f). Supplementary Fig. 16a, b show the biodistribution of Gly-PFOBs at various time points after injection. These data demonstrated the advantages of Gly-PFOBs in providing ORTW for RT with substantially enhanced radio-sensitive properties against advanced metastatic tumors.

## Gly-PFOBs cytotoxicity and biocompatibility

Gly-PFOBs cytotoxicity was assessed in NCI-H460 lung cancer cells using the MTT cell viability assay. NCI-H460 cells exhibited negligible cytotoxicity after 24 h and 48 h at indicated high concentrations (Supplementary Fig. 17a). Moreover, no noticeable body weight loss was observed in healthy BALB/c nude mice treated with Gly-PFOBs (Supplementary Fig. 17b). In vivo toxicity of Gly-PFOBs in healthy BALB/c nude mice was assessed by histological analysis. Major organs (heart, liver, spleen, lung, and kidneys) were harvested 14d after intravenous injection of the probes. Supplementary Fig. 17c shows that Gly-PFOBs did not induce apparent pathological changes, including cytoplasm loss, cell atrophy, or inflammation, suggesting excellent biocompatibility. Also, a standard biochemical index analysis revealed no changes in blood alanine aminotransferase (ALT), alkaline phosphatase (ALP), and aspartate aminotransferase (AST) levels between the Gly-PFOBs-injected and saline control groups, demonstrating healthy liver function. Blood urea (UREA) and creatinine (CREA) levels also remained normal in the Gly-PFOBs-injected group, indicating normal kidney function. Finally, no significant changes were detected in white blood cells (WBCs), red blood cells (RBCs), hemoglobin (HGB), hematocrit (HCT), mean corpuscular volume (MCV), mean corpuscular hemoglobin (MCH), mean corpuscular hemoglobin concentration (MCHC), platelets (PLT), mean platelet volume (MPV), or platelet distribution width (PDW), showing that the mice had no apparent infections or defects in the physiological regulation of their immune system (Supplementary Fig. 17d).

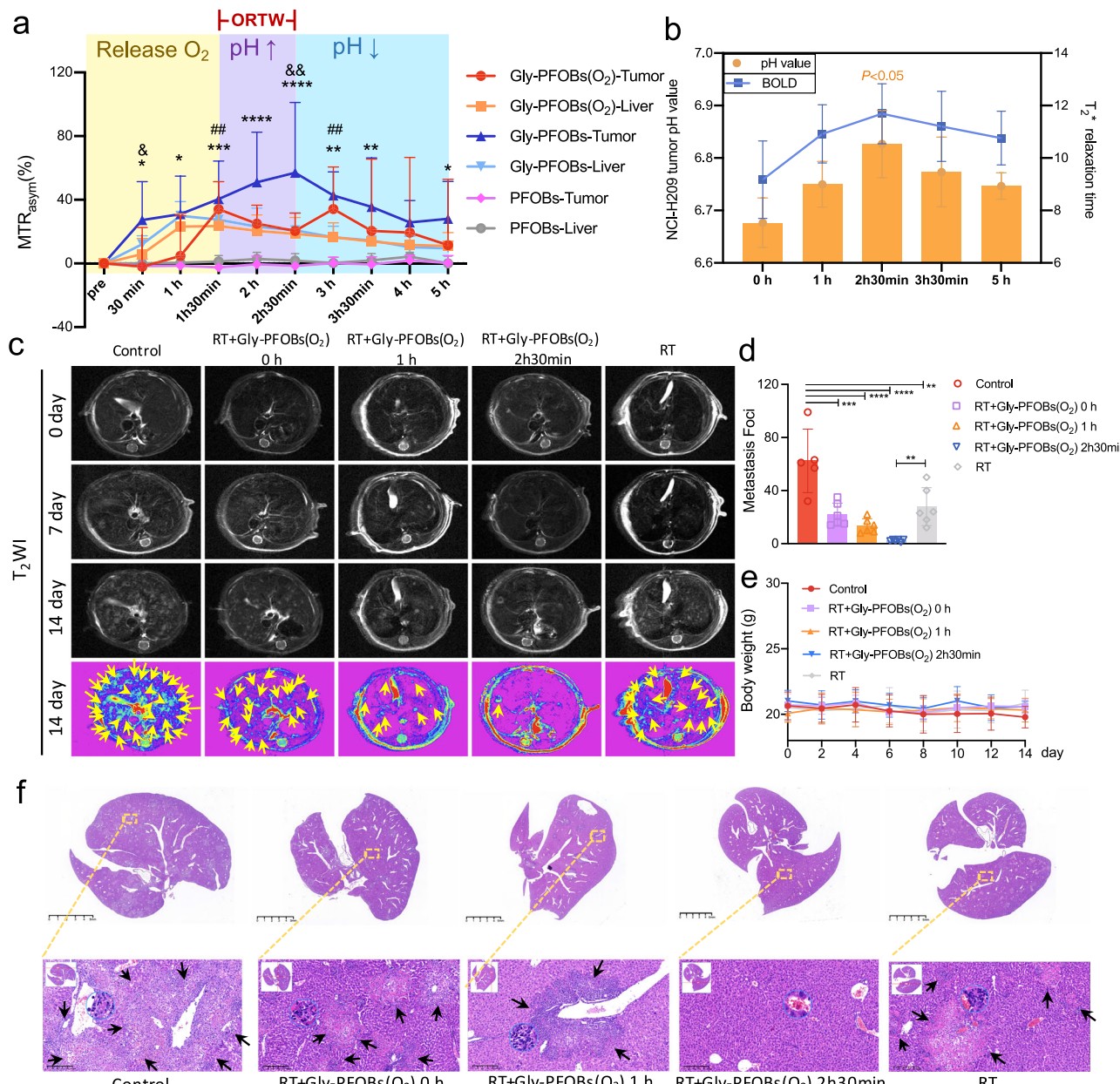

**Fig. 6 | ORTW and RT outcomes in SCLC metastasis in the liver. a** Dynamic CEST MR "M-type" signal changes curve of H209 SCLC liver metastasis and normal liver tissue after the injection of Gly-PFOBs(O₂), Gly-PFOBs, and PFOBs. *Gly-PFOBs-Tumor *vs.* PFOBs-tumor,*$P$ = 0.035, 30 min; *$P$ = 0.0109, 1 h; ***$P$ = 0.0002, 1h 30 min; ****$P$ < 0.0001, 2 h, 2h 30 min; **$P$ = 0.0002, 3 h; *$P$ = 0.0028, 3h 30 min; *$P$ = 0.0476, 5 h; # Gly-PFOBs(O₂)-Tumor *vs.* PFOBs-tumor, ##$P$ = 0.0024, 1h 30 min; ##$P$ = 0.0061, 3 h; & Gly-PFOBs(O₂) -Tumor *vs.* Gly-PFOBs-tumor, &$P$ = 0.0298, 30 min; &&$P$ = 0.0023, 2h 30 min. Two-way ANOVA, Tukey's multiple comparisons test. Data are presented as mean ± SD ($n$ = 6 mice). **b** Determination of the pH changes and $T_2$*values in TME before and after Gly-PFOBs(O₂) (pH of 2 h 30 min *vs.* 0 h, *$P$ = 0.0331). One-way ANOVA, Tukey's multiple comparisons test. Data are presented as mean ± SD ($n$ = 3 biologically independent measurements). **c** Representative T₂WI MR images of mice liver in each treatment group. Yellow

arrow points to H209 SCLC liver metastasis. **d** H209 SCLC liver metastasis foci on each liver were counted. **$P$ = 0.0013; ***$P$ = 0.0002; ****$P$ < 0.0001. One-way ANOVA, Tukey's multiple comparisons test. Data are presented as mean ± standard deviation (SD), $n$ = 6 mice for all groups except control group ($n$ = 5). **e** Body weight changes of mice in each treatment group. Data are presented as mean ± standard deviation (SD), $n$ = 3 mice. **f** Representative H&E-stained liver slices collected from mice after the indicated treatment on day 14. Black arrow points to H209 SCLC liver metastasis. Scale bar: 5 mm, and corresponding enlarged lung metastasis sections. Scale bar: 200 μm. $n$ = 5 biologically independent samples for all groups except control group ($n$ = 3). ORTW optimized radiotherapy time window, BOLD blood oxygen level-dependent, RT radiotherapy. Source data are provided as a Source data file.

## Discussion

RT is an extensively used modality in current clinical cancer therapy; however, tumor hypoxia in the TME contributes to radiation resistance[47,48]. The acidic TME is partially due to metabolic shifts in response to tumor hypoxia, also known as the Pasteur effect[49]. The designing of more effective and precision radiotherapy strategies requires a deeper molecular understanding of TME, its clinical relevance to radiotherapy resistance, and the influence of hypoxia on malignant cells[50,51]. Instead of the standard treatment approaches[52–54], developing new precision RT with high radiosensitivity, such as the determination of the optimal time window for tumor radiotherapy following tumor oxygenation, is highly desirable.

Since hypoxia in solid tumors often renders them resistance to radiation therapy, the detection of hypoxia status in malignant tumors

could conceivably be significant for this treatment modality. Various PET hypoxia tracers, such as [18 F]- fluoromisonidazole ([18]F-FMISO) and [18 F]-fluoroazomycin arabinoside ([18]F-FAZA), are widely used to characterize the hypoxia status of tumors in clinical settings and to delineate the target area for radiotherapy[55,56]. However, these radio-tracers mostly provide limited or single metabolic information and require much time for processing and interpretation[57]. Also, due to the radiation exposure associated with PET/CT imaging and the high cost of tracer production, the technique has limited applicability in determining the hypoxia status of tumors at multiple time points and providing ORTW information.

Previously, pH-sensitive CEST MR imaging nanoparticles were developed to detect the acidic tumor microenvironment and indirectly reflect tumor hypoxia[58–60]. CEST constitutes a powerful sensitivity enhancement mechanism in which the signal of low-concentration solutes can be amplified and further visualized through the bulk water signal[38,58,61]. In this study, different from the single pH sensitivity agents, we have developed Gly-PFOBs with $^{19}$F/$^{1}$H-CEST dual-modality imaging to dynamically provide important metabolic changes of the malignant tumor microenvironment, enabling synchronous pH and oxygen molecular imaging on a single MR machine. This complementary approach was also efficacious in illustrating the complexity and spatiotemporal heterogeneity of the oxygenation status and acidic microenvironment in solid tumors, providing integrated imaging information of precise ORTW. In this process, a pH-sensitive chemical optical pH-1 microsensor was applied to verify the accuracy of the $^{1}$H-CEST MR imaging results, especially the dynamic CEST signal trend.

Furthermore, radiotherapy of multiple lesions is more challenging than single lesions. Usually, radiosensitizers are injected locally in the tumor site, which is not suitable for deep and multiple metastatic lesions[21]. Our strategy, except single lesion precision RT processing, also enables simultaneous multi-focal RT by revealing the ORTW and displaying the characteristics of TME in multiple lesions achieved after the intravenous injection of Gly-PFOBs ($O_2$).

Finally, most radiosensitizers are inorganic materials, which may have potential long-term toxicity. Clinical radiotherapy sensitizers, such as hyperbaric oxygen and oxygen mimetics, have severe toxicities, especially neurotoxicity, making them undesirable for clinical applications[21]. Hence, there is a need for novel drugs and/or strategies with radio-sensitization capability but low systemic toxicity. In this context, PFCs are inert organic compounds used as "artificial blood" that improve tissue oxygenation and have been widely used in the clinic for various purposes, including artificial blood substitution, organ preservation, ultrasound, and $^{19}$F magnetic resonance imaging[27,28]. Perfluorooctylbromide (PFOB) used in this study stands out among PFCs for medical use as it is non-toxic, high stability, inertness, and an acceptable excretion profile[30]. Therefore, the PFC-based Gly-PFOBs nanoplatform presented in this study has desirable biocompatibility and holds great potential in clinical translation.

One of the limitations of our work is that all imaging studies were performed on the 9.4 T MRI system. However, to enhance the clinical applicability of Gly-PFOBs and their compatibility with the conventional 3.0 T MRI system, it is necessary to further optimize and broaden the narrow chemical shift of Gly-PFOBs (0.68 ppm). This is crucial as a significant chemical shift difference offers potential advantages for utilization at lower magnetic fields and addresses the inherent challenge of relatively low sensitivity in MRI-guided irradiation. Furthermore, the development of saturation schemes, readout patterns, and saturation-editing techniques, in conjunction with advanced postprocessing algorithms, has the potential to enhance the sensitivity of CEST imaging. Consequently, future research should focus on conducting more extensive investigations into Gly-PFOBs with greater chemical shift frequency modification, as well as integrating updated CEST techniques[62]. Moreover, although the duration

of a single radiotherapy session for a clinical cancer patient is generally a few minutes to 20 minutes, for practical applications, there still needs to be a bit of time for irradiation field design and patient set up. From pH and $O_2$ dual-sensitive Gly-PFOBs ($O_2$) $^{19}$F/$^{1}$H-CEST dual-modality MR imaging, we could notice that ORTW was related to dynamic changes of tumor hypoxia and acidic microenvironment, and the time window width was not that large. Thus, rapid radiation field design for radiation therapy is needed. Another area requiring in-depth study is the optimization of the precise radiotherapy strategy through the precise spectral-temporal guidance of MRI. Gly-PFOBs ($O_2$) $^{19}$F/$^{1}$H-CEST dual-modality MR imaging strategy is particularly well suited to be combined with the current cutting-edge MR-guided RT system (Elekta Unity, Sweden). Further exploitation of this approach would hold great potential for future diagnostic imaging and precision RT. The next step would be to utilize ORTW information provided by $^{19}$F/$^{1}$H-CEST MR molecular imaging to precisely delineate targeted areas. Of course, not all clinical centers are equipped with this kind of high-end equipment, thus, optimizing and modifying Gly-PFOBs remains an important work in the next phase of research to make them better distributed in tumor lesions and function for longer periods of time. The faster $^{19}$F/$^{1}$H-CEST dual-modality MR imaging sequences are also ideal for clinicians to obtain ORTW information. Finally, our study has delineated the excellent therapeutic effect of Gly-PFOBs in molecular imaging-guided lung cancer precision RT. Application of Gly-PFOBs in other cancers and oxygen- or pH-related diseases would be a subject worthy of further investigation.

The need for successful pharmacological interventions and integration into combined therapy modalities is due to the complexity and spatiotemporal heterogeneity of tumor hypoxia and is also related to the slow image-oriented development of radiation oncology. In this work, we provided an integrated, imaging-guided precision radiotherapy approach with better translational potential by utilizing the multifunctional advantages of PFC-based nano-platforms. Furthermore, we have conducted a pioneering study that offers a comprehensive analysis of the CEST signal properties demonstrated by glycerol and explored the impact of oxygen on the CEST signal intensity of Gly-PFOBs. Overall, the developed pH- and $O_2$-sensitive Gly-PFOBs with $^{19}$F/$^{1}$H-CEST MR dual-modality imaging properties, with superior efficacy and biosafety, were successfully employed in the highly sensitized imaging-guided lung cancer precision RT, establishing a strong precedence for precision radiotherapy in the future.

## Methods

### Reagents and animals

Unless otherwise stated, all solvents and reagents were purchased from Aldrich Chemical Co. and used as received. The perfluorooctylbromide, (PFOB, $C_8BrF_{17}$), also known as perflubron, was obtained from Exfluor Research Corporation (Round Rock, TX) and used as acquired. Phospholipids were purchased from Avanti Polar Lipids, Inc. (Alabama, USA). Mice were purchased from Vital River Laboratory Animal Technology Co. Ltd. (certificate number: SCXK (Jing) 2016-0006, Beijing, China) and housed in a special pathogen-free (SPF) barrier facility in groups of 4–5 per ventilated cage with ad libitum access to food and water and kept in a 12 h light/dark cycle. All animal procedures were carried out with the approval (permit 2022-DWSYLLCZ-67) from the Animal Ethics Committee of Harbin Medical University (Harbin, China) and were in agreement with the guidelines of the Institutional Animal Care and Use Committee.

### Preparation of Gly-PFOBs and PFOBs

The lissamine rhodamine B sulfonyl labeled perflubron-based Gly-PFOBs were prepared as described previously[63]. The purpose of incorporating lissamine rhodamine B sulfonyl was to investigate cellular uptake through confocal fluorescence microscopy and to examine ex vivo biodistribution using IVIS Imaging. The components of Gly-

PFOBs were 20% (v/v) PFOB, 76% (v/v) ultra-pure water, 2% (v/v) glycerol or $^{13}$C-glycerol (for tissue $^{13}$C-NMR and GC-MS analysis), and 2% (w/v) of a surfactant. The surfactant (lipids) included 88.9 mol% dipalmitoyl phosphatidylcholine (DPPC), 1 mol% 1,2-dipalmitoyl-sn-glycero-3-phosphoethanolamine (DPPE), 10 mol% cholesterol, and 0.1 mol% lissamine rhodamine B sulfonyl (16:0 LissRhod PE). Briefly, the lipids were dissolved in a mixture of methanol and chloroform, filtered through a small bed of cotton, evaporated under reduced pressure using a rotary evaporator at 45 °C to form a thin film, and then further dried in a vacuum oven (45 °C) for 24 h. The resuspended surfactant was combined with PFOB, water, and glycerol or $^{13}$C-glycerol according to the above proportions. The solution was extruded through an Avanti Mini Extruder (Alabaster, Alabama) with a 200 nm nucleopore polycarbonate membrane. The completed emulsions were placed in crimp-sealed vials, blanketed with argon, and stored at 4 °C until use.

The preparation procedure for perflubron-based nano-molecular imaging probes (PFOBs), which are CEST signal-free control probes, was identical to that of Gly-PFOBs. However, there were slight variations in the components used. Specifically, PFOBs consisted of 20% (v/v) PFOB, 78% (v/v) ultra-pure water, and 2% (w/v) surfactant, without the addition of glycerol. Consequently, two types of probes were prepared: Gly-PFOBs and PFOBs.

### Characterization of Gly-PFOBs and PFOBs

Hydrodynamic diameter distribution and zeta potential (ζ) of Gly-PFOBs and PFOBs were determined by dynamic light scattering (DLS) with a Malvern Nano ZS Zetasizer (Malvern Instruments Ltd, Malvern, UK) at different temperatures and time intervals. All determinations were made in multiples of three consecutive measurements. The absorbance of lissamine rhodamine B sulfonyl was measured using a multi-plate UV Absorbance Spectrophotometer (BioTek, Winooski, VT). The morphology and microstructure of Gly-PFOBs and PFOBs were determined by transmission electron microscopy (TEM) on a Hitachi TEM system at an accelerating voltage of 80 kV. Briefly, a drop of the sample was deposited on a carbon grid coated with copper, and the excess sample was drawn off with filter paper 1 min later and was then left for 5 min for drying. The TEM element mapping analysis of Gly-PFOBs was performed using FEI Talos F200X Super-X SDD detector at an acceleration voltage of 200 kV.

### In vitro MR imaging studies

In vitro and in vivo MR images were performed on a 9.4 T MRI scanner (Bruker BioSpin 94/20 USR system, Germany). The Z-spectra were interpolated by smoothing splines to identify the zero offset on a pixel-by-pixel basis of the bulk water and to assess the correct saturation frequency (ST%) value over the entire range of frequency offsets investigated.

The CEST contrast is usually characterized as saturation transfer (ST) using asymmetry analysis. Thus, Z-spectrum and magnetization transfer ratio asymmetry ($MTR_{asym}$) curve were used to assess CEST signal properties of the probe, in which the water proton signal after radio frequency (RF) saturation is normalized by the control water signal without RF saturation, plotted as a function of saturation frequency (ST%), and the $MTR_{asym}$ curve was defined by using the equation: $MTR_{asym} = (S^{-\Delta\omega} - S^{-\Delta\omega})/S_O$, where $S^{-\Delta\omega}$ and $S^{\Delta\omega}$ are reference and label signals of RF saturation at $-\Delta\omega$ and $\Delta\omega$, respectively, and $\Delta\omega$ is the labile proton frequency shift from the water resonance (i.e., +0.68 ppm for Gly-PFOBs). $S_O$ is the intensity of the bulk water CEST MR signal after irradiation at $-\Delta\omega$. Different glycerol concentrations of CEST probes (208 mM, 425 mM, and 623 mM) were prepared to investigate the CEST signal properties of Gly-PFOBs and test with different pre-saturation pulse powers (0.6 μT, 0.8 μT, 1.0 μT, 1.2 μT, and 1.8 μT) and durations (1 s, 2 s, 3 s, and 5 s). This could also be used to derive the pulse powers and pulse durations for the desired saturation level in μT

and s. For pH sensitivity evaluation of Gly-PFOBs, different pH solutions were prepared by using PBS with pH titrated to 5.9, 6.5, 7.1, 7.5, and 8.0, and the concentration of Gly-PFOBs remained the same (208 mM based on glycerol) to eliminate the concentration influence before CEST scanning. For exploring the oxygen sensitivity of Gly-PFOBs, CEST sequence scanning was performed immediately after the CEST probe was fully saturated with oxygen for 15 min (saturation pulse $B_1 = 1$ μT for 5 s). The temperature was set at 21 °C during scanning. For in vitro experiments, an echo planar imaging (EPI) sequence was used with the following parameters: TR = 6.0 s, TE = 22.2 ms, averages = 31, repetitions = 201, slice thickness = 1 mm, FOV = 35 × 35 mm, and image size = 96 × 96. Raw CEST images were acquired at varying saturation offset frequencies from −6 to +6 ppm regarding water peak with a step-size of 24 Hz, and different saturation powers (0.6 μT-1.8 μT) and saturation pulse durations (1 s-5 s) were applied to find the optimized conditions. The CEST MRI data were processed using the manufacturer-specific software (Image Display and Processing tool, Bruker Paravision 6.0.1, Germany) and Matlab (version 2020a, Mathworks). CEST analysis on Matlab proceeded using existing code shared online (https://github.com/cest-sources and https://godzilla.kennedykrieger.org/CEST).

### Cell cultures

The non-small cell lung cancer (NSCLC) cell lines NCI-H460 (bio-69123) and small cell lung cancer (SCLC) cell lines NCI-H209 (bio-69576) were purchased from ATCC and authenticated by STR profiling. Both lung cancer cells were cultured in the RPMI 1640 medium supplemented with 10% fetal bovine serum (FBS) at 37 °C with 5% $CO_2$ humidified atmosphere. All cell lines were tested routinely for mycoplasma contamination with a universal mycoplasma detection kit (30–1012 K; American Type Culture Collection, Manassas (ATCC), VA).

### Antiacid treatment of cells

As the previously described pHmed assay[64], the pHmed (80% normal saline, 10% unbuffered RPMI 1640, and 10% FBS) solution minimized the buffering activity of phosphate and bicarbonate in the medium but contained sufficient nutrients and growth factors to support cell growth (data not shown). Around 70% confluency, NCI-H460 cells in a 75 cm$^2$ cell culture flask (-7 × 10$^6$ cells) were washed twice with pHmed and then incubated with 160 μM EMSO (APExBIO Technology LLC, USA) in pHmed solution for 2 h at 37 °C. A saline-treated group was set up as a control. Cells were harvested by centrifugation (300 × $g$ for 5 min) and the supernatant was collected for extracellular pH (pHe) measurements. Gly-PFOBs or PFOBs were added and CEST signal intensity of the pHmed cell cultures was assessed using the Bruker 9.4 T MR imaging system. To confirm imaging results, pHe of redundant cell supernatant was quantified by a digital pH-meter, which is a microfiber optic pH transmitter equipped with a pH-sensitive chemical optical pH-1 microsensor (Presens Precision Sensing GmbH, Regensburg, Germany). The experiment was repeated with six replicates.

### Measurement of $O_2$ release

The oxygen concentration in aqueous solutions was measured using a portable dissolved oxygen meter (Rex, JPB-607A, China). For the measurement of the oxygen loading capacity of PFOB, 40 mL of deoxygenated water was added to a 45 mL flask that was closed with a rubber plug. The oxygen electrode probe was inserted through the rubber plug into the flask to measure the oxygen concentration of the solution in real-time. Then, 10 mL or 3 mL of the oxygen-saturated PFOB was injected via a syringe into the closed system. The oxygen concentration was recorded in real-time for 3 minutes.

### Establishment of experimental mouse models

For the murine xenograft model, 3 × 10$^6$ NCI- H460 lung cancer cells (2 × 10$^7$ cell ml$^{-1}$, 150 μL) were inoculated subcutaneously in the right

flank of 5-week-old female BALB/c nude mice. Liver ischemia was induced in 6-week-old male BALB/c mice by establishing a temporary occlusion of the pedicle of the left and middle lobes (approximately 70% of the total volume of the liver) as previously described[65]. For the liver metastasis model, NCI-H209 SCLC cells ($1\times 10^6$ cell ml$^{-1}$, 200 μL) were injected into the tail vein of 5-week-old female BALB/c BALB/c nude mice. $T_1$WI and $T_2$WI anatomical magnetic resonance imaging (MRI) was used to determine the successful establishment of the NCI-H209 SCLC liver metastasis model. The maximal tumor size/burden of 2000 mm$^3$ was permitted by the ethics committee and the maximal tumor size/burden in this study was not exceeded.

## In vivo MRI studies

Mice were anesthetized by inhalation with an initial dose of 2% isoflurane (RWD Life Science) and were maintained spontaneously breathing 1.5% isoflurane, supplied via a nose cone. During imaging, respiratory rates were continuously monitored by a small animal monitoring and gating system (SA Instruments), and the body temperature was maintained at 37 °C using a temperature-controlled heating system (Thermo).

For the antiacid treatment monitoring, $T_1$WI ($T_1$ weighted imaging) and CEST MRI were carried out when the tumor size reached about 150–350 mm$^3$. BALB/c nude mice bearing NCI-H460 lung cancer xenografts were administered the antiacid agent ESOM for 3 consecutive days, and CEST MRI was performed 4 h after intravenously injecting 300 μL Gly-PFOBs, PFOBs, or glycerol. For the T1-weighted MRI, we used a rapid acquisition with relaxation enhancement (RARE) sequence with the following parameters: TE = 15 ms, TR = 400 ms, averages = 8, repetitions = 1, echo spacing = 7.5 ms, RARE factor = 8, slice thickness = 1 mm, image size = 256 × 256, FOV = 38.4 × 38.4 mm. CEST MRI images were acquired using a single continuous-wave (CW) saturation pre-pulse (1 μT for 5 s), followed by a single-sliced RARE sequence with the following parameters: TE = 4.64 ms, TR = 6.0 s, averages = 1, repetitions = 33, echo spacing = 4.64 ms, RARE factor = 16, slice thickness = 1 mm, image size = 64 × 64, FOV = 38.4 × 38.4 mm, and a total of 33 frequency offsets distributed from −5 to +5 ppm relative to the water resonance ($n$ = 4 mice per group).

For in vivo, real-time and dynamic monitoring of the $O_2$ release and the acid-base fluctuations of tumor extracellular acidic microenvironment, after the pre-injection CEST sequence scanning, oxygen-saturated Gly-PFOBs ($O_2$) or PFOBs ($O_2$) were intratumorally injected at a 50 μL dose (623 mM based on glycerol), and subjected to CEST MRI and blood oxygen level-dependent (BOLD) imaging over time, wherein the variations of CEST signals were calculated as post-injections minus pre-injections at each time point. In addition, BOLD MRI noninvasively monitored tissue oxygenation by utilizing paramagnetic deoxyhemoglobin and measured tumor oxygenation quantitatively by calculating changes of $T_2^*$ values negatively related to tumor hypoxia. Quantitative $T_2^*$ MRI was performed using a multi-echo gradient-echo sequence, MSME sequence protocol (Bruker MSME-$T_2$-map), with the following parameters: TE = 3.5 ms, TR = 800 ms, averages = 2, repetitions = 1, echo images = 8, echo spacing = 5 ms, image size = 256 × 256, FOV = 38.4 × 38.4 mm. $N$ = 7 mice for Gly-PFOBs ($O_2$) group and $n$ = 6 mice for PFOBs($O_2$) group.

$T_1$WI, CEST MR (0.8 μT for 5 s), $^{19}$F-MRI, and chemical shift imaging (CSI) were performed 15 min after the successful establishment of ischemic liver mice models. Prior to this, 200 μL Gly-PFOBs (623 mM based on glycerol) was intravenously injected into each mouse. Non-ischemic liver mice models were set up as control. $^{19}$F-MRI images were acquired using $^1$H/$^{19}$F double-tune volume coil and applying $^{19}$F chemical shift-selective (CSSI) pulse sequences with a multi-slice rapid acquisition. Parameters of $^{19}$F_CSSI sequence were as follows: TE = 156 ms, averages = 128, repetitions = 1, RARE factor = 64, slice thickness = 3 mm, image size = 64 × 64, and FOV = 38.4 × 38.4 mm. For fusion with $^{19}$F images, additional $^1$H data sets with a slice thickness of

1.0 mm were recorded. The point-resolved spectroscopy method was used to obtain chemical shift images of lactate, and the proton signal was controlled by a variable pulse power and optimized relaxation delays method for which the parameters were, TE = 20 ms, TR = 2000 ms, averages = 2, repetitions = 1, flip angle excitation -90°, slice thickness = 3 mm, image size = 32 × 32, and FOV = 22 × 22 mm. N = 5 mice for ischemia models, and $n$ = 3 mice for Non-ischemic models as control.

For the NCI-H209 SCLC liver metastasis model, the $^1$H-CEST MRI and BOLD fMRI were carried out to obtain ORTW for precise RT when the individual liver metastasis reached about 8–15 mm$^3$ after the intravenously injection of 200 μL oxygen-saturated Gly-PFOBs ($O_2$) or Gly-PFOBs or PFOBs without oxygen saturation. The anatomical $T_1$WI and $T_2$WI MR imaging were conducted to monitor the RT efficacy in NCI-H209 SCLC liver metastases after the injection of 200 μL Gly-PFOBs ($O_2$) (623 mM based on glycerol). For the $T_2$-weighted MRI, we used a RARE sequence with the following parameters: TE = 64 ms, TR = 5000 ms, averages = 1, repetitions = 1, echo spacing = 8 ms, RARE factor = 16, slice thickness = 1 mm, image size = 256 × 256, FOV = 35 × 35 mm ($n$ = 6 mice per group).

## pH measurements

All pH values were measured by using a pH-sensitive optical pH-1 microsensor (Presens Precision Sensing GmbH, Regensburg, Germany) equipped with a PT 1000 temperature sensor, as we previously described[39]. The pH-1 microsensor (diameter of 140 μm optical fibers) was calibrated before measurements according to manufacture's instructions. The temperature sensor PT 1000 was calibrated to 37° and connected to the pH-1 microsensor for temperature-compensated measurements. All measurements were performed in an animal operation room under a constant temperature of 24° and sterile conditions.

## Glycerol concentration determination

For the rapid, sensitive, and accurate measurement of free glycerol concentration in the dialysate samples, we applied a free glycerol assay kit (Abcam, ab65337). In the assay, glycerol is enzymatically oxidized to generate a product that reacts with the probe to generate color ($\lambda$ = 570 nm). Specifically, Gly-CESTs (with a glycerol content of 3115 μmol in 5 ml) were subjected to dialysis against either 1000 ml of ultra-pure water or normal saline dialysate media for varying durations of 1 h, 2 h, 4 h, and 6 h (Spectra/Por, COMW 1000). The dialysate samples obtained at each time point were subsequently analyzed to determine the glycerol content (50 μL/well). Operations were carried out according to kit instructions. Briefly, set up plate for standard (50 μL) and samples (50 μL). Add 50 μL of Glycerol Reaction Mix to the standard and sample wells. Incubate plate at RT for 30 minutes protected from light. Measure plate at OD570 nm for colorimetric assay. To test the tumor glycerol concentrations, tumor tissues were collected and weighted at different time points (1 h, 2 h, 4 h, and 6 h) after NCI-H460 subcutaneous xenograft tumor models were intratumorally injected with Gly-PFOBs. In addition, BALB/c nude mice intravenously injected with Gly-PFOBs were sacrificed at different time points (1 h, 2 h, 4 h, and 6 h), and the liver tissues were obtained for glycerol test. The glycerol concentration in the tissue supernatant was determined by free glycerol assay kit (ab65337). To further verify the measurement results of free glycerol assay kit, Gas Chromatography/Mass Spectrometry (GC/MS) analysis was used to quantify glycerol concentration in Gly-PFOBs dialyzed against ultra-pure water or saline for different lengths of time (1 h, 2 h, 4 h, and 6 h). GC-MS analysis was performed on Agilent Technologies GC-MS system GC-7890A/MS-5977B model (Agilent Technologies, USA) equipped with a DB-WAX capillary column (30 m × 0.25 mm × 0.25 μm) for separating and identifying glycerol. Glycerol can exist in the body environment, involved in fat metabolism. Therefore, to further confirm our tested results, and to eliminate the influence of endogenous glycerol in the in vivo setting, we prepared $^{13}$C-Gly-CESTs using $^{13}$C isotopically labeled glycerol (Sigma-Aldrich)

and tested tissue glycerol content with carbon-13 nuclear magnetic resonance spectroscopy ($^{13}$C-NMR) and GC-MS. The applied injection approaches and animal models were same as above. For $^{13}$C-NMR measurements, the resulting supernatants were supplemented with D$_2$O (Sigma-Aldrich, Germany) and subsequently subjected to $^{13}$C-NMR (150 MHz for $^{13}$C) measurements by using a 5 mm QCI, with 5 μs delay and 200 scans. The $^{13}$C-Glycerol exhibits three characteristic peaks at 72 ppm and two characteristic peaks at 62.5 ppm.

### Immunofluorescence staining for cellular hypoxia evaluation

$2 \times 10^4$ NCI-H460 lung cancer cells were seeded in a small confocal dish for 24 h at 37 °C under 5% CO$_2$/1% O$_2$/94% N$_2$. Subsequently, the cells were treated with lissamine rhodamine B sulfonyl labeled oxygenated Gly-PFOBs (O$_2$) and Gly-PFOBs without oxygenation, and immediately the confocal dish was covered with a sealing membrane. After 1 h, the cells were fixed with 4% paraformaldehyde for 10 min and rinsed with PBS. Slides were then exposed to a SuperBlock (PBS) blocking buffer (Thermo Scientific) at room temperature for 30 min and incubated thereafter with the HIF-1α primary antibody (Abcam, H1alpha67, 1:100 dilution) in the SuperBlock blocking buffer at 1:200 dilution overnight at 4 °C. Cells were then washed three times with PBS and incubated in the SuperBlock blocking buffer with Alexa488 anti-mouse secondary antibody (1:500 dilution) for 1 h at room temperature. Cell nuclei were stained with DAPI for 5 min at room temperature. Images of cells were captured by a confocal laser scanning microscope (Nikon, Japan). Quantitative analysis of HIF-1α positive area (%) was automatically measured using the Image J software.

### Hypoxia evaluation in the tumor and liver

NCI-H460 tumor-bearing mice were randomly divided into 5 groups based on the time before and after oxygenated Gly-PFOBs (O$_2$) injections, which were pre, 30 min, 1 h, 2 h, and 3 h. The tumors of each group were surgically excised at 60 min after intravenous injection of pimonidazole hydrochloride at 60 mg/kg (Hypoxyprobe TM-1 plus kit, Hypoxyprobe Inc., Burlington, MA). For ischemic and non-ischemic liver mouse models, pimonidazole hydrochloride was injected at 60 mg/kg. The ischemia of the liver was induced in 6-week-old BALB/c male mice 15 min after the injection of pimonidazole hydrochloride, and tissues were surgically excised at 45 min after ischemia. Frozen tissue sections were interrogated with mouse monoclonal antibody FITC-Mab (dilution 1:200, Hypoxyprobe Inc.) according to the manufacturer's recommended procedure. Images of the tumor or liver immunofluorescence slides were acquired with a confocal laser scanning microscope (Nikon, Japan) Image intensity quantification of tumor hypoxia was performed with Image J software according to the formula: hypoxic-positive area (%) = visible hypoxic marker in tissue section/total area.

### Cell apoptosis assay

Cell apoptosis was performed with the cell apoptosis assay kit (Beyotime). Briefly, NCI-H460 cells ($5 \times 10^5$/per well) were plated in 6-well plates and cultured under hypoxic conditions (37 °C, 5% CO$_2$/1% O$_2$/94% N$_2$) for 24 h prior to RT. The cells in RT+Gly-PFOBs (O$_2$) 0 h, RT +Gly-PFOBs (O$_2$) 1-2 h, RT+Gly-PFOBs (O$_2$) 3 h and RT groups were subjected to RT at the dosage of 6 Gy. The control group was not subjected to any intervention. Cells were collected the next day, washed twice with cold PBS, and centrifuged at $300 \times g$ for 5 min at 4 °C. Subsequently, cells were resuspended in 195 μL of Annexin V -FITC binding buffer and incubated with 5 μL of Annexin V-FITC and 10 μL of PI staining solution at room temperature for 10–20 min. The stained cells were examined using a flow cytometer (Beckman Coulter, Fullerton, CA, United States). The results were analyzed with FlowJo 10 software (FlowJo, LLC). A figure exemplifying the gating strategy is provided in the Supplementary Information (Supplementary Fig. 12c). The MTT assay was used to measure the viability of hypoxic NCI-H460

cells after being subjected to RT at different doses (0 Gy, 2 Gy, 4 Gy, 6 Gy, and 8 Gy) with the addition of oxygenated Gly-PFOBs (O$_2$).

### Therapeutic effect evaluation

Tumor sections collected from different groups (Group I: PBS control; Group II: RT+ Gly-PFOBs (O$_2$) 0 h; Group III: RT+ Gly-PFOBs (O$_2$)1 h-2 h; Group IV: RT+ Gly-PFOBs (O$_2$) 3 h; Group V: RT alone) of mice were H&E-stained the day after a single radiation dose of 6 Gy delivered over the tumor sites using clinic radiotherapy system (VARIAN CLINAC 21EX, US). Ki67(Abcam, ab15580, 0.5 μg/mL) and CD31 antibodies (Abcam, ab28364, 1:50 dilution) and terminal deoxynucleotidyl transferase dUTP nick-end labeling assay (TUNEL, Beyotime Biotechnology, Shanghai, China) were used to verify the therapeutic effect and explore changes in tumor cell morphologies, proliferation, angiogenesis, and the cell apoptosis. After various treatments, the lengths and widths of the NCI-H460 subcutaneous tumors were measured by a digital caliper every two days. The weight of mice was also measured every two days. The tumor volume was calculated according to the following formula: volume= width$^2$× length/2. Relative tumor volume was calculated as $V/V_O$ ($V_O$ was the initial tumor volume). Tumors were collected from each group of mice and weighed on day 14 post treatments. For NCI-H209 SCLC liver metastasis, the metastasis foci were examined under a dissecting microscope, and the number of metastatic foci on each liver was counted.

### Biodistribution study

For the ESMO antiacid treatment experiment, 300 μL lissamine rhodamine B sulfonyl labeled Gly-PFOBs (623 mM based on glycerol) were intravenously injected into H460 tumor-bearing mice, which were sacrificed 4 h after the injection. Subsequently, tumors and main organs were collected and subjected to fluorescence measurements for lissamine rhodamine B sulfonyl by the IVIS Lumina III imaging system (Caliper Life Science, USA). The biodistribution of Gly-PFOBs was also examined in the NCI-H209 SCLC liver metastasis mouse model following the injection of 200 μL (623 mM) nanoprobes at assigned time points. The regions of interest (ROI) were quantified as average radiant efficiency [p/s/cm$^2$]/[μW/cm$^2$] (IVIS Living Image 4.3.1).

### Biological safety evaluation

The standard MTT cell cytotoxicity assay protocol was employed to assess in vitro cytotoxicity. NCI- H460 cancer cells were seeded in a 96-well plate at a density of $5 \times 10^3$ cells per well and cultured in 5% CO$_2$ at 37 °C for 24 h. The culture media were replaced with Gly-PFOBs at 3.375, 6.75, 13.5, 27, 54, and 108 mM. The cells were further incubated for 24 h and 48 h, following which the culture media were removed and the cells were washed with PBS twice. Then, 120 μL of MTT RPMI 1640 solution (0.8 mg/mL) was added to cells and incubated for 4 h. Finally, the MTT media were replaced with 150 μL dimethyl sulfoxide per well. Absorbance was monitored using a microplate reader (Bio-Tek, USA) at a wavelength of 490 nm, and cytotoxicity was expressed as the percentage of cell viability compared to control cells. The MTT assay was performed six independent measurements ($n = 6$). The body weight of animals was measured every other day for 33 consecutive days and histological assessment was conducted after the injection of 200 μL 623 mM Gly-PFOBs or saline used as the control. After 14 days, all animals were sacrificed and their heart, liver, spleen, kidney, and lung were removed, washed with saline, and fixed with 10% formaldehyde at 4 °C for at least 24 h through perfusion. Next, the organs were sectioned and embedded into paraffin blocks for H&E staining. The histological sections were examined under a microscope (OLYM-PUS, Japan) at ×20 magnification. Furthermore, the blood biochemical index was measured to evaluate the cumulative effect of Gly-PFOBs. Female BALB/c mice aged 4-6 weeks were selected for routine blood tests and biochemical assays, and control mice were injected with saline. After 3 days and 30 days of treatment, blood samples were

collected from the eye sockets of mice and analyzed with a hematology analyzer ($n$ = 3 mice per group).

## Statistical analysis

All studies were repeated at least three times or measured in triplicate. Each assay measurement were taken from distinct samples. Results were reported as means ± SD (SD = standard deviation). The student's $t$ test was used to compare quantitative data between two groups, and the statistical significance was calculated using a one-way analysis of variance (ANOVA). For the comparison among multiple groups, two-way ANOVA was used. Statistical differences between experimental groups were considered significant when the $P$ value was less than 0.05 ($P$ < 0.05). All statistical analyses were performed by GraphPad Prism 8.4.0 software (GraphPad Software, CA).

## Reporting summary

Further information on research design is available in the Nature Portfolio Reporting Summary linked to this article.

## Data availability

The authors declare that all data supporting the findings of this study are available within the Article, Supplementary Information, or Source Data file. Source data are provided with this paper.

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

## Acknowledgements

This work was supported by the National Natural Science Foundation of China (81627901, 82202226), the Natural Science Foundation of Heilongjiang Province (grant no. JQ2020H002), HMU Marshal Initiative Funding (HMUMIF-21003) and Heilongjiang Provincial Key Laboratory of Molecular Imaging foundation. X. Sun is the recipient of the above funds except 82202226. Z.H. is the recipient of the National Natural Science Foundation of China (82202226).

## Author contributions

R.A. wrote the manuscript draft and designed, performed, and analyzed all experiments. H.W. and C.N. assisted with establishing animal models and in vivo radiotherapy experiments. Z.H. contributed to the determination of glycerol concentration. R.A., H.W., and O.O.A. contributed to 1H-CEST MR imaging and data analysis. R.A. and H.W. participated in the preparation of Gly-PFOBs. R.A. and M.Z. performed H&E, immunohistochemical staining, and flow cytometry experiments. X.W. assisted with the pH measurement experiments. K.W. and S.L. assisted with the preparation of phantoms. R.A. and J.Z. contributed to the in vivo antiacid therapy. H.W., W.Q., and Xiaohong S. contributed to the cell culture. L.W. guided the probe preparation. Xilin S. designed, supervised, guided, and analyzed all experiments, and assisted in manuscript preparation. The final draft of the manuscript was approved by all co-authors.

## Competing interests

The authors declare no competing interests.
