## [Peer Review File · Nature Communications]

Glycerol-weighted chemical exchange saturation transfer
nano-probes allow $^{19}\text{F}/^1\text{H}$ dual-modality magnetic resonance
imaging-guided cancer radiotherapyREVIEWER COMMENTS

Reviewer #1 (Remarks to the Author): with expertise in MRI probes for imaging, cancer

Identifying the optimal radiotherapy time window is of great importance for improving cancer radiotherapy efficacy. In this manuscript, Rong and coworkers developed a Gly-CEST MRI method to identify the optimal radiotherapy time window, in which the pH and O₂ dual sensitive PFC-based Gly-CESTs nanoparticles play a central role. However, the manuscript suffers serious drawbacks, including language issues, inconsistency of data, unsupported conclusions, etc. One major issue is that glycerol is highly hydrophilic and would quickly diffuse from the nanoparticles of Gly-CESTs into the surrounding water. If the author can not prove that glycerol always stays at the surface of the nanoparticles, all the downstream study is baseless. Overall, the manuscript didn't meet the standard of Nature Communications. I recommend rejection.

Just list a few comments as follows:

1. The language is terribly awkward and hard to understand. There are numerous grammar issues and clues of copy&paste, such as "Radiotherapy (RT) is a primary modality and powerful cancer therapeutic tool." "This vicious cycle eventually leading to the exacerbation of tumor hypoxia, RT resistance" "The rhodamine B labelled perflubron based Gly-CESTs". Just name a few. It should be revised by a native English speaker.

2. The ways to describe the imaging modality are misleading. The authors used "19F-CEST dual-imaging". 19F-CEST means 19F signal-based CEST MRI, while the authors intended to say 19F MRI and 1H signal-based Gly-CEST MRI. The misleading description was repeatedly used in the manuscript, which must be corrected.

3. The way of indicating concentration is wrong. "The culture media were replaced with Gly-CESTs at 3.375, 6.75, 13.5, 27, 54, and 108 mM/L." mM already means mmol/L.

4. In the formulation, "and 0.1 mol% lissaminerhodamine B sulfonyl (16:0 LissRhod PE)" means no rhodamine B labeling? Lissaminerhodamine B sulfonyl is a reactive agent to label DPPE. How the rhodamine B was labeled onto the nanoparticles?

5. Page 12: "Also, when the 19F-CEST dual-imaging capability of Gly-CESTs was evaluated, Gly-CESTs provided stronger CEST signal at higher concentration, but the glycerol modification dose did not affect the 19F-MR signal intensity (Fig. 2f)." The signal intensity comparison should be quantified rather than visual observation. It seems that the 19F MRI signal intensity of 415 mM is much higher than the rest ones.

6. In the pH-dependent MTR measurement of 3f, the difference between pH 6.95 and pH 6.80 at 1.0 μ T is about 1-2%. However, in the cell culture pH-dependent MTR measurement, the difference between pH 6.95 and pH 6.80 at 1.0 μ T is about 10%. The significant difference indicated that factors rather than pH play a dominant role in the cell culture MTR changes. Therefore, the use of MTR changes to measure the cellular pH is not appropriate.

7. In the reference section, the references are not presented in Nature Communication style. Moreover, there are no page numbers in many references, including references 15, 18, 29.

8. In Supplementary Figure 8, it is obvious the Gly-CEST fluorescence intensity in the spleen is far higher than that of the lung, while the quantitative analysis showed the opposite result. Moreover, in Supplementary Figure 14, there is no obvious fluorescence signal in the spleen, while the quantitative analysis showed it is as high as in the heart and the kidney. There are obvious inconsistencies between the data. So the results are shaky. By the way, why there is an intense fluorescence signal in the spleen of Supplementary Figure 8 and no obvious fluorescence signal in the spleen of Supplementary Figure 14, in which the same nanoparticles were used?

Reviewer #2 (Remarks to the Author): with expertise in radiation oncology, MRI, nanotechnology

In this manuscript, a Gly-CEST based theranostic nanopaltform is developed for dual modality pH&Oxygen MR imaging, hence provide pivotal optimized radiotherapy time window (ORTW) information for boosted radiotherapy. Thanks to its good biocompatibility, this theranostic nanoplatform shows its translational potential in MRI guided radiotherapy. Nevertheless, some issues need to be addressed.

1. Please provide more details on radiation experiments, such as: what system used, what energy, is there image guidance, how many segments/fractions, in Material&Methods Section.
2. In this study, all the MR Images are collected on 9.4T small animal system, wherein the high magnetic field affords higher detection sensitivity than clinic diagnostic MRI (3T) and MRI-guided radiation machine (e.g. 0.35T ViewRay, 1.5T Elekta). Please add more discussion on how to overcome the low-sensitivity challenge on MRI-guided irradiator.
3. Is it possible to developed active targeting Gly-CEST nanoplatform for tumor-targeted theranostics?
4. The ORTW in Figs. 5 and 6 is relatively short ~1 hr, which seems too short for clinic manipulations. Please provide discussion on how to prolong this ORTW.

Reviewer #3 (Remarks to the Author): with expertise in radiotherapy, imaging

This manuscript, entitled "Optimized radiotherapy time window-facilitated 19F-CEST dual-modality MR imaging with pH and O₂ dual-sensitive Gly-CESTs for precision radiotherapy of lung cancer" by Rong A et al., demonstrates the feasibility of simultaneously measuring hypoxia status, including pH and O₂ levels, determining the best RT windows, and using a probe to carry O₂ to increase O₂ levels in the tumor microenvironment to relieve hypoxia status. The authors employed versatile MRI and biological techniques and conducted extensive work to analyze and verify the results, making the data reliable and the manuscript interesting to read. However, the study does not introduce groundbreaking methodology, as the strategy is based on the combination of existing and well-developed methods, from probe preparation to data acquisition. It represents a novel application of existing techniques to generate synergistic effects and multifunctional methods, but they are not conceptually new. Moreover, there is no compelling advantage demonstrated over the numerous similar technologies that have been published over the years.

Additionally, the manuscript requires substantial improvements, particularly in the Abstract and Introduction sections. The current descriptions in these sections are ambiguous and cumbersome, making it challenging to discern the study's purpose, methods, and findings. The writing in the Methods and Results sections is relatively lucid and aids in comprehending the study when read in reverse order.

Some other comments include:

(i) In the Introduction, it may be helpful to briefly explain how radiation therapy (RT) works and why it is dependent on oxygen levels in the tumor microenvironment. It would also be helpful to highlight how hypoxia can cause radiation resistance, making it imperative to investigate hypoxia in cancer treatment. Additionally, it would be also useful to clarify what the RT time window is and why it is critical in RT.

(ii) line 56-60: a long and unclear sentence, using terms like "the degree of hypoxia improvement" and "quantitative determination and visualization" without specifying which parameters are being measured, like O₂ level or pH? Line 59 change "include" to ", including"

(iii) In the Introduction, it may be helpful to include some comments on the design and synthesis of the probes, as well as clear chemical structures to help readers in understanding the study's experimental procedures and the mechanism how the probe work.

2. The authors should exercise caution when defining terms and making statements.

(i) The use of the technique name CEST to refer to a chemical probe, such as Gly-CESTs, can be confusing. While the probe is based on PFOB, it would be more appropriate to use a PFOB-related name to refer to the probe to avoid confusion.

(ii) The term "19F-CEST" used to describe the dual modality of fluorine and CEST MR in the manuscript may be misleading. Using the term "19F&CEST" instead would be more appropriate and accurate.

(iii) Line 45, "prognosis" could be replaced by "outcome", since RT is not the tool for prognosis.

(iv) Clarifying that perfluorinated compounds are prepared into emulsions would be helpful in understanding the description of "nano-size PFCs" (line 67) and similar to nanomaterials throughout the manuscript.

(v) line 69, "exchangeable molecules" is unclear, since talking about CEST, "exchangeable protons" will be clear enough.

(vi) Line 75-80, not sure the statement is accurate. Is this MOST advanced MRI guided the radiotherapy system is a milestone?

A review paper published in 2021 in Radiology, MRI-guided Radiation Therapy: An Emerging Paradigm in Adaptive Radiation Oncology including many MRI guided RT studies.

(vii) Line 382. The exchange rate depends on temperature not pH, higher pH suggests low free H⁺, which impact the CEST.

(viii) Line 162 and 205, "Anti-acid treatment" may not be a correct term, or should be "antacid"

(ix) Line 624-625: is this a true and accurate statement?

(x) Line 643: is this an accurate statement, metal ion itself could be toxic, but once coordinating with inert ligands (very slow exchange), the complex can be very stable and non-toxic. There are many medical applications of using metal complexes as contrast agents.

(xi) Line 647 – 650: PFCs could be low toxic, but some PFCs disrupt normal endocrine activity; reduce immune function; cause adverse effects on multiple organs, including the liver and pancreas.

3. Methods

Line 100. Description of Preparation of Gly-CESTs and PFOBs are unclear and confusing. It would be better to have a clear description of how many probes are prepared, what they are, give the proper name, and how to prepare each of them.

(i) Did not mention how to make Gly-CESTs,

(ii) Line 101: Did not mention the purpose of using fluorescence previously, while making rhodamine B labelled perflubron-Gly-CESTs here.

(iii) line 102, Gly-CESTs consisting of 20% PFOB and 2% of a surfactant, then rest of what?

(iv) Line 111. "The synthesis of", it seems no reaction going on, it may just say the "preparation"...

4. Results

Line 348: no "synthesis" is described, only "characterization" in this section.

5. Figures: Figure 3 is small and difficult to read, panel a is pixelated.

The figures are cluttered and many panels could be moved to the supplemental to highlight the more important findings.

The figure legends do not clearly describe what is being done in the experiments and requires going back and forth between the text and figure to understand the experiment.

6. Some other minor issues:

Line 50: change "leading to " to "lead"

Line 75-75: "with the mostwas introduced..." wrong grammar, change to "with the introduction of ...,"

Line 145, 220 and others: wrong unit "mM/L" expression was used, it should be mM or mmol/L.

Line 163: insert reference properly, instead of using PMID number.

Line 225: no need to write "(T2-star)"

Line 229: "19F (19F) MRI" is strange, 19F MRI will be good.

Line 272: It is better to use the term "intravenous injection" instead of "i.v. injection". Line 474 suddenly showed up "the intratumoral injection" The manuscript should specify the type of injection used at the beginning and consistently throughout the text to avoid confusion. It should also clarify what kind of injection was used for the biocompatibility study (line 593) and whether it was injected into healthy or tumor mice.

Line 297: "from different groups", although it mentioned in PDX mouse preparation, it would be better to say it again here what the groups are.

Line 375: confusing sentence to read through.

Line 618: change "resistant" to "resistance"

Line 694: It would be better to point out in caption the numbers on the top of Fig 2d are glycerol concentration.

Line 697 Fig. 3.: There are spaces between labels like "Gly-C EST s" and "PF OBs"

Line 715 Fig. 4.: Boxes outlined texts in c and d. in h legend should be saline not PBS.

Line 720. Fig 4 e.: would indicate pH value measured by microsensor.

Reviewer #4 (Remarks to the Author): with expertise in imaging, probes

First, I was very interested in the paper and the extensive high level studies employed. As such, I made suggested editorial suggests for the abstract and introduction that improve the grammar and understanding at least for me.

I am familiar with all aspects of this paper and I have no significant comments regarding the studies, but I have one fundamental questions that could be clarified.

1) The glycerol is added to the water and PFC. The effects are glycerol level-dependent up to a maximal effect plateau. Fine, but glycerol is not chemically coupled to the particle but only in the particle environment. In vitro this environment is constrained.

Glycerol is not miscible in PFC, virtually nothing is. How is it associated with the particles themselves? In vivo, how does the glycerol remain with the particles and not diffuse into tissues, and behave like the control particles? Was phosphatidyl glyceride (PG) considered? The particles are surprisingly anionic, -60mV, and this is very negative for a DPPC/cholesterol/PE surfactant which should be -20 to -40 mV. Adding PG to this surfactant may make the particle Zeta potential more negative, which could trigger acute complement activation. So there are issues to work through to make the compelling results jive with these unaddressed points.

REVIEWER COMMENTS

Reviewer #1 (Remarks to the Author): with expertise in MRI probes for imaging. Cancer

Identifying the optimal radiotherapy time window is of great importance for improving cancer radiotherapy efficacy. In this manuscript, Rong and coworkers developed a Gly-CEST MRI method to identify the optimal radiotherapy time window, in which the pH and O₂ dual sensitive PFC-based Gly-CESTs nanoparticles play a central role. However, the manuscript suffers serious drawbacks, including language issues, inconsistency of data, unsupported conclusions, etc.

One major issue is that glycerol is highly hydrophilic and would quickly diffuse from the nanoparticles of Gly-CESTs into the surrounding water. If the author can not prove that glycerol always stays at the surface of the nanoparticles, all the downstream study is baseless. Overall, the manuscript didn't meet the standard of Nature Communications. I recommend rejection.

We express our sincere gratitude for the reviewer's profound and valuable insights, which have greatly contributed to the enhancement of our manuscript. We wholeheartedly concur with the reviewer's assertion that the investigation of the permanence of glycerol on the surface of nanoparticles holds significant merit. In response to this astute observation, we have diligently incorporated a comprehensive set of *in vitro* and *in vivo* experiments to elucidate this phenomenon, as outlined below: It is widely recognized that glycerol is classified as a hydrophilic substance due to its possession of hydroxyl groups, which could form bonds with water molecules. Consequently, the inclusion of glycerol in composite membranes has the potential to result in a slight increase in water content [*Molecules*, 2023-01-02;28(1)]. Glycerol possesses both hydrophilic and hygroscopic characteristics, rendering it a commonly employed cosmetic component functioning as a humectant and hygroscopic agent [*Bioresource technology*, 2009-12-01;100(24):6362-8].

To address the reviewer's concern regarding the potential rapid diffusion of the glycerol component from the nanoprobe, we initially performed *in vitro* dialysis experiments.

Free Glycerol Assay Kit (ab65337) was firstly used to test the glycerol content in the dialysate. In the assay, glycerol is enzymatically oxidized to generate a product which reacts with the probe to generate color ($\lambda = 570$ nm) or fluorescence (Ex/Em = 535/587 nm). This assay can detect 50 pmol ~10 nmol of glycerol sensitively and accurately. Specifically, Gly-CESTs (with a glycerol content of 3115 μmol in 5 ml) were subjected to dialysis against either 1000 ml of ultra-pure water or normal saline dialysate media for varying durations of 1 h, 2 h, 4 h, and 6 h (Spectra/Por, COMW 1000). The dialysate samples obtained at each time point were subsequently analyzed to determine the glycerol content (50 μl /well).

The results showed that the concentration of glycerol in the dialysate slightly increased over time. At the 6 h of dialysis, the glycerol concentration measured in ultra-pure water and normal saline were found to be 32.3 ± 0.39 μmol , and 41.43 ± 3.25 μmol , respectively. We added a new table with the mean and standard deviation to Supplementary Information of the revised manuscript (Supplementary Table 1). In both ultra-pure water and normal saline environments, the release of glycerol from the Gly-CESTs into the external environment after 6 hours of dialysis was found to be only 1.04% (32.3 μmol / 3115 μmol) and 1.33% (41.43 μmol / 3115 μmol), respectively. These results were also added to the Supplementary Information of the revised manuscript (Supplementary Fig. 3a).

To further verify the above results, the Gas Chromatography-Mass Spectrometry (GC-MS) analysis was performed to quantify glycerol concentration in Gly-CESTs dialyzed against ultra-pure water or saline for different lengths of time. Corresponding statistical results showed that the variation occurring in the reducing glycerol concentration appeared to be minor. We added a new table with the mean and standard deviation to Supplementary Information of the revised manuscript (Supplementary Table 2). Compared with Gly-CESTs without dialysis, the glycerol concentration in Gly-CESTs was only slightly decreased after dialyzed against ultra-pure water or saline (Supplementary Fig. 3b). These results were consistent with the results of the Free Glycerol Assay Kit.

A new table added in the revised Supplementary Information (Page 2)

Gly-CESTs dialyzed against	Ultra-pure water(μmol)	Gly-CESTs dialyzed against saline (μmol)	Saline (μmol)
---	---	--

	ultra-pure water (μmol)			
1 h	3099 \pm 0.59	16.04 \pm 0.59	3098 \pm 0.84	16.74 \pm 0.84
2 h	3094 \pm 0.38	21.43 \pm 0.38	3086 \pm 0.87	29.14 \pm 0.87
4 h	3090 \pm 0.86	25.23 \pm 0.86	3078 \pm 1.8	37.16 \pm 1.80
6 h	3083 \pm 0.39	32.30 \pm 0.39	3074 \pm 3.3	41.43 \pm 3.25

Supplementary Table 1. The glycerol content in Gly-CESTs and in different dialysates (ultra-pure water or saline) at different time points were determined by Free Glycerol Assay Kit (abcam).

A new table added in the revised Supplementary Information (Page 2)

	Gly-CESTs without dialysis	Gly-CESTs dialyzed against Ultra-pure water (mmol/L)	Gly-CESTs dialyzed against Saline (mmol/L)
1 h	623.51 \pm 39.42	622.93 \pm 12.18	572.44 \pm 33.97
2 h		615.39 \pm 4.87	571.62 \pm 12.72
4 h		614.64 \pm 18.42	562.21 \pm 20.13
6 h		612.96 \pm 21.5	561.51 \pm 15.83

Supplementary Table 2. The glycerol concentration in Gly-CESTs without dialysis and glycerol concentration in Gly-CESTs dialyzed against different dialysates (ultra-pure water or saline) at different time points were determined by Gas Chromatography-Mass Spectrometry (GC-MS) analysis.

A new figure added in the revised Supplementary Information (Page 3)

Supplementary Fig.3. Evaluation of the glycerol-retaining ability of Gly-CESTs.

a. The glycerol content in Gly-CESTs and different dialysate (ultra-pure water or saline) at different time points was determined by Free Glycerol Assay Kit (abcam);
 b. Gas Chromatography-Mass Spectrometry (GC-MS) analysis was performed to

determine glycerol concentrations in Gly-CESTs after different duration of dialysis (1 h, 2 h, 4 h, 6 h). * $P < 0.05$, ** $P < 0.01$, *** $P < 0.001$, **** $P < 0.0001$. Data are presented as mean \pm standard deviation (SD) (n = 3).

Description added in the revised manuscript (Results section, line 412-422)

To evaluate the glycerol-retaining ability of Gly-CESTs, we applied free glycerol assay kit for glycerol content test. The results showed that the content of glycerol in the dialysate slightly increased over time. At the 6 h of dialysis, only 1.04% (32.3 μmol / 3115 μmol) and 1.33% (41.43 μmol / 3115 μmol) of glycerol from Gly-CESTs free into the external environment (Supplementary Table 1, Supplementary Fig.3a). To further verify the above measured results, the Gas Chromatography-Mass Spectrometry (GC-MS) was used to quantify glycerol concentration in Gly-CESTs dialyzed against ultra-pure water or saline for different lengths of time. Compared with Gly-CESTs without dialysis, the glycerol concentration in Gly-CESTs was only slightly decreased after dialyzed against ultra-pure water or saline (Supplementary Table 2, Supplementary Fig. 3b). These results were consistent with the results of the Free Glycerol Assay Kit.

However, the environment *in vivo* is significantly different from *in vitro*. Therefore, to further explore the glycerol retaining ability of Gly-CESTs, we performed relevant *in vivo* experiments. Glycerol Assay Kit (ab65337) was also applied to determine tissue glycerol concentration. Firstly, considering the presence of endogenous glycerol within living organisms, we tested the endogenous glycerol concentration in both tumor and liver tissues, which were found to be 677.23 ± 21.94 nmol/g and 2429.15 ± 214.17 nmol/g, respectively. In addition, since both intratumoral injection and intravenous injection were applied in our study for different research aim, we explored the changes of glycerol content under two injection approaches and compared the results with those of exogenous injection of free glycerol. NCI-H460 subcutaneous xenograft tumor tissues were collected at different time points (1 h, 2 h, 4 h, and 6 h) after intratumoral injection of Gly-CESTs (623mM based on glycerol, 50 μl) or exogenous free glycerol (623mM based on glycerol, 50 μl), and the glycerol concentration was determined. Balb/c nude mice were sacrificed after intravenously injected with Gly-CESTs (623mM based on glycerol, 200 μl) or exogenous free glycerol (623mM based on glycerol, 200 μl) at different time points (1 h, 2 h, 4 h, and 6 h) and the liver tissue were

obtained for the determination of glycerol concentration. Corresponding results were added to the Supplementary Information of the revised manuscript (Supplementary Fig. 11). Following the exogenous introduction of Gly-CESTs or free glycerol, the findings indicate that the exogenous free glycerol, being a hydrophilic substance, exhibits a tendency to rapidly clear from tissues irrespective of intratumoral or intravenous administration. Furthermore, it maintains a relatively low concentration comparable to the endogenous glycerol level at all subsequent time intervals (2 h, 4 h, and 6 h, except 1 h). Compared with free glycerol injection group, the clearance of glycerol happens more gradually in Gly-CESTs intratumoral injection group. After 1 h of intratumoral injection of Gly-CESTs, although the glycerol concentration decreased 17.67%, the concentration was still reached 18251 ± 475 nmol/g in NCI-H460 subcutaneous lung tumor tissue. This concentration was found to be 6.22 times higher than that observed in the free glycerol injection group (2932 ± 162 nmol/g). Similar findings were observed in liver tissue after intravenous injection of Gly-CESTs (Supplementary Fig. 11a and b). These experimental results illustrated the glycerol retaining ability of Gly-CESTs.

A new figure added in the revised Supplementary Information (Page 7)

Supplementary Fig. 11 a-b. The determination of *in vivo* glycerol-retaining ability of Gly-CESTs in subcutaneous tumor (a) or liver tissue (b) . a. After intratumoral injection of Gly-CESTs or the same amount of free glycerol (623mM based on glycerol, 50μl), the glycerol concentration in tumor tissues at different time points was measured by free glycerol assay kit. Meanwhile, the concentration of endogenous glycerol in tumor was also quantified for accurately reflect the exogenous glycerol concentration. b. The glycerol concentration in liver tissue was determined by Free Glycerol Assay Kit (abcam) at different time points after intravenous injection of Gly-CESTs or free glycerol containing same amount of glycerol (623mM based on glycerol, 200 μl). The concentration of endogenous

glycerol in liver was also quantified. ns, no significance, * $P < 0.05$, ** $P < 0.01$, **** $P < 0.0001$. Data are presented as mean \pm standard deviation (SD) (n = 3)

Description added on in the revised manuscript (Results section, line 535-554)

To further explore the *in vivo* glycerol retaining ability of Gly-CESTs, NCI-H460 subcutaneous xenograft tumor tissues or BALB/c nude mice liver tissues were collected before and after intratumoral or intravenous injection of Gly-CESTs at different time points (1 h, 2 h, 4 h, and 6 h), respectively. The tissue glycerol concentration was also determined by Free Glycerol Assay Kit. We first determined the endogenous glycerol level existed in tumor and liver tissue, which were found to be 677.23 ± 21.94 nmol/g and 2429.15 ± 214.17 nmol/g, respectively. Following the exogenous introduction of Gly-CESTs or free glycerol, the findings indicate that the exogenous free glycerol, being a hydrophilic substance, exhibits a tendency to rapidly clear from tissues irrespective of intratumoral or intravenous administration. Furthermore, it maintains a relatively low concentration comparable to the endogenous glycerol level at all subsequent time intervals (2 h, 4 h, and 6 h, except 1 h). Compared with free glycerol injection group, the clearance of glycerol happens more gradually in Gly-CESTs intratumoral injection group. After 1 h of intratumoral injection of Gly-CESTs, although the glycerol concentration decreased 17.67%, the concentration was still reached 18251 ± 475 nmol/g in NCI-H460 subcutaneous lung tumor tissue. This concentration was found to be 6.22 times higher than that observed in the free glycerol injection group (2932 ± 162 nmol/g). Similar findings were observed in liver tissue after intravenous injection of Gly-CESTs (Supplementary Fig. 11a and b). These experimental results illustrated the glycerol retaining ability of Gly-CESTs.

Glycerol can also be existed in the body environment as substantiated by results of our experiments. Therefore, to further confirm our tested results, and to eliminate the influence of endogenous glycerol in the *in vivo* setting, we prepared ^{13}C -Gly-CESTs using ^{13}C isotopically labeled glycerol and tested tissue ^{13}C -glycerol content with carbon-13 nuclear magnetic resonance spectroscopy (^{13}C -NMR) and GC-MS (Supplementary Fig. 11c-f). The injection method and animal model were same as above. The concentration of ^{13}C -glycerol in NCI-H460 subcutaneous tumor tissues was 1360.9 ± 33.8 mg/L after 1 hour intratumoral injection of ^{13}C -Gly-CESTs, which was

7 times higher than that in free ^{13}C -glycerol injection group ($194.2 \pm 10.94 \text{ mg/L}$). Subsequently, with the extension of time, there was a very slow decline of ^{13}C -glycerol concentration in ^{13}C -Gly-CESTs injection group. Moreover, even after 6 hours of injection, the ^{13}C -glycerol concentration in Gly-CESTs injection group was still higher than that in the free ^{13}C -glycerol group ($555.79 \pm 14.87 \text{ mg/L}$ vs. undetectable, 6 h). Similar observation was found in liver tissue after intravenous injection of ^{13}C -Gly-CESTs or free ^{13}C -glycerol. Even though the hepatic clearance of ^{13}C -glycerol was observed to be more rapid following intravenous administration of ^{13}C -Gly-CESTs compared to subcutaneous tumors, the rate of ^{13}C -glycerol clearance in the ^{13}C -Gly-CESTs injection group remained slower than that in the intravenous free ^{13}C -glycerol injection group (Supplementary Fig. 11c-f). Taken together, above results have confirmed the glycerol-retaining ability of Gly-CESTs and have suggested that glycerol is not that rapidly diffused and cleared from the Gly-CESTs.

A new figure added in the revised Supplementary Information (Page 7)

A new figure added in the revised Supplementary Information (Page 7)

Supplementary Fig. 11c-f. The determination of *in vivo* ^{13}C - glycerol retaining ability of ^{13}C -Gly-CESTs in subcutaneous tumor or liver tissue. c and e. ^{13}C -NMR (c), GC-MS(e) analysis was performed to determine the metabolic tendency or concentration of ^{13}C -glycerol in NCI-H460 tumor tissue at different time points after intratumoral injection of ^{13}C -Gly-CESTs or free ^{13}C -glycerol containing same

amount of ^{13}C -glycerol (623mM based on ^{13}C -glycerol, 50 μl); d and f. ^{13}C -NMR (d), GC-MS (f) analysis was used to determine the metabolic tendency or concentration of ^{13}C -glycerol in liver tissue at different time points after intratumoral injection of ^{13}C -Gly-CESTs or free ^{13}C -glycerol containing same amount of ^{13}C -glycerol (623mM based on ^{13}C -glycerol, 200 μl). ns, no significance, * $P < 0.05$, ** $P < 0.01$, *** $P < 0.001$, **** $P < 0.0001$. Data are presented as mean \pm standard deviation (SD) (n = 3)

Description added in the revised manuscript (Results section, line 555-576)

Glycerol can be existed in the body environment, involved in fat metabolism. The content of endogenous glycerol was also substantiated by results of our experiments. Therefore, to further confirm our tested results, and to eliminate the influence of endogenous glycerol in the *in vivo* setting, we prepared ^{13}C -Gly-CESTs using ^{13}C isotopically labeled glycerol and tested tissue glycerol content with carbon-13 nuclear magnetic resonance spectroscopy (^{13}C -NMR) and GC-MS. The injection method and animal model were same as above. The concentration of ^{13}C -glycerol in NCI-H460 subcutaneous tumor tissues was 1360.9 ± 33.8 mg/L after 1 hour intratumoral injection of ^{13}C -Gly-CESTs, which was 7 times higher than that in free ^{13}C -glycerol injection group (194.2 ± 10.94 mg/L). Subsequently, with the extension of time, there was a very slow decline of ^{13}C -glycerol concentration in ^{13}C -Gly-CESTs injection group. Moreover, even after 6 hours of injection, the ^{13}C -glycerol concentration in Gly-CESTs injection group was still higher than that in the free ^{13}C -glycerol group (555.79 ± 14.87 mg/L vs. undetectable, 6 h). Similar observation was found in liver tissue after intravenous injection of ^{13}C -Gly-CESTs or free ^{13}C -glycerol. Even though the hepatic clearance of ^{13}C -glycerol was observed to be more rapid following intravenous administration of ^{13}C -Gly-CESTs compared to subcutaneous tumors, the rate of ^{13}C -glycerol clearance in the ^{13}C -Gly-CESTs injection group remained slower than that in the intravenous free ^{13}C -glycerol injection group (Supplementary Fig. 11c-f). Taken together, above results have confirmed the glycerol-retaining ability of Gly-CESTs and have suggested that glycerol is not that rapidly diffused and cleared from the Gly-CESTs.

Previous studies reported that glycerol and water interact with lipid head groups in a similar fashion, and to partition equally between the bulk and the lipid membrane surface and enhances and strengthens the hydrogen bond network of lipid membrane surface and bulk water similarly. In addition, Glycerol by altering the hydrogen bond structure and intermolecular cohesion of the global solvent, as manifested by increased solvent viscosity [*The Journal of Chemical Physics*, 2016-07-28;145(4):041101]. Moreover, studies showed that the application of hygroscopic glycerol in the solvent of double-network hydrogel could enhance the water retaining ability of hydrogels, generating the water-Gly binary hydrogel with boosted sensitivity to NO₂ and significantly enhanced stability [*ACS Applied Materials and Interfaces*. 2019-01-16;11(2):2364-2373]. Most importantly, in this study we applied dipalmitoylphosphatidylcholine (DPPC) in the Gly-CESTs preparation, constituting approximately 88.9% of the molar ratio of surfactants (lipids). Harvey et al. examined the effects of bulk glycerol (0-30% w/w) on dipalmitoylphosphatidylcholine (DPPC) monolayers structures and dynamics using complementary biophysical measurements and molecular dynamics (MD) simulations, in which DPPC monolayers and liposomes were used as model pulmonary interfaces. Glycerol was found to preferentially interact with the carbonyl groups in the interfacial region of DPPC and with phosphate and choline in the headgroup, thus causing an increase in the size of the headgroup solvation shell, as evidenced by an expansion of DPPC monolayers (molecular area increased from 52 to 68 Å²) and bilayers seen in both Langmuir isotherms and MD simulations. They illustrated that both small angle neutron scattering, and MD simulations indicated a reduction in gel phase DPPC bilayer thickness by ~3 Å in 30% w/w glycerol, a phenomenon consistent with the observation from FTIR data, that glycerol caused the lipid headgroup to remain oriented parallel to the membrane plane in contrast to its more perpendicular conformation adopted in pure water. Furthermore, in their study, the FTIR measurements suggested that the terminal methyl groups of the DPPC acyl chains were constrained in the presence of glycerol. This observation was supported by MD simulations, which predict bridging between adjacent DPPC headgroups by glycerol as a possible source of its putative membrane stiffening effect. Collectively, these data indicate that glycerol preferentially solvates DPPC headgroups and localizes in specific areas of the interfacial region, resulting in structural changes to DPPC bilayers [*Langmuir*. 2018-06-12;34(23):6941-6954].

In summary, the PFC nanoparticles were composed of a hydrophobic PFC core and a surrounding lipid with a hydrophobic inner layer and hydrophilic outer layers. The preferential interaction of glycerol with the carbonyl groups in the interfacial region of DPPC, as well as with phosphate and choline in the headgroup, along with other established mechanisms, indicates that glycerol locates and retains at the surface of DPPC-lipid nanoprobe.

Just list a few comments as follows:

1. The language is terribly awkward and hard to understand. There are numerous grammar issues and clues of copy&paste, such as “Radiotherapy (RT) is a primary modality and powerful cancer therapeutic tool.” “This vicious cycle eventually leading to the exacerbation of tumor hypoxia, RT resistance” “The rhodamine B labelled perflubron based Gly-CESTs”. Just name a few. It should be revised by a native English speaker.

We express our gratitude for the valuable suggestions provided by the reviewer. We sincerely apologize for the presence of grammatical errors in our initial manuscript. The manuscript has been revised by a proficient native English language editor specializing in scientific papers to enhance the accuracy and clarity of our manuscript. The manuscript was totally authored by us. To avoid similar presentation, significant modifications have been implemented in the revised manuscript, particularly in the sections where references to prior research are cited. The reviewer's reminder holds significant importance, and we extend our gratitude to the reviewer for their diligent efforts in enhancing our manuscript. Concurrently, we have performed a thorough duplicate check on the manuscript, and the outcome meets the journal's stipulated criteria.

2. The ways to describe the imaging modality are misleading. The authors used “¹⁹F-CEST dual-imaging”. ¹⁹F-CEST means ¹⁹F signal-based CEST MRI, while the authors intended to say ¹⁹F MRI and ¹H signal-based Gly-CEST MRI. The misleading description was repeatedly used in the manuscript, which must be corrected.

We appreciate the helpful comments from the reviewer. The term "¹⁹F-CEST MR imaging" used to describe the dual modality of fluorine and hydrogen proton based CEST MR in the manuscript may be misleading. We used the term "¹⁹F/¹H-CEST MR imaging" instead in our revised manuscript, which would be more appropriate and accurate.

It also reminds us that the naming of Gly-CESTs will cause misunderstandings among readers since we inappropriately used the technique name CEST to refer to a chemical probe. Therefore, we also changed "Gly-CESTs" to "Gly-PFOBs" in the subsequent response letter and revised manuscript, as well as revised supplementary information.

3. The way of indicating concentration is wrong. "The culture media were replaced with Gly-CESTs at 3.375, 6.75, 13.5, 27, 54, and 108 mM/L." mM already means mmol/L.

We express our gratitude to the reviewer for the meticulous examination. We have diligently reviewed and rectified all instances of spelling errors present within the manuscript.

4. In the formulation, "and 0.1 mol% lissaminerhodamine B sulfonyl (16:0 LissRhod PE)" means no rhodamine B labeling? Lissaminerhodamine B sulfonyl is a reactive agent to lable DPPE. How the rhodamine B was labeled onto the nanoparticles?

We express our gratitude to the reviewer for bringing this matter to our attention. We sincerely apologize for the lack of clarity in our statement, which resulted in confusion. In the process of preparing Gly-PFOBs (formerly named Gly-CESTs), we used 0.1 mol% lissamine-rhodamine B sulfonyl (16:0 LissRhod) instead of Rhodamine B. These descriptions were added in the Methods section (Page 4, line 114-121) of the revised manuscript and highlighted in yellow. Additionally, we have thoroughly reviewed the entire manuscript to ensure accurate description of lissamine-rhodamine B sulfonyl.

5. Page 12: "Also, when the ¹⁹F-CEST dual-imaging capability of Gly-CESTs was evaluated, Gly-CESTs provided stronger CEST signal at higher concentration,

but the glycerol modification dose did not affect the ^{19}F -MR signal intensity (Fig. 2f).” The signal intensity comparison should be quantified rather than visual observation. It seems that the ^{19}F MRI signal intensity of 415 mM is much higher than the rest ones.

We express our gratitude for the reviewer's valuable advice. In accordance with the reviewer's suggestion, the ^{19}F -MRI signal-to-noise ratio (SNR) was calculated and quantified in each phantom studies, and the statistical results illustrated that the SNR of Gly-PFOBs (formerly named Gly-CESTs) with different glycerol concentrations as well as the SNR of Gly-PFOBs (formerly named Gly-CESTs) in different pH solutions showed no significant difference among all groups. This section has been updated in the manuscript and added to the Supplementary Data (Supplementary Fig.6).

A new figure added in the revised Supplementary Information (Page 4)

Supplementary Fig. 6. Statistical results of ^{19}F -MRI signal-to-noise ratio (SNR) in the phantom studies. a. ^{19}F -MRI SNR statistical results of Gly-PFOBs with different glycerol concentrations b. ^{19}F -MRI SNR statistical results of Gly-PFOBs in different pH solutions. ns, no significance. Data are presented as mean \pm standard deviation (SD) (n = 3).

Description added on Page 14 of the revised manuscript (Results section, line 440-443, line 453-456)

Also, when the $^{19}\text{F}/^1\text{H}$ -CEST dual-imaging capability of Gly-PFOBs was evaluated, Gly-PFOBs provided stronger CEST signal at higher concentration. Expectedly, the glycerol modification dose did not affect the ^{19}F -MR signal intensity (Fig. 2f and Supplementary figure 6a).

In addition, different pH values of Gly-PFOBs solutions did not interfere with the ^{19}F -MR signals due to the hydrogen proton exchange between glycerol and surrounding physiological water with no effect on the relaxation properties of fluorine atoms (Fig. 3c and Supplementary figure 6b).

6. In the pH-dependent MTR measurement of 3f, the difference between pH 6.95 and pH 6.80 at 1.0 μT is about 1-2%. However, in the cell culture pH-dependent MTR measurement, the difference between pH 6.95 and pH 6.80 at 1.0 μT is about 10%. The significant difference indicated that factors rather than pH play a dominant role in the cell culture MTR changes. Therefore, the use of MTR changes to measure the cellular pH is not appropriate.

We appreciate the reviewer for raising this point. The reason why reviewer has this confusion is that the principle of ^1H -CEST MR reflect pH changes is quite different from that of pH meter. We sincerely apologize for the lack of clarity in our text. The subsequent statements have been included in the revised manuscript's discussion section to enhance readers' comprehension (Page 23, line 738-750).

The digital pH-meter, a microfiber optic pH transmitter equipped with a pH-sensitive chemical optical pH-1 microsensor (Presens Precision Sensing GmbH, Regensburg, Germany), creates very stable, internally referenced measured values, and directly reflects the specific pH value of solution or tissue substance. It is considered the gold standard for pH measurement.

However, for ^1H -CEST MR imaging, there is a "signal amplification process". The specific imaging principle of CEST imaging is as follows:

Chemical exchange saturation transfer (CEST) imaging is a specific type of magnetic resonance imaging (MRI) technique, produced by exchangeable solute protons that resonate at a frequency different from the bulk water protons are selectively saturated using RF irradiation [*Chemical Society Reviews*, 2006-06-01;35(6):500-111] [*Nature Medicine*, 2013-08-01;19(8):1067-72] [*Science Translational Medicine*, 2015-10-14;7(309):309ra161] [*Accounts of Chemical Research*, 2009-07-21;42(7):915-24] [*Journal of American Chemical Society*, 2016-09-07;138(35):11136-9]. This saturation is subsequently transferred to bulk water when solute protons exchange with water protons and the water signal becomes slightly attenuated. In view of the low

concentration of solute protons (μM to mM range), a single transfer of saturation would be insufficient to show any discernable effect on water protons. However, because the water pool is much larger than the saturated solute proton pool, each exchanging saturated solute proton is replaced by a nonsaturated water proton, which is then again saturated. The exchange with water functions as a sensitivity enhancer because the nonsaturated protons of the large solvent pool continuously replace the saturated solute protons, which are again saturated and undergo fast exchange in a process that benefits from fast exchange with water. This repeated exchange, leading to saturation of the water signal and enhancement of the effect. Corresponding CEST papers showed that enhancements of the order of $10 \sim 10\,000$ can be achieved by this process, giving rise to possibilities for imaging low concentration metabolites and contrast materials based on the water protons [*Journal of American Chemical Society*,2001-09-05;123(35):8628-9] [*Journal of Magnetic resonance*, 2000-03-01;143(1):79-87]. The prolonged irradiation leads to substantial enhancement of this saturation effect, which eventually becomes visible on the water signal, allowing the presence of low-concentration solutes to be imaged indirectly. Therefore, CEST constitutes a powerful sensitivity enhancement mechanism in which the signal of low concentration solutes can be amplified and further visualized through the water signal [*Journal of American Chemical Society*,2001-02-21;123(7):1517-8] [*Accounts of Chemical Research*, 2009-07-21;42(7):948-57] [*Magnetic Resonance in Medicine*. 2011-04-01;65(4):927-48]

The Z-spectrum and magnetization transfer ratio asymmetry (MTR_{asym}) curve were used to assess CEST signal contrast ($\text{MTR}_{\text{asym}} = (S^{\Delta\omega} - S^{-\Delta\omega})/S_0$, where $S^{-\Delta\omega}$ and $S^{\Delta\omega}$ are reference and label signals of RF saturation at $-\Delta\omega$ and $\Delta\omega$, respectively, and $\Delta\omega$ is the labile proton frequency shift from the water resonance. S_0 is the intensity of the bulk water CEST MR signal after irradiation at $-\Delta\omega$). The signal contrast varied as a function of the applied radio frequency (RF) saturation pulse parameters or pulse durations.

Therefore, CEST is not a way to directly provide the precise concentrations or pH values of metabolite indicators in the tumor microenvironment. In this research, we assessed the dynamic trends of acidic and hypoxic microenvironment changes in tumors through the analysis of the dynamic and amplified ^1H -CEST MR water signal trend.

The purpose of this study is to accurately provide ORTW (Optimized radiotherapy time window) through obtained ^1H -CEST MR signal trend. In this process, pH-sensitive chemical optical pH-1 microsensor was applied to verify our obtained ^1H -CEST MR imaging results, especially, the signal trend. The above statements are partially added to the discussion section to avoid any misapprehension. We respectfully request the reviewer to provide us with additional suggestions on effectively articulating this issue. Finally, we hope reviewer can revalue the impact of our work, the importance of the problem addressed, and the implications for future work in this interesting field.

Updated content added to Discussion section (Page 23, line 738-750) in the revised manuscript.

CEST constitutes a powerful sensitivity enhancement mechanism in which the signal of low concentration solutes can be amplified and further visualized through the bulk water signal. In this study, different from the single pH sensitivity agents, we have developed Gly-PFOBs with $^{19}\text{F}/^1\text{H}$ -CEST dual-modality imaging to dynamically provide important metabolic changes of the malignant tumor microenvironment, enabling synchronous pH and oxygen molecular imaging on a single MR machine. This complementary approach was also efficacious in illustrating the complexity and spatiotemporal heterogeneity of the oxygenation status and acidic microenvironment in solid tumors, providing integrated imaging information of precise ORTW. In this process, pH-sensitive chemical optical pH-1 microsensor was applied to verify the accuracy of the ^1H -CEST MR imaging results, especially the dynamic CEST signal trend.

7. In the reference section, the references are not presented in Nature Communication style. Moreover, there are no page numbers in many references, including references 15, 18, 29.

We appreciate the reviewer for carefully reading our manuscript and pointing out this error, which are very helpful to improve our manuscript. We have modified the reference format to align with the conventions of *Nature Communications*.

8. In Supplementary Figure 8, it is obvious the Gly-CEST fluorescence intensity in the spleen is far higher than that of the lung, while the quantitative analysis showed the opposite result. Moreover, in Supplementary Figure 14, there is no

obvious fluorescence signal in the spleen, while the quantitative analysis showed it is as high as in the heart and the kidney. There are obvious inconsistencies between the data. So the results are shaky. By the way, why there is an intense fluorescence signal in the spleen of Supplementary Figure 8 and no obvious fluorescence signal in the spleen of Supplementary Figure 14, in which the same nanoparticles were used?

We would like to express my gratitude to the reviewer for bringing up this inquiry. We sincerely apologize for the confusion caused by our inappropriate selection of representative image.

In the provided raw data (Raw data 1. a-b), it is observed that the fluorescence intensity of Gly-PFOBs (formerly named Gly-CESTs) in the spleen is higher than that in the lung in two data sets. In another two data sets, the fluorescence intensity of Gly-PFOBs in the spleen is lower than that in the lung. Considering the overall statistical findings, which indicate that the fluorescence intensity in the lung is higher than that in the spleen, we have decided to modify the representative image for Supplementary Figure 8 to avoid any unnecessary confusion. The statistical calculations and conclusions remain unchanged.

Raw data 1. Biodistribution of Gly-PFOBs in NCI-H460 tumor bearing mice after intravenous injection of Gly-PFOBs. a. Color bar values range from Min:5.12e7 to Max:1.66e9. b. Color bar values range from Min:1.00e8 to Max:4.15e9, n=4, i.v. 300 μ l. (IVIS image below “n=4, 01” is the result after intravenous injection of 50 μ l probe, which is not related with the content of main manuscript)

Raw statistic data 1

	Avg Radiant Efficiency [p/s/cm2/sr] / [μW/cm2]			
Heart	2.19E+07	1.02E+07	5.83E+07	3.71E+07
Liver	9.44E+08	7.77E+08	1.74E+09	1.63E+09
Spleen	6.20E+08	2.32E+08	2.14E+08	1.73E+08
Lung	2.00E+08	8.26E+07	8.16E+08	7.93E+08
Kidney	3.33E+07	2.23E+07	1.00E+08	5.60E+07
Tumor	1.58E+08	6.75E+07	4.40E+07	8.74E+07
Bone	2.37E+07	3.17E+07	6.33E+07	4.20E+07
Muscle	1.87E+07	1.19E+07	7.10E+07	2.90E+07
Intestine	2.91E+07	3.30E+08	5.93E+08	5.14E+08
Brain	2.15E+07	1.79E+07	7.51E+07	4.18E+07

Original Supplementary figure 8 updated to Supplementary figure 10

Updated Supplementary figure 10 (replaced original Supplementary figure 8)

Furthermore, the reviewer noted the absence of a discernible fluorescence signal in the spleen, as depicted in supplementary figure 14 (updated to supplementary figure 17 in revised manuscript). This lack of signal can be attributed to the color bar's display range. Specifically, upon adjusting the color bar values from the initial range of Min:4.48e4~Max:4.02e5 to Min:2.48e4~Max:4.02e5, the fluorescence signal of the spleen becomes clearly visible (Raw data 2. a-b).

Furthermore, due to variations in experimental objectives, experimental models, injection dosage, and post-injection sampling time between the two experiments presented in Table 3, it is unlikely that the fluorescence signals of major organs in these experiments could be exactly same.

Overall, the fluorescence intensity of the raw data in good accordance with the statistical results. There are no inconsistencies between the data. Once again, we apologize for the misinterpretation caused by the inappropriate use of the representative figure and the lack of emphasis on experimental methods. We have provided all raw data as well as statistical data to prove the accuracy of these results. We sincerely hope that the reviewer will excuse our negligence and reconsider and revalue the impact of our work. We express our sincere gratitude to the reviewer for affording us the chance to address a misinterpretation and provide a more comprehensive elucidation of the data.

Raw data 2

Raw data 2. Biodistribution of Gly-PFOBs in NCI-H209 liver metastasis mice model after intravenous injection of Gly-PFOBs. a. Color bar values range from Min:4.48e4 to Max:4.02e5. b. Color bar values range from Min:2.48e4 to Max:4.02e5. n=3, *i.v.* 200 μ l.

Raw statistic data

	Avg Radiant Efficiency [p/s/cm2/sr] / [μW/cm2]			
	30 min	1 h	2h30min	5 h
Heart	2.50E+04	1.57E+04	1.47E+04	1.17E+04
	2.03E+04	1.70E+04	1.58E+04	1.23E+04
Lung	2.12E+04	1.43E+04	1.40E+04	1.71E+04
	7.53E+04	3.96E+04	2.21E+04	1.92E+04
Liver	3.85E+04	2.91E+04	3.17E+04	1.96E+04
	7.70E+04	3.99E+04	1.29E+05	3.72E+04
Spleen	6.17E+04	8.72E+04	1.59E+05	4.15E+04
	8.41E+04	7.46E+04	4.77E+04	2.31E+05
Kidney	5.07E+04	2.29E+04	1.22E+04	1.29E+04
	1.48E+04	2.19E+04	1.29E+04	1.16E+04
Intestine	1.97E+04	2.23E+04	1.81E+04	1.37E+04
	3.53E+04	1.95E+04	2.23E+04	2.27E+04
Muscle	2.59E+04	2.41E+04	2.02E+04	1.64E+04
	3.53E+04	1.83E+04	2.29E+04	2.09E+04
Bone	4.02E+04	2.68E+04	2.09E+04	4.12E+04
	1.82E+04	5.40E+04	6.42E+04	3.42E+04
	2.26E+04	1.94E+04	6.09E+04	6.03E+04
	2.09E+04	1.35E+04	1.31E+04	1.34E+04
	2.23E+04	1.88E+04	1.61E+04	2.18E+04
	2.02E+04	1.56E+04	1.96E+04	9.44E+03

	2.05E+04	2.06E+04	1.39E+04	1.42E+04
	2.65E+04	3.32E+04	1.62E+04	1.28E+04
	2.20E+04	1.62E+04	1.92E+04	1.65E+04
Table 3. The difference between the above two groups of biological distribution experiments				
	Supplementary Fig.8 (updated to Supplementary Fig.10)		Supplementary Fig.14 (updated to Supplementary Fig.17)	
Nanoprobes	Gly-PFOBs (formerly named Gly-CESTs)		Gly-PFOBs (formerly named Gly-CESTs)	
Dose	300 μ l		200 μ l	
Manner	i. v.		i. v.	
Mouse model	NSCLC NCI-H460 xenograft tumor model		SCLC NCI-H209 liver metastasis model.	
Sampling time point	4 h		30 min, 1 h, 2h30min, 5 h	

Once again, we would like to express our heartfelt thanks to the reviewer#1 for your continued interest in our manuscript and the insightful comments that helped to strengthen our research results. We look forward to hearing from you regarding our manuscript. We would be glad to respond to any further questions and comments that you may have.

Reviewer #2 (Remarks to the Author): with expertise in radiation oncology, MRI, nanotechnology.

In this manuscript, a Gly-CEST based theranostic nanopaltform is developed for dual modality pH&Oxygen MR imaging, hence provide pivotal optimized radiotherapy time window (ORTW) information for boosted radiotherapy.

Thanks to its good biocompatibility, this theranostic nanoplatform shows its translational potential in MRI guided radiotherapy. Nevertheless, some issues need to be addressed

1. Please provide more details on radiation experiments, such as: what system used, what energy, is there image guidance, how many segments/fractions, in Material&Methods Section.

Thanks for the reviewer's suggestion. These descriptions were added to Methods Section (Page 11, line 348-349) in the revised manuscript and highlighted in yellow.

Description added to Methods section in revised manuscript (Page 11, line 348-349)

A single radiation dose of 6 Gy was delivered over the tumor sites using clinic radiotherapy system (VARIAN CLINAC 21EX, US) without MRI guidance.

2. In this study, all the MR Images are collected on 9.4T small animal system, wherein the high magnetic field affords higher detection sensitivity than clinic diagnostic MRI (3T) and MRI-guided radiation machine (e.g. 0.35T ViewRay, 1.5T Elekta). Please add more discussion on how to overcome the low-sensitivity challenge on MRI-guided irradiator.

We appreciate the reviewer's comments. This is indeed a very important issue and should be highlighted in the discussion. These descriptions were added in the Discussion section (Page 24, line 769-780) to the revised manuscript and highlighted in yellow.

Discussion added (Page 24, line 769-780)

One limitation of our work is that all imaging studies were performed on the 9.4 T MRI system. However, to enhance the clinical applicability of Gly-CESTs and their compatibility with the conventional 3.0 T MRI system, it is necessary to further

optimize and broaden the narrow chemical shift of Gly-CESTs (0.68 ppm). This is crucial as a significant chemical shift difference offers potential advantages for utilization at lower magnetic fields and to address the inherent challenge of relatively low sensitivity in MRI-guided irradiation. Furthermore, the development of saturation schemes, readout patterns, saturation-editing techniques, in conjunction with advanced postprocessing algorithms, has the potential to enhance the sensitivity of CEST imaging. Consequently, future research should focus on conducting more extensive investigations into Gly-CESTs with greater chemical shift frequency modification, as well as integrating updated CEST techniques.

3. Is it possible to developed active targeting Gly-CEST nanoplatform for tumor-targeted theranostics?

We are grateful to the reviewer for raising this important point. PFCs nano-emulsions possess a large surface area, high porosity, and ease of modification. Hence, PFCs based Gly-CESTs nanoplatform has the potential to be developed as a multifunctional magnetic resonance (MR) theranostics agents, which may promote the therapeutic outcomes and imaging performance. Our team has dedicated significant effort to the investigation of targeted theranostics based on perfluorocarbons (PFCs) over years. (Single low-dose INC280-loaded theranostic PFCE nanoparticles achieve multirooted delivery for MET-targeted primary and liver metastatic NSCLC [*Molecular Cancer*. 2022-12-01;21(1):212]; An osimertinib-perfluorocarbon nanoemulsion with excellent EGFR targeted therapeutic efficacy in non-small cell lung cancer [*ACS NANO*. 2022-08-23;16(8):12590-12605]; c-Met-Targeting ¹⁹F MRI nanoparticles with ultralong tumor retention for precisely detecting small or Ill-Defined colorectal liver metastases [*International Journal of Nanomedicine*. 2023-01-01; 18:2181-2196], *etc.*). Therefore, in the future, we may devote in developing active targeting Gly-CEST nanoplatform for tumor-targeted theranostics. Thanks again for the reviewer's insightful advice.

4. The ORTW in Figs. 5 and 6 is relatively short ~1 hr, which seems too short for clinic manipulations. Please provide discussion on how to prolong this ORTW.

As mentioned by reviewer, for intratumoral Gly-CESTs injection, ORTW was within 1 h to 2 h. For liver metastasis, the ORTW was between 1 h 30 min to 2 h 30 min following intravenous Gly-CESTs (O_2) injection. We have followed the reviewer's valuable suggestion and added the relevant discussion to the revised manuscript, as highlighted in yellow in Discussion sections (Page 24, line 780-798).

Discussion added (Page 24, line 780-798)

Although the duration of a single radiotherapy session for a clinical cancer patients is generally a few minutes to 20 minutes, for practical applications, there still needs to be a bit of time for irradiation field design and patient set up. From pH and O_2 dual-sensitive Gly-CESTs (O_2) ^{19}F -CEST dual-modality MR imaging, we could notice that ORTW was related to dynamic changes of tumor hypoxia and acidic microenvironment, and the time window width was not that large. Thus, rapid radiation field design for radiation therapy is needed. Another area requiring in-depth study is the optimization of the precise radiotherapy strategy through the precise spectral-temporal guidance of MRI. Gly-CESTs (O_2) ^{19}F -CEST dual-modality MR imaging strategy is particularly well suited to be combined with the current cutting-edge MR-guided RT system (Elekta Unity, Sweden). Further exploitation of this approach would hold great potential for future diagnostic imaging and precision RT. The next step would be to utilize ORTW information provided by ^{19}F -CEST MR molecular imaging to precisely delineate targeted areas. Of course, not all clinical centers are equipped with this kind of high-end equipment, thus, optimizing and modifying Gly-CESTs remains an important work in the next phase of research to make them better distributed in tumor lesions and function for longer periods of time. The faster ^{19}F -CEST dual-modality MR imaging sequences are also ideal for clinicians to obtain ORTW information.

Once again, we would like to express our heartfelt thanks to the reviewer #2 for your continued interest in our manuscript and the precious comments that helped to strengthen our manuscript. In addition to the above valuable comments, we have also received comments from other reviewers regarding the nomenclature of dual-mode imaging as "19F-CEST" and the designation of the probe as "Gly-CESTs". Consequently, in the revised manuscript and supplementary information, we have adopted the terms "19F/1H-CEST" and "Gly-PFOBs" respectively. We hereby

respectfully inform reviewer #2 of this modification. We are pleased to address any additional inquiries and remarks that you may possess.

Reviewer #3 (Remarks to the Author): with expertise in radiotherapy, imaging. This manuscript, entitled "Optimized radiotherapy time window-facilitated 19F-CEST dual-modality MR imaging with pH and O2 dual-sensitive Gly-CESTs for precision radiotherapy of lung cancer" by Rong A et al., demonstrates the feasibility of simultaneously measuring hypoxia status, including pH and O2 levels, determining the best RT windows, and using a probe to carry O2 to increase O2 levels in the tumor microenvironment to relieve hypoxia status. The authors employed versatile MRI and biological techniques and conducted extensive work to analyze and verify the results, making the data reliable and the manuscript interesting to read. However, the study does not introduce groundbreaking methodology, as the strategy is based on the combination of existing and well-developed methods, from probe preparation to data acquisition. It represents a novel application of existing techniques to generate synergistic effects and multifunctional methods, but they are not conceptually new. Moreover, there is no compelling advantage demonstrated over the numerous similar technologies that have been published over the years. Additionally, the manuscript requires substantial improvements, particularly in the Abstract and Introduction sections. The current descriptions in these sections are ambiguous and cumbersome, making it challenging to discern the study's purpose, methods, and findings. The writing in the Methods and Results sections is relatively lucid and aids in comprehending the study when read in reverse order.

We express our sincere gratitude to the reviewer for providing constructive comments, which have greatly contributed to the enhancement of our manuscript. We acknowledge that we did not adequately highlight the novelty of our research. In this study, for the first time, we disclosed the CEST signal properties of the commonly used glycerol. CEST, as a relatively promising magnetic resonance molecular imaging technique, holds considerable promise and has demonstrated exceptional capabilities in non-invasively assessing pH levels of the tumor extracellular acidic microenvironment, exhibiting excellent accuracy and heightened sensitivity. Glycerol can be present in the body, but the content is limited for specific and high sensitivity ¹H-CEST MR imaging. The glycerol-weighted PFOBs facilitate the interaction of sufficient exogenous glycerol

with the surrounding water, enabling responsiveness to environmental conditions and the attainment of ^1H -CEST MR imaging. Considering that our research team was the first to propose the utilization of glycerol, a commonly used substance, in the application of ^1H -CEST MR Imaging, and that all components of Gly-CESTs possess a clinically available safety profile, we regard this as the primary innovative aspect and strength of our study. Furthermore, we conducted a thorough investigation of the viability of Gly-CESTs in the precise ^{19}F -CEST image-guiding RT-sensitized strategy, particularly in determining the optimal radiotherapy time window (ORTW). With the introduction of most advanced MRI guided radiotherapy system in clinical practice, overcoming the disadvantages of CT and PET guided radiotherapy, the deep explorations and development of the multi-functional nanoprobe have been promoted to solve the still existing limitations in cancer RT. Our proposed Gly-CESTs (O_2) ^{19}F -CEST dual-modality MR imaging approach is particularly well suited to be combined with the current cutting-edge MR-guided RT system. Therefore, this work also provides potential applications and innovative research ideas for the development of MRI guided precision RT.

In accordance with the recommendations provided by the reviewer, we proceeded to condense, rephrase, and further elucidate the innovative aspects. Consequently, various revisions were implemented across the entirety of the manuscript, encompassing the Abstract and Introduction sections. In particular, to further elucidate clearly the mentioned point: “The Gly-CESTs facilitate the interaction of sufficient exogenous glycerol with the surrounding water, enabling responsiveness to environmental conditions and the attainment of ^1H -CEST MR imaging”, a series of *in vitro* and *in vivo* experiments were incorporated. We hope that the incorporation of the revised Abstract and Introduction, accompanied by the integration of experimental procedures and corresponding outcomes, along with the addition of a comprehensive mechanism diagram, will serve to augment lucidity and facilitate readers' comprehension of our research. These modifications have been incorporated into our revised manuscript, with the highlights indicated in yellow.

To verify the glycerol-retaining ability and the glycerol diffusion-limited feature of Gly-CESTs, we initially tested the glycerol content of Gly-CESTs in the dialysate at the *in vitro* level. Both Free Glycerol Assay Kit and Gas Chromatography-Mass Spectrometry (GC-MS) analysis were confirmed that only 1.04% and 1.33% glycerol

of Gly-CESTs free into the external environment (ultra-pure water or saline) after the 6 h of dialysis. Moreover, to investigate the glycerol-retaining capacity of Gly-CESTs *in vivo*, we assessed alterations in glycerol levels within subcutaneous tumor and liver tissue following intratumoral or intravenous administration, employing the Free Glycerol Assay Kit.

To mitigate the influence of endogenous glycerol *in vivo* and verify the findings, we prepared ^{13}C -Gly-CESTs using ^{13}C isotopically labeled glycerol and tested tissue ^{13}C -glycerol metabolic trend and concentration by ^{13}C -NMR spectra and GC-MS analysis. We found that the exogenous free glycerol, as a hydrophilic substance, tends to clear rapidly from tissues, regardless of intratumoral or intravenous injection, and maintains a consistently low level throughout the observation period. In comparison to the exogenous free glycerol injection group, the diffusion of glycerol occurs in a more gradual manner in Gly-CESTs injection group. Furthermore, Gly-CESTs effectively restrict the rapid diffusion of glycerol from the nanoprobe into the surrounding environment, irrespective of whether it is *in vitro* or *in vivo*.

A new figure added in the revised Supplementary Information (Page 3)

Supplementary Fig.3. Evaluation of the glycerol-retaining ability of Gly-CESTs.

a. The glycerol content in Gly-CESTs and in different dialysate (ultra-pure water or saline) at different time points was determined by Free Glycerol Assay Kit (abcam);
b. Gas Chromatography-Mass Spectrometry (GC-MS) analysis was performed to determine glycerol concentrations of Gly-CESTs after different duration of dialysis (1 h, 2 h, 4 h, 6 h). * $P < 0.05$, ** $P < 0.01$, *** $P < 0.001$, **** $P < 0.0001$. Data are presented as mean \pm standard deviation (SD) (n = 3).

A new figure added in the revised Supplementary Information (Page 7)

Supplementary Fig. 11. The determination of *in vivo* glycerol retaining status of Gly-CESTs in subcutaneous tumor or liver tissue. a. The glycerol concentration in tumor tissue was determined by Free Glycerol Assay Kit at different time points after intratumoral injection of Gly-PFOBs or free glycerol containing same amount of glycerol (623mM based on glycerol, 50 μ l). Meanwhile, the concentration of endogenous glycerol in tumor was also quantified for accurately reflect the exogenous glycerol concentration; b. The glycerol concentration in liver tissue was determined by Free Glycerol Assay Kit (abcam) at different time points after intravenous injection of Gly-PFOBs or free glycerol containing same amount of glycerol (623mM based on glycerol, 200 μ l). The concentration of endogenous glycerol in tumor was also quantified; c and e. ^{13}C -NMR (c), GC-MS (e) analysis was performed to determine the metabolic tendency or concentration of ^{13}C -glycerol in NCI-H460 tumor tissue at different time points after intratumoral injection of ^{13}C -Gly-PFOBs or free ^{13}C -glycerol containing same amount of ^{13}C -glycerol (623mM based on ^{13}C -glycerol, 50 μ l); d and f. ^{13}C -NMR (d), GC-MS (f) analysis was used to determine the metabolic tendency or concentration of ^{13}C -glycerol in liver tissue at different time points after intratumoral injection of ^{13}C -Gly-PFOBs or free ^{13}C -

glycerol containing same amount of ^{13}C -glycerol (623mM based on ^{13}C -glycerol, 200 μl). n. s., no significance, * $P < 0.05$, ** $P < 0.01$, *** $P < 0.001$, **** $P < 0.0001$. Data are presented as mean \pm standard deviation (SD) (n = 3).

In conclusion, the additional experiments and results conducted *in vitro* and *in vivo* have confirmed the glycerol-retaining ability and glycerol diffusion-limited characteristic of Gly-CESTs. These attributes ultimately allow PFC-based Gly-CESTs to achieve high sensitivity and facilitate the utilization of multi-functional and multi-modal imaging techniques, which have been successfully employed in precise radiotherapy for lung cancer guided by sensitive imaging. This also demonstrates the potential of multi-functional imaging in noninvasively monitoring and enhancing the effectiveness of integrated radiotherapy.

Some other comments include:

(i) In the Introduction, it may be helpful to briefly explain how radiation therapy (RT) works and why it is dependent on oxygen levels in the tumor microenvironment. It would also be helpful to highlight how hypoxia can cause radiation resistance, making it imperative to investigate hypoxia in cancer treatment. Additionally, it would be also useful to clarify what the RT time window is and why it is critical in RT.

We really appreciated the reviewer's quite reasonable suggestions. According to the reviewer suggestions, we proceeded to completely revise the whole paragraph in order to enhance its clarity for the reader. The updated sections were highlighted in yellow in revised manuscript (Page2 line 42-69). Thank you again for the helpful guidance.

Updated Introduction section (Page2 line 42-69)

Radiotherapy (RT) is a primary modality in cancer treatment, with more than half of all cancer patients receiving RT for curative or palliative reasons. Radiotherapy could induce irreparable DNA damage that positively correlated with the level of tumor oxygenation. However, the tumor hypoxic environment characterized by disordered vasculature and rapid proliferation of tumors is involved in the repair of radiation-mediated DNA damage, ultimately resulting in the failure of tumor eradication. Therefore, in attempting to achieve better outcomes, the RT encounters several

hurdles, such as radiation resistance from the hypoxic microenvironment and excessive radiation to overcome hypoxia causing damage to adjacent healthy tissues. Moreover, tumor hypoxia is often associated with the excessive accumulation of H^+ ions in the tumor microenvironment (TME), in turn, acidic TME further facilitates the development of hypoxic tumor regions. This vicious cycle eventually leads to the exacerbation of tumor hypoxia, RT resistance, and even worsen therapeutic efficacy. Clinically, RT sensitizers, such as oxygen, hyperbaric oxygen, and oxygen mimetics, have been used to overcome tumor hypoxia. Also, proton pump inhibitors, such as esomeprazole (EMSO) and lansoprazole (LAN), have been proven to be effective against tumor acidic microenvironment. With the introduction of most advanced MRI guided radiotherapy system in the clinic, magnetic resonance guided precision radiotherapy overcoming the disadvantages of CT and PET guided radiotherapy has entered a new stage, which further pushes the deep explorations and development of the multifunctional nanoprobe to solve the still existing limitations in cancer RT. Another critical issue that needs to be addressed while administering oxygen-based RT sensitizers is the optimal timing for RT. In other words, to obtain ideal synergistic therapeutic effects, it is essential to figure out whether the oxygen is effectively supplied to tumor hypoxia microenvironment, and what is the real-time status of tumor hypoxia and acidic microenvironment, and which time is optimal for the implementation of radiotherapy.

(ii) line 56-60: a long and unclear sentence, using terms like "the degree of hypoxia improvement" and "quantitative determination and visualization" without specifying which parameters are being measured, like O₂ level or pH? Line 59 change "include" to ", including"

We appreciate the reviewer's comments. To make it more comprehensible, we rephrased that sentence in introduction section (Page2 line 64-69).

Updated Introduction section (Page2 line 64-69)

Another critical issue that needs to be addressed while administering oxygen-based RT sensitizers is the optimal timing for RT. In other words, to obtain ideal synergistic therapeutic effects, it is essential to figure out whether the oxygen is effectively supplied to tumor hypoxia microenvironment, and what is the real-time status of

tumor hypoxia and acidic microenvironment, and which time is optimal for the implementation of radiotherapy.

(iii) In the Introduction, it may be helpful to include some comments on the design and synthesis of the probes, as well as clear chemical structures to help readers in understanding the study's experimental procedures and the mechanism how the probe work.

We express our gratitude for the valuable suggestions provided by the reviewer. It is our belief that a consensus can be reached regarding the occasional serendipitous nature of scientific discoveries. Specifically, the Gly-CESTs, being perfluorinated compounds (PFCs) based nano-emulsions, serve as the predominant synthetic precursor for nano-pharmaceuticals intended for pulmonary delivery within our research group. Initially, the design and synthesis of multifunctional probes of this nature were nonexistent. Serendipitously, we discovered the CEST signal characteristics of conventional nanoemulsions based on perfluorocarbons (PFCs) and identified glycerol as the source of this CEST signal. Additionally, we uncovered the pH and oxygen sensitivity of PFC-based nanoemulsions. To the best of our knowledge, no previous studies have reported the CEST effects of glycerol, nor have any studies demonstrated an oxygen sensitive CEST contrast agent thus far. Gly-CESTs represent the pioneering instance of perfluorocarbons (PFCs) possessing chemical exchange saturation transfer (CEST) characteristics for the purpose of visualizing pH and oxygen levels. Additionally, they can function as radiosensitizers, thereby playing a crucial role in providing an optimized radiotherapy time window (ORTW) for precision radiotherapy.

Above mentioned were also the innovation of our research. We would like to express our gratitude to the reviewer for affording us the opportunity to revise our work. We have clarified and emphasized relevant content in the introduction section. In this study, we focused on the progress of cutting-edge MR-guided radiotherapy systems and techniques in the current field of radiotherapy, and deeply explored the new applications of Gly-CESTs. Furthermore, the progress in equipment necessitates the development of contrast agents/sensitizers with substantial potential for clinical translation.

PFCs are inert organic compounds used as "artificial blood" that improve tissue oxygenation and have been widely used in the clinic for various purposes, including artificial blood substitution, organ preservation, ultrasound, and ^{19}F magnetic resonance imaging [*Journal of Experimental & Clinical Cancer Research*, 2021-06-21;40(1):197] [*Journal of Pharmaceutical Investigation*, 2023-01-01;53(1):153-190]. Moreover, there have been PFC nano-emulsions related phase I and phase III clinical trials. All these suggest that Gly-CESTs has great potential for clinical translation. Corresponding content was mentioned in our discussion section. Additionally, to enhance reader comprehension of our experimental procedures and mechanisms, we presented a schematic diagram (Fig.1), illustrating the probe's fundamental structure and the chemical structure of important chemicals (eg. Glycerol, $\text{C}_3\text{H}_8\text{O}_3$). This endeavor aims to facilitate the reader in acquiring a comprehensive understanding of the functional mechanisms of the probe. Furthermore, pertinent content has been revised and incorporated into the introduction section.

Fig. 1 Schematic illustration of ¹⁹F-CEST dual-imaging modality Gly-CESTs.

2. The authors should exercise caution when defining terms and making statements.

(i) The use of the technique name CEST to refer to a chemical probe, such as Gly-CESTs, can be confusing. While the probe is based on PFOB, it would be more appropriate to use a PFOB-related name to refer to the probe to avoid confusion.

We express our gratitude for the reviewer's suggestion. To mitigate any potential confusion, we have decided to rename the term "Gly-CESTs" as "Gly-PFOBs" throughout the entirety of the text. We hope that this revised nomenclature aligns with the naming convention of the probe. It is important to note that the designation of the control probe (PFOBs) without CEST signal remains unchanged.

(ii) The term "19F-CEST" used to describe the dual modality of fluorine and CEST MR in the manuscript may be misleading. Using the term "19F&CEST" instead would be more appropriate and accurate.

We would like to express our gratitude for your valuable suggestions aimed at enhancing the quality of this manuscript. In accordance with the nomenclature convention for multimodal imaging, we have made the necessary modification by replacing "¹⁹F-CEST" with "¹⁹F/¹H-CEST" throughout the entirety of the manuscript. This alteration aims to accurately depict the dual modality of fluorine and hydrogen proton-based CEST MR imaging technique. We hope that this revised nomenclature aligns with the terminology commonly used in the field of dual-modality MR imaging.

(iii) Line 45, “prognosis” could be replaced by “outcome”, since RT is not the tool for prognosis.

We express our gratitude to the reviewer for diligently reviewing our manuscript and identifying this error, which proves to be highly beneficial in enhancing the quality of our work. Consequently, we have substituted the term "prognosis" with "outcome".

(iv) Clarifying that perfluorinated compounds are prepared into emulsions would be helpful in understanding the description of "nano-size PFCs" (line 67) and similar to nanomaterials throughout the manuscript.

We express our gratitude to the reviewer for drawing our attention to this matter. In accordance with the reviewer's recommendation, the term "nano-size PFCs" has been substituted with "PFCs nano-emulsions" throughout the entirety of the manuscript.

(v) line 69, “exchangeable molecules” is unclear, since talking about CEST, “exchangeable protons” will be clear enough.

We express our gratitude to the reviewer for their valuable suggestion. We have modified “exchangeable molecules” to “exchangeable protons” according to the reviewer’s suggestion.

(vi) Line 75-80, not sure the statement is accurate. Is this MOST advanced MRI guided the radiotherapy system is a milestone?

A review paper published in 2021 in Radiology, MRI-guided Radiation Therapy: An Emerging Paradigm in Adaptive Radiation Oncology including many MRI guided RT studies.

Thanks to the reviewer for the comments. According to the latest report: “On February 28, 2023, Elekta Medical (Elekta, Stockholm:EKTA-B) announced that the Elekta Unity MR-Linac radiation therapy System received 510(k) approval from the FDA, marking a new era in precision radiation therapy for cancer.” Above review paper published in 2021 in Radiology mainly reviewed the development of MRI-guided Radiation Therapy, including the clinical trials (2017-2020) of MRI guided RT, and indicated that there are still ongoing developments in MRI-guided RT, and signifies that MRI-guided RT promises to be the next big step in RT.

However, due to variations in the years of clinical implementation of MRI-guided Radiation Therapy across different countries, we have opted to rephrase the sentence without specifying the exact timeframe and have included a pertinent reference. We appreciate the reviewer's consideration.

(vii) Line 382. The exchange rate depends on temperature not pH, higher pH suggests low free H⁺, which impact the CEST.

We appreciate the valuable feedback provided by the reviewer. The chemical exchange saturation transfer (CEST) technique measures proton exchange between the exchangeable protons of the solute with the much larger pool of bulk water

protons [*Science Translational Medicine*, 2015-10-14;7(309):309ra161]. The magnetization of exchangeable solute protons is labeled by spin manipulations (e.g., saturation) and transferred to water via exchange. The exchange rates of these exchangeable protons often depend on pH, and CEST has been used to image pH changes ([*Journal of the American Chemical Society*, 2016-09-07;138(35):11136-9], [*Chemical Society Reviews*, 2006-06-01;35(6):500-11], [*Chemical Society*, 2017-06-01;8(6):4424-4430], [*Journal of Biomolecular NMR*, 2005-07-01;32(3):195-207]). Therefore, pH does significantly affect exchange rates of nearly all exchangeable protons. To enhance the comprehensibility and coherence of our work, we have incorporated the pertinent references for the convenience of our readers.

(viii) Line 162 and 205, “Anti-acid treatment” may not be a correct term, or should be “antacid”

We express our gratitude to the reviewer for diligently reviewing our manuscript. In accordance with the reviewer's recommendations, we have made the necessary amendment of replacing "anti-acid" with "antacid" throughout the entirety of the manuscript.

(ix) Line 624-625: is this a true and accurate statement?

We appreciate the reviewer's question and would like to affirm the veracity of our statement. It is worth noting that our research team is actively engaged in the development of radiotracers for tumor molecular imaging. However, it is important to acknowledge that radiotracers predominantly offer restricted or singular metabolic information, necessitating substantial time for processing and interpretation.

(x) Line 643: is this an accurate statement, metal ion itself could be toxic, but once coordinating with inert ligands (very slow exchange), the complex can be very stable and non-toxic. There are many medical applications of using metal complexes as contrast agents.

We express our gratitude to the reviewer for drawing our attention to this matter. It is indeed plausible that the inorganic nanoparticles may demonstrate negligible toxicity when administered at low dosage regimens. Therefore, we would like to reformulate the sentence “Most radiosensitizers are inorganic materials with considerable toxicity” into “Most radiosensitizers are inorganic materials, which may have potential long-term toxicity.

(xi) Line 647 – 650: PFCs could be low toxic, but some PFCs disrupt normal endocrine activity; reduce immune function; cause adverse effects on multiple organs, including the liver and pancreas.

This point holds significant importance and has been elucidated in the discussion section of the revised manuscript.

As stated by the reviewer, certain fluorinated compounds, namely perfluorooctane sulfonate (PFOS) and perfluorooctanoate (PFOA), are currently recognized for their persistent, bioaccumulative, and toxic characteristics. However, it is improbable for perfluorooctylbromide (PFOB) to exhibit toxicity.

In 1996, perfluorooctylbromide (PFOB), also known as perflubron, partial liquid ventilation was shown to alleviate neonatal respiratory distress syndrome and improve pulmonary function, due to its high oxygen dissolving and releasing capability, low surface tension, and excellent physical chemical properties [*New England Journal of Medicine*. 1996-09-12;335(11):761-7]. PFOB used in this study stands out among PFCs for medical use as it, non-toxic, high stability, inertness, possess the unique property of being both hydrophobic and lipophobic, and an acceptable excretion profile [*New England Journal of Medicine*. 1996-09-12;335(11):761-7]. It was less likely to have chemical reactions and intermolecular interactions, thus, have been widely used in the clinic for various purposes, including artificial blood substitution, organ preservation, ultrasound, and ¹⁹F magnetic resonance imaging [*Journal of Experimental & Clinical Cancer Research*, 2021-06-21;40(1):197] [*Journal of Pharmaceutical Investigation*, 2023-01-01;53(1):153-190]. Of course, PFOB nano-emulsions may exhibit potential toxic when modified with other functional groups or drugs. The prepared Gly-PFOBs (formerly named Gly-CESTs) exhibit excellent dispersibility in water, which can be attributed to the specific surfactant combination of

dipalmitoyl phosphatidylcholine and cholesterol. The incorporation of this simple surfactant combination ensured the absence of any supplementary toxicity. In addition, the biosafety and biocompatibility of PFOB based Gly-PFOBs (formerly named Gly-CESTs) also confirmed in this study. The aforementioned content has been incorporated into the discussion section (Page 24, line 761-766).

3. Methods

Line 100. Description of Preparation of Gly-CESTs and PFOBs are unclear and confusing. It would be better to have a clear description of how many probes are prepared, what they are, give the proper name, and how to prepare each of them.

We express our gratitude for the valuable feedback provided by the reviewer. To enhance reader comprehension, we have revised the description pertaining to the preparation of Gly-PFOBs (formerly named Gly-CESTs) and PFOBs in the Methods section, emphasizing these modifications by highlighting them in yellow.

Description added to Methods section on revised Manuscript (Page 4, line 114-133)

The lissamine rhodamine B sulfonyl labelled perflubron based Gly-PFOBs were prepared as described previously. The purpose of incorporating lissamine rhodamine B sulfonyl was to investigate cellular uptake through confocal fluorescence microscopy and to examine *ex vivo* biodistribution using IVIS Imaging. The components of Gly-PFOBs were 20% (v/v) PFOB, 76% (v/v) ultra-pure water, 2% (v/v) glycerol, and 2% (w/v) of a surfactant. The surfactant (lipids) included 88.9 mol% dipalmitoyl phosphatidylcholine (DPPC), 1 mol% 1,2-dipalmitoyl-sn-glycero-3-phosphoethanolamine (DPPE), 10 mol% cholesterol, and 0.1 mol% lissamine rhodamine B sulfonyl (16:0 LissRhod PE). Briefly, the lipids were dissolved in a mixture of methanol and chloroform, filtered through a small bed of cotton, evaporated under reduced pressure using a rotary evaporator at 45 °C to form a thin film, and then further dried in a vacuum oven (45 °C) for 24 h. The resuspended surfactant was combined with PFOB, water, and glycerol according to the above proportions. The solution was extruded through an Avanti Mini Extruder (Alabaster, Alabama) with a 200 nm nucleopore polycarbonate membrane. The completed

emulsions were placed in crimp-sealed vials, blanketed with argon, and stored at 4 °C until use.

The preparation procedure for perflubron-based nano-molecular imaging probes (PFOBs), which are CEST signal-free control probes, was identical to that of Gly-PFOBs. However, there were slight variations in the components used. Specifically, PFOBs consisted of 20% (v/v) PFOB, 78% (v/v) ultra-pure water, and 2% (w/v) surfactant, without the addition of glycerol. Consequently, two types of probes were prepared: Gly-PFOBs and PFOBs.

(i) Did not mention how to make Gly-CESTs,

To enhance reader comprehension, we have revised the description pertaining to the preparation of Gly-PFOBs (formerly named Gly-CESTs) and PFOBs in the Methods section, emphasizing these modifications by highlighting them in yellow.

(ii) Line 101: Did not mention the purpose of using fluorescence previously, while making rhodamine B labelled perflubron-Gly-CESTs here.

We express our gratitude to the reviewer for diligently reviewing our manuscript and providing valuable feedback. The utilization of rhodamine B in this study is intended for investigating cellular uptake and conducting *ex vivo* biodistribution studies through confocal fluorescence or IVIS Imaging. The purpose of employing fluorescence has been incorporated into the relevant methods section.

(iii) line 102, Gly-CESTs consisting of 20% PFOB and 2% of a surfactant, then rest of what?

Thanks for reviewer's questions. The components of Gly-PFOBs (formerly named Gly-CESTs) were 20% (v/v) PFOB, 76% (v/v) ultra-pure water, 2% (v/v) glycerol, and 2% (w/v) of a surfactant. The surfactant (lipids) included 88.9 mol% dipalmitoyl phosphatidylcholine (DPPC), 1 mol% 1,2-dipalmitoyl-sn-glycero-3-phosphoethanolamine (DPPE), 10 mol% cholesterol, and 0.1 mol%

lissaminerhodamine B sulfonyl (16:0 LissRhod PE). We have updated the methods section accordingly (Page 4, line 117-121) .

(iv) Line 111. “The synthesis of”, it seems no reaction going on, it may just say the “preparation”...

We express our gratitude for the valuable suggestions provided by the reviewer. In accordance with the reviewer's recommendation, we have replaced the term "synthesis" with "preparation" throughout the entirety of the manuscript.

4. Results

Line 348: no “synthesis” is described, only “characterization” in this section.

We express our gratitude for the valuable suggestions provided by the reviewer. In accordance with the reviewer's recommendation, we have modified the title from "Synthesis and characterization of $^{19}\text{F}/^1\text{H}$ -CEST dual-imaging modality Gly-PFOBs" to "Characterization of $^{19}\text{F}/^1\text{H}$ -CEST dual-imaging modality Gly-PFOBs."

5. Figures: Figure 3 is small and difficult to read, panel a is pixelated.

The figures are cluttered and many panels could be moved to the supplemental to highlight the more important findings.

The figure legends do not clearly describe what is being done in the experiments and requires going back and forth between the text and figure to understand the experiment.

We express our gratitude to the reviewer for providing this valuable suggestion. Within main figures, we have emphasized the more significant findings, and we reorganized the Figure 3 to make it more legible. It seems that the manuscript turns to be a bit fuzzy after uploading. Therefore, when submitting the revised manuscript, we have also uploaded the high-resolution original image in PDF format to facilitate easy comprehension by the reviewers. Furthermore, we have made the necessary revisions to the figure legends as suggested.

Rearranged Figure 3

Updated figure legends:

Fig. 1 Schematic illustration of $^{19}\text{F}/^1\text{H}$ -CEST dual-imaging modality Gly-PFOBs.

Fig. 2 Characterization of $^{19}\text{F}/^1\text{H}$ -CEST dual-imaging modality Gly-PFOBs. a. Morphologies of Gly-PFOBs and control probes without CEST signals (PFOBs) were determined by TEM. Scale bars, 1 μm . b. Elemental mapping analysis of Gly-PFOBs. Scale bars, 50 nm. c. Measurement of diameter and zeta potential changes of Gly-PFOBs. d. CEST signal properties of Gly-PFOBs with different glycerol concentrations (0 mM, 208 mM, 415 mM and 623 mM) at different saturation pulse powers (0.6 μT , 0.8 μT , 1.0 μT , 1.2 μT and 1.8 μT). e. ^1H -CEST MR imaging results of aqueous glycerol solutions with different concentrations. f. $T_1\text{WI}$, $^{19}\text{F}/^1\text{H}$ -CEST MR imaging results of Gly-PFOBs with different glycerol concentrations.

Fig.3 *In vitro* pH and O_2 dual sensitivities and hypoxia alleviation by oxygenated Gly-PFOBs. a. Z-Spectra and MTR_{asym} curve of the Gly-PFOBs at different pH values and saturation pulse powers (0.6 μT , 0.8 μT , 1.0 μT , 1.2 μT , and 1.8 μT , 5 s), and corresponding statistical analysis (b). c. ^1H -CEST MRI (1.0 μT , 5 s), $T_1\text{WI}$, and ^{19}F MRI results of Gly-PFOBs in different pH solutions. d. Comparison of CEST signal intensity between Gly-PFOBs and PFOBs in the supernatant of NCI-H460 cells incubated with ESOM or saline, * $P < 0.05$; n.s indicates no statistical significance ($P > 0.05$). Data are presented as mean \pm SD ($n = 3$). e. the pH value of the supernatant of NCI-H460 cells incubated with ESOM or saline in pHmed for 2 h were measured by pH-sensitive optical microsensors, *** $P < 0.001$. Data are presented as mean \pm SD ($n = 6$). f. Measurement of oxygen loading and gradual release from PFOB saturated with oxygen in deoxygenated water and corresponding statistical analysis (g), **** $P < 0.0001$. Data are presented as mean \pm SD ($n = 3$). h. CEST signal intensity changes of Gly-PFOBs with oxygen and argon (Ar) release over time. i. CEST signal intensity changes of Gly-PFOBs with oxygen release. j. Immunofluorescence results of HIF-1 α staining of NCI-H460 hypoxic cells co-incubated with Gly-PFOBs with or without oxygenation and corresponding statistical results (k). *** $P < 0.001$. Data are presented as mean \pm SD ($n = 3$). Scale bar: 50 μm .

Fig.4 Detection of *in vivo* pH changes. a. Representative $T_1\text{WI}$, $^{19}\text{F}/^1\text{H}$ -CEST MRI, and CSI results following Gly-PFOBs injection. b. Statistical results of $^{19}\text{F}/^1\text{H}$ -CEST MR signal intensities. ** $P < 0.01$, n. s., no significance. Data are presented as mean \pm SD ($n \geq 3$). c. H&E staining of the non-ischemic and ischemic liver. d. Pimonidazole hydrochloride immunohistochemical staining of the non-

ischemic and ischemic liver. e. the pH values of the non-ischemic and ischemic liver were measured by pH-sensitive optical microsensor, ** $P < 0.01$. Data are presented as mean \pm SD (n = 6). f. CEST MR images of Gly-PFOBs, PFOBs and glycerol in monitoring antiacid therapy, and corresponding statistical analyses (g, h), * $P < 0.05$; ** $P < 0.01$; n. s indicates no statistical significance ($P > 0.05$). Data are presented as mean \pm SD (n = 4). i. pH determination in the TME of NCI-H460 subcutaneous tumor-bearing mice treated with ESOM or saline, ** $P < 0.01$. Data are presented as mean \pm SD (n = 6).

Fig.5 ORTW and RT outcomes in NSCLC xenograft tumor model. a. CEST and BOLD MR imaging of BALB/c nude mice bearing NCI-H460 lung xenografts after intra-tumoral injection of oxygenated Gly-PFOBs(O₂) or PFOBs(O₂) (b) over time. c. Dynamic CEST MR signal changes of tumor region after Gly-PFOBs (O₂) or PFOBs(O₂) injection. * $P < 0.05$; ** $P < 0.01$; **** $P < 0.0001$. Data are presented as mean \pm SD (n \geq 6). d. Determination of the pH changes in the TME before and after Gly-PFOBs (O₂) probe injection after fully oxygenated. **** $P < 0.0001$. Data are presented as mean \pm standard deviation (SD) (n = 6). e. Monitoring of tumor tissue oxygenation by calculating changes in T_2^* values. * $P < 0.05$. Data are presented as mean \pm SD (n = 5). f-g. Statistical results of relative tumor volume and tumor weight (e) of each treatment group. * $P < 0.05$, ** $P < 0.01$, **** $P < 0.0001$. Data are presented as mean \pm standard deviation (SD) (n = 4). h. Representative H&E, CD31, Ki67, and TUNEL-related antigen staining in different treatment groups (n=4), scale bar: 100 μ m.

Fig. 6 ORTW and RT outcomes in SCLC metastasis in the liver. a. Dynamic CEST MR “M-type” signal changes curve of H209 SCLC liver metastasis and normal liver tissue after the injection of Gly-PFOBs(O₂), Gly-PFOBs, and PFOBs. * Gly-PFOBs-Tumor vs. PFOBs-tumor; # Gly-PFOBs(O₂)-Tumor vs. PFOBs-tumor; & Gly-PFOBs(O₂) -Tumor vs. Gly-PFOBs-tumor. * $P < 0.05$; ** $P < 0.01$; *** $P < 0.001$; **** $P < 0.0001$. Data are presented as mean \pm SD (n = 6). b. Determination of the pH changes and T_2^* values in TME before and after Gly-PFOBs(O₂) (pH of 2 h 30 min vs. 0 h, * $P < 0.05$). Data are presented as mean \pm SD (n = 3). c. Representative T₂WI MR images of mice liver in each treatment group. Yellow arrow points to H209 SCLC liver metastasis. d. H209 SCLC liver metastasis foci on each liver were counted. ** $P < 0.01$; *** $P < 0.001$; **** $P <$

0.0001. Data are presented as mean \pm standard deviation (SD) ($n \geq 5$). e. Body weight changes of mice in each treatment group. f. Representative H&E-stained liver slices collected from mice after indicated treatment on day 14. Black arrow points to H209 SCLC liver metastasis. Scale bar: 5 mm, and corresponding enlarged lung metastasis sections. Scale bar: 200 μ m.

Supplementary Table 1. The glycerol content in Gly-PFOBs and in different dialysate (ultra-pure water or saline) dialyzed Gly-PFOBs at different time points was determined by Free Glycerol Assay Kit (abcam). The calculation of the glycerol content in Gly-PFOBs after dialyzed against different dialysate was based on the theoretical concentration of glycerol in the Gly-PFOBs, which was 623mM.

Supplementary Table 2. The glycerol concentration in Gly-PFOBs without dialysis and glycerol concentration of Gly-PFOBs dialyzed against different dialysate (ultra-pure water or saline) at different time points was determined by Gas Chromatography/Mass Spectrometry (GC/MS) analysis.

Supplementary Fig.3. Evaluation of the persistence of glycerol component on the Gly-PFOBs. a. The glycerol content in Gly-PFOBs and in different dialysate (ultra-pure water or saline) dialyzed Gly-PFOBs at different time points was determined by Free Glycerol Assay Kit (abcam); b. Gas Chromatography-Mass Spectrometry (GC-MS) analysis was performed to determine glycerol concentrations of Gly-CESTs after different duration of dialysis (1 h, 2 h, 4 h, 6 h). * $P < 0.05$, ** $P < 0.01$, *** $P < 0.001$, **** $P < 0.0001$. Data are presented as mean \pm standard deviation (SD) ($n = 3$).

Supplementary Fig. 6. Statistical results of ^{19}F -MRI signal-to-noise ratio (SNR) of the phantom. a. ^{19}F -MRI SNR statistical results of Gly-PFOBs with different glycerol concentrations b. ^{19}F -MRI SNR statistical results of Gly-PFOBs in different pH solutions. n. s., no significance. Data are presented as mean \pm standard deviation (SD) ($n = 3$).

Supplementary Fig. 11. The determination of *in vivo* glycerol retaining status of Gly-PFOBs in subcutaneous tumor or liver tissue. a. The glycerol concentration in tumor tissue was determined by Free Glycerol Assay Kit at different time points after intratumoral injection of Gly-PFOBs or free glycerol containing same amount of glycerol (623mM based on glycerol, 50 μ l); b. The endogenous glycerol concentration in tumor or liver tissue. c. The glycerol

concentration in liver tissue was determined by Free Glycerol Assay Kit (abcam) at different time points after intravenous injection of Gly-PFOBs or free glycerol containing same amount of glycerol (623mM based on glycerol, 200 μ l); c and e. ^{13}C -NMR (c), GC-MS (e) analysis was performed to determine the metabolic tendency or concentration of ^{13}C -glycerol in NCI-H460 tumor tissue at different time points after intratumoral injection of ^{13}C -Gly-PFOBs or free ^{13}C -glycerol containing same amount of ^{13}C -glycerol (623mM based on ^{13}C -glycerol, 50 μ l); d and f. ^{13}C -NMR (d), GC-MS (f) analysis was used to determine the metabolic tendency or concentration of ^{13}C -glycerol concentration in liver tissue at different time points after intratumoral injection of ^{13}C -Gly-PFOBs or free ^{13}C -glycerol containing same amount of ^{13}C -glycerol (623mM based on ^{13}C -glycerol, 200 μ l). n. s., no significance, * $P < 0.05$, ** $P < 0.01$, *** $P < 0.001$, **** $P < 0.0001$. Data are presented as mean \pm standard deviation (SD) (n = 3).

6. Some other minor issues:

Line 50: change “leading to ” to “lead”

Line 75-75: “with the mostwas introduced...” wrong gramma, change to “with the introduction of”

We extend our sincere apologies for the grammatical issues identified in our manuscript and express our gratitude to the reviewer for their meticulous reading and bringing this error to our attention. In addition, the manuscript has been edited by a professional native English language editor on scientific papers to ensure accuracy and clarity of our manuscript.

Line 145, 220 and others: wrong unit “mM/L” expression was used, it should be mM or mmol/L.

We express our gratitude for thoroughly reviewing our manuscript. We have diligently conducted a comprehensive examination of the manuscript, identifying, and rectifying all instances of spelling errors.

Line 163: insert reference properly, instead of using PMID number.

We express our gratitude to the reviewer for their valuable suggestion. We inserted relative reference.

Line 225: no need to write “(T2-star)”

We express our gratitude to the reviewer for providing this valuable suggestion. We deleted (T2-star) in the corresponding sentence.

Line229: “19F (19F) MRI” is strange, 19F MRI will be good.

We express our gratitude to the reviewer for providing this valuable suggestion, and we are eager to incorporate this recommendation.

Line 272: It is better to use the term "intravenous injection" instead of "i.v. injection". Line 474 suddenly showed up “the intratumoral injection” The manuscript should specify the type of injection used at the beginning and consistently throughout the text to avoid confusion. It should also clarify what kind of injection was used for the biocompatibility study (line 593) and whether it was injected into healthy or tumor mice.

In accordance with the recommendations provided by the reviewer, we have made alterations to the phrasing as suggested.

In this study we established three kinds of mouse model:1) NCI-H460 NSCLC subcutaneous xenograft tumor bearing mouse model; 2) NCI-H209 SCLC liver metastasis mouse model;3) Liver ischemia mouse model. Hence, except for the NCI-H460 NSCLC subcutaneous xenograft tumor bearing mouse model, which utilized intratumoral injection to investigate the optimized radiotherapy time window (ORTW), all other mouse models were administered intravenously via the tail vein. We concur with the reviewer's suggestion to explicitly specify the injection method to prevent any ambiguity. Accordingly, we have made the necessary revisions to the In vivo MRI studies section in the revised manuscript. (Page 7-Page 8).

The *in vivo* toxicity of Gly-PFOBs (formerly named Gly-CESTs) was evaluated in healthy BALB/c nude mice following the intravenous injection of the probes. We have made further improvements to the pertinent content. Once again, we express our gratitude to the reviewer for their meticulous examination.

Line 297: “from different groups”, although it mentioned in PDX mouse preparation, it would be better to say it again here what the groups are.

We express our gratitude to the reviewer for their valuable suggestion. In accordance with the reviewer's recommendation, we have mentioned all research groups in the appropriate section.

Line 375: confusing sentence to read through.

We re-worded the sentence “Based on the above imaging results, we applied lower saturation powers (0.8 μ T or 1.0 μ T) with 5 s pulse duration for 623 mM Gly-CESTs as broadening the direct saturation line width at higher saturation powers might partially obscure the CEST effects in the *in vivo* experiments.” into “In consideration of the potential obscuring of CEST effects *in vivo* experiments caused by the broadening of the direct saturation line width at higher saturation powers, we opted to utilize lower saturation powers (0.8 μ T or 1.0 μ T) with a pulse duration of 5 s in subsequent experiments.”

Line 618: change “resistant” to “resistance”

We have changed “resistant” to “resistance” in corresponding sentence.

Line 694: It would be better to point out in caption the numbers on the top of Fig 2d are glycerol concentration.

We express our gratitude for the valuable suggestion provided by the reviewer. In accordance with the reviewer's recommendation, we have duly annotated in the caption that the numbers on the top of Fig 2d correspond to the glycerol concentration.

Line 697 Fig. 3.: There are spaces between labels like “Gly-C EST s” and “PF OBs”

We appreciate the reviewer's considerate reminder. It appears that there has been a slight alteration in the format of the manuscript during the uploading process. We have thoroughly reconfirmed that our original manuscript did not contain any superfluous spaces.

Line 715 Fig. 4.: Boxes outlined texts in c and d. in h legend should be saline not PBS.

Thanks to the reviewer for checking all this carefully.

The identified inaccuracies have been rectified accordingly.

Line 720. Fig 4 e.: would indicate pH value measured by microsensor.

Thanks for reviewer 's suggestion. we have indicated in the caption that the pH values were measured by pH-sensitive optical microsensor.

Once again, we would like to express our heartfelt thanks to the reviewer# 3 for your continued interest in our manuscript and the precious comments that helped to strengthen our manuscript. We look forward to hearing from you regarding our manuscript. We would be glad to respond to any further questions and comments that you may have.

Reviewer #4 (Remarks to the Author): with expertise in imaging, probes.

First, I was very interested in the paper and the extensive high level studies employed. As such, I made suggested editorial suggests for the abstract and introduction that improve the grammar and understanding at least for me.

Thanks for the reviewer's highly positive comments about our work. In addition, we really appreciate the reviewer for the nice editorial suggestions. The section of abstract and introduction have been revised following the suggestions of the reviewer. In addition, the manuscript has been edited by a professional native English language editor on scientific papers to ensure accuracy and clarity of our manuscript.

I am familiar with all aspects of this paper and I have no significant comments regarding the studies, but I have one fundamental questions that could be clarified.

1) The glycerol is added to the water and PFC. The effects are glycerol level-dependent up to a maximal effect plateau. Fine, but glycerol is not chemically coupled to the particle but only in the particle environment. In vitro this environment is constrained. Glycerol is not miscible in PFC, virtually nothing is. How is it associated with the particles themselves? In vivo, how does the glycerol remain with the particles and not diffuse into tissues, and behave like the control particles? Was phosphatidyl glyceride (PG) considered? The particles are surprisingly anionic, -60mV, and this is very negative for a DPPC/cholesterol/PE surfactant which should be -20 to -40 mV. Adding PG to this surfactant may make the particle Zeta potential more negative, which could trigger acute complement activation. So there are issues to work through to make the compelling results jive with these unaddressed points.

For the question about "glycerol":

We wholeheartedly concur with the reviewer's assertion regarding the significance of this point and express our utmost gratitude for the reviewer's comprehensive and valuable insights. We sincerely hope that our explanations will sufficiently address all inquiries.

As the reviewer mentioned, glycerol is not chemically coupled to the particle, and glycerol is not miscible in PFOB. In the preparation of Gly-CESTs, the nanoprobe

were composed of a hydrophobic PFOB core and a surrounding lipid with a hydrophobic inner layer and hydrophilic outer layers, the glycerol should stay at the surface environment of the nanoparticles. To address the reviewer's inquiries more comprehensively, we explored the diffusion characteristics and *in vivo* pharmacokinetics of glycerol in Gly-CESTs, and conducted a series of experiments to enrich our manuscript:

Firstly, to address the reviewer's concern regarding the potential rapid diffusion of the glycerol component from the nanoprobe, we performed *in vitro* dialysis experiments. Free Glycerol Assay Kit (ab65337) was firstly used to test the glycerol content in the dialysate. In the assay, glycerol is enzymatically oxidized to generate a product which reacts with the probe to generate color ($\lambda = 570$ nm) or fluorescence (Ex/Em = 535/587 nm). This assay can detect 50 pmol ~10 nmol of glycerol sensitively and accurately. Specifically, Gly-CESTs (with a glycerol content of 3115 μmol in 5 ml) were subjected to dialysis against either 1000 ml of ultra-pure water or normal saline dialysate media for varying durations of 1 h, 2 h, 4 h, and 6 h (Spectra/Por, COMW 1000). The dialysate samples obtained at each time point were subsequently analyzed to determine the glycerol content (50 μl /well).

The results showed that the concentration of glycerol in the dialysate slightly increased over time. At the 6 h of dialysis, the glycerol concentration measured in ultra-pure water and normal saline were found to be 32.3 ± 0.39 μmol , and 41.43 ± 3.25 μmol , respectively. We added a new table with the mean and standard deviation to Supplementary Information of the revised manuscript (Supplementary Table 1). In both ultra-pure water and normal saline environments, the release of glycerol from the Gly-CESTs into the external environment after 6 hours of dialysis was found to be only 1.04% (32.3 μmol / 3115 μmol) and 1.33% (41.43 μmol / 3115 μmol), respectively. These results were also added to the Supplementary Information of the revised manuscript (Supplementary Fig. 3a).

To further verify the above results, the Gas Chromatography-Mass Spectrometry (GC-MS) analysis was performed to quantify glycerol concentration in Gly-CESTs dialyzed against ultra-pure water or saline for different lengths of time. Corresponding statistical results showed that the variation occurring in the reducing glycerol concentration appeared to be minor. We added a new table with the mean and standard deviation to Supplementary Information of the revised manuscript (Supplementary Table 2).

Compared with Gly-CESTs without dialysis, the glycerol concentration in Gly-CESTs was only slightly decreased after dialyzed against ultra-pure water or saline (Supplementary Fig. 3b). These results were consistent with the results of the Free Glycerol Assay Kit.

A new table added in the revised Supplementary Information (Page 2)

	Gly-CESTs dialyzed against ultra-pure water(μmol)	Ultra-pure water(μmol)	Gly-CESTs dialyzed against saline (μmol)	Saline (μmol)
1 h	3099 ± 0.59	16.04 ± 0.59	3098 ± 0.84	16.74 ± 0.84
2 h	3094 ± 0.38	21.43 ± 0.38	3086 ± 0.87	29.14 ± 0.87
4 h	3090 ± 0.86	25.23 ± 0.86	3078 ± 1.8	37.16 ± 1.80
6 h	3083 ± 0.39	32.30 ± 0.39	3074 ± 3.3	41.43 ± 3.25

Supplementary Table 1. The glycerol content in Gly-CESTs and in different dialysates (ultra-pure water or saline) at different time points were determined by Free Glycerol Assay Kit (abcam).

A new table added in the revised Supplementary Information (Page 2)

	Gly-CESTs without dialysis	Gly-CESTs dialyzed against Ultra-pure water (mmol/L)	Gly-CESTs dialyzed against Saline (mmol/L)
1 h	623.51 ± 39.42	622.93 ± 12.18	572.44 ± 33.97
2 h		615.39 ± 4.87	571.62 ± 12.72
4 h		614.64 ± 18.42	562.21 ± 20.13
6 h		612.96 ± 21.5	561.51 ± 15.83

Supplementary Table 2. The glycerol concentration in Gly-CESTs without dialysis and glycerol concentration in Gly-CESTs dialyzed against different dialysates (ultra-pure water or saline) at different time points were determined by Gas Chromatography-Mass Spectrometry (GC-MS) analysis.

A new figure added in the revised Supplementary Information (Page 3)

Supplementary Fig.3. Evaluation of the glycerol-retaining ability of Gly-CESTs.

a. The glycerol content in Gly-CESTs and different dialysate (ultra-pure water or saline) at different time points was determined by Free Glycerol Assay Kit (abcam);

b. Gas Chromatography-Mass Spectrometry (GC-MS) analysis was performed to determine glycerol concentrations in Gly-CESTs after different duration of dialysis (1 h, 2 h, 4 h, 6 h). * $P < 0.05$, ** $P < 0.01$, *** $P < 0.001$, **** $P < 0.0001$. Data are presented as mean \pm standard deviation (SD) (n = 3).

Description added in the revised manuscript (Results section, Page 13, line 412-422)

To evaluate the glycerol-retaining ability of Gly-CESTs, we applied free glycerol assay kit for glycerol content test. The results showed that the content of glycerol in the dialysate slightly increased over time. At the 6 h of dialysis, only 1.04% (32.3 μmol / 3115 μmol) and 1.33% (41.43 μmol / 3115 μmol) of glycerol from Gly-CESTs free into the external environment (Supplementary Table 1, Supplementary Fig.3a). To further verify the above measured results, the Gas Chromatography-Mass Spectrometry (GC-MS) was used to quantify glycerol concentration in Gly-CESTs dialyzed against ultra-pure water or saline for different lengths of time. Compared with Gly-CESTs without dialysis, the glycerol concentration in Gly-CESTs was only slightly decreased after dialyzed against ultra-pure water or saline (Supplementary Table 2, Supplementary Fig. 3b). These results were consistent with the results of the Free Glycerol Assay Kit.

However, the environment *in vivo* is significantly different from *in vitro*. Therefore, to further explore the glycerol retaining ability of Gly-CESTs, we performed relevant *in vivo* experiments. Glycerol Assay Kit (ab65337) was also applied to determine tissue glycerol concentration. Firstly, considering the presence of endogenous glycerol within living organisms, we tested the endogenous glycerol concentration in both tumor and

liver tissues, which were found to be 677.23 ± 21.94 nmol/g and 2429.15 ± 214.17 nmol/g, respectively. In addition, since both intratumoral injection and intravenous injection were applied in our study for different research aim, we explored the changes of glycerol content under two injection approaches and compared the results with those of exogenous injection of free glycerol. NCI-H460 subcutaneous xenograft tumor tissues were collected at different time points (1 h, 2 h, 4 h, and 6 h) after intratumoral injection of Gly-CESTs (623mM based on glycerol, 50 μ l) or exogenous free glycerol (623mM based on glycerol, 50 μ l), and the glycerol concentration was determined. Balb/c nude mice were sacrificed after intravenously injected with Gly-CESTs (623mM based on glycerol, 200 μ l) or exogenous free glycerol (623mM based on glycerol, 200 μ l) at different time points (1 h, 2 h, 4 h, and 6 h) and the liver tissue were obtained for the determination of glycerol concentration. Corresponding results were added to the Supplementary Information of the revised manuscript (Supplementary Fig. 11). Following the exogenous introduction of Gly-CESTs or free glycerol, the findings indicate that the exogenous free glycerol, being a hydrophilic substance, exhibits a tendency to rapidly clear from tissues irrespective of intratumoral or intravenous administration. Furthermore, it maintains a relatively low concentration comparable to the endogenous glycerol level at all subsequent time intervals (2 h, 4 h, and 6 h, except 1 h). Compared with free glycerol injection group, the clearance of glycerol happens more gradually in Gly-CESTs intratumoral injection group. After 1 h of intratumoral injection of Gly-CESTs, although the glycerol concentration decreased 17.67%, the concentration was still reached 18251 ± 475 nmol/g in NCI-H460 subcutaneous lung tumor tissue. This concentration was found to be 6.22 times higher than that observed in the free glycerol injection group (2932 ± 162 nmol/g). Similar findings were observed in liver tissue after intravenous injection of Gly-CESTs (Supplementary Fig. 11a and b). These experimental results illustrated the glycerol retaining ability of Gly-CESTs.

A new figure added in the revised Supplementary Information (Page 7)

Supplementary Fig. 11 a-b. The determination of *in vivo* glycerol-retaining ability of Gly-CESTs in subcutaneous tumor (a) or liver tissue (b). a. After intratumoral injection of Gly-CESTs or the same amount of free glycerol (623mM based on glycerol, 50µl), the glycerol concentration in tumor tissues at different time points was measured by free glycerol assay kit. Meanwhile, the concentration of endogenous glycerol in tumor was also quantified for accurately reflect the exogenous glycerol concentration. b. The glycerol concentration in liver tissue was determined by Free Glycerol Assay Kit (abcam) at different time points after intravenous injection of Gly-CESTs or free glycerol containing same amount of glycerol (623mM based on glycerol, 200 µl). The concentration of endogenous glycerol in liver was also quantified. ns, no significance, * $P < 0.05$, ** $P < 0.01$, **** $P < 0.0001$. Data are presented as mean \pm standard deviation (SD) (n = 3)

Description added on in the revised manuscript (Results section, Page 17, line 535-554)

To further explore the *in vivo* glycerol retaining ability of Gly-CESTs, NCI-H460 subcutaneous xenograft tumor tissues or BALB/c nude mice liver tissues were collected before and after intratumoral or intravenous injection of Gly-CESTs at different time points (1 h, 2 h, 4 h, and 6 h), respectively. The tissue glycerol concentration was also determined by Free Glycerol Assay Kit. We first determined the endogenous glycerol level existed in tumor and liver tissue, which were found to be 677.23 ± 21.94 nmol/g and 2429.15 ± 214.17 nmol/g, respectively. Following the exogenous introduction of Gly-CESTs or free glycerol, the findings indicate that the exogenous free glycerol, being a hydrophilic substance, exhibits a tendency to rapidly clear from tissues irrespective of intratumoral or intravenous administration. Furthermore, it maintains a relatively low concentration comparable to the endogenous glycerol level at all subsequent time intervals (2 h, 4 h, and 6 h, except 1 h). Compared with free glycerol injection group, the clearance of glycerol happens

more gradually in Gly-CESTs intratumoral injection group. After 1 h of intratumoral injection of Gly-CESTs, although the glycerol concentration decreased 17.67%, the concentration was still reached 18251 ± 475 nmol/g in NCI-H460 subcutaneous lung tumor tissue. This concentration was found to be 6.22 times higher than that observed in the free glycerol injection group (2932 ± 162 nmol/g). Similar findings were observed in liver tissue after intravenous injection of Gly-CESTs (Supplementary Fig. 11a and b). These experimental results illustrated the glycerol retaining ability of Gly-CESTs.

Glycerol can also be existed in the body environment as substantiated by results of our experiments. Therefore, to further confirm our tested results, and to eliminate the influence of endogenous glycerol in the *in vivo* setting, we prepared ^{13}C -Gly-CESTs using ^{13}C isotopically labeled glycerol and tested tissue ^{13}C -glycerol content with carbon-13 nuclear magnetic resonance spectroscopy (^{13}C -NMR) and GC-MS (Supplementary Fig. 11c-f). The injection method and animal model were same as above. The concentration of ^{13}C -glycerol in NCI-H460 subcutaneous tumor tissues was 1360.9 ± 33.8 mg/L after 1 hour intratumoral injection of ^{13}C -Gly-CESTs, which was 7 times higher than that in free ^{13}C -glycerol injection group (194.2 ± 10.94 mg/L). Subsequently, with the extension of time, there was a very slow decline of ^{13}C -glycerol concentration in ^{13}C -Gly-CESTs injection group. Moreover, even after 6 hours of injection, the ^{13}C -glycerol concentration in Gly-CESTs injection group was still higher than that in the free ^{13}C -glycerol group (555.79 ± 14.87 mg/L vs. undetectable, 6 h). Similar observation was found in liver tissue after intravenous injection of ^{13}C -Gly-CESTs or free ^{13}C -glycerol. Even though the hepatic clearance of ^{13}C -glycerol was observed to be more rapid following intravenous administration of ^{13}C -Gly-CESTs compared to subcutaneous tumors, the rate of ^{13}C -glycerol clearance in the ^{13}C -Gly-CESTs injection group remained slower than that in the intravenous free ^{13}C -glycerol injection group (Supplementary Fig. 11c-f). Taken together, above results have confirmed the glycerol-retaining ability of Gly-CESTs and have suggested that glycerol is not that rapidly diffused and cleared from the Gly-CESTs.

A new figure added in the revised Supplementary Information (Page 7)

A new figure added in the revised Supplementary Information (Page 7)

Supplementary Fig. 11c-f. The determination of *in vivo* ^{13}C - glycerol retaining ability of ^{13}C -Gly-CESTs in subcutaneous tumor or liver tissue. c and e. ^{13}C -NMR (c), GC-MS(e) analysis was performed to determine the metabolic tendency or concentration of ^{13}C -glycerol in NCI-H460 tumor tissue at different time points after intratumoral injection of ^{13}C -Gly-CESTs or free ^{13}C -glycerol containing same amount of ^{13}C -glycerol (623mM based on ^{13}C -glycerol, 50 μl); d and f. ^{13}C -NMR (d), GC-MS (f) analysis was used to determine the metabolic tendency or concentration of ^{13}C -glycerol in liver tissue at different time points after intratumoral injection of ^{13}C -Gly-CESTs or free ^{13}C -glycerol containing same amount of ^{13}C -glycerol (623mM based on ^{13}C -glycerol, 200 μl). ns, no significance, * $P < 0.05$, ** $P < 0.01$, *** $P < 0.001$, **** $P < 0.0001$. Data are presented as mean \pm standard deviation (SD) (n = 3)

Description added in the revised manuscript (Results section, Page 17, line 555-576)

Glycerol can be existed in the body environment, involved in fat metabolism. The content of endogenous glycerol was also substantiated by results of our experiments. Therefore, to further confirm our tested results, and to eliminate the influence of endogenous glycerol in the *in vivo* setting, we prepared ^{13}C -Gly-CESTs using ^{13}C isotopically labeled glycerol and tested tissue glycerol content with carbon-13

nuclear magnetic resonance spectroscopy (^{13}C -NMR) and GC-MS. The injection method and animal model were same as above. The concentration of ^{13}C -glycerol in NCI-H460 subcutaneous tumor tissues was 1360.9 ± 33.8 mg/L after 1 hour intratumoral injection of ^{13}C -Gly-CESTs, which was 7 times higher than that in free ^{13}C -glycerol injection group (194.2 ± 10.94 mg/L). Subsequently, with the extension of time, there was a very slow decline of ^{13}C -glycerol concentration in ^{13}C -Gly-CESTs injection group. Moreover, even after 6 hours of injection, the ^{13}C -glycerol concentration in Gly-CESTs injection group was still higher than that in the free ^{13}C -glycerol group (555.79 ± 14.87 mg/L vs. undetectable, 6 h). Similar observation was found in liver tissue after intravenous injection of ^{13}C -Gly-CESTs or free ^{13}C -glycerol. Even though the hepatic clearance of ^{13}C -glycerol was observed to be more rapid following intravenous administration of ^{13}C -Gly-CESTs compared to subcutaneous tumors, the rate of ^{13}C -glycerol clearance in the ^{13}C -Gly-CESTs injection group remained slower than that in the intravenous free ^{13}C -glycerol injection group (Supplementary Fig. 11c-f). Taken together, above results have confirmed the glycerol-retaining ability of Gly-CESTs and have suggested that glycerol is not that rapidly diffused and cleared from the Gly-CESTs.

For the question about how the glycerol is associated with the particles themselves, in our perspective, glycerol should be located at the surface of the nanoparticles in relation to the inquiry regarding the association between glycerol and the particles themselves. Previous studies reported that glycerol and water interact with lipid head groups in a similar fashion, and to partition equally between the bulk and the lipid membrane surface and enhances and strengthens the hydrogen bond network of lipid membrane surface and bulk water similarly. In addition, Glycerol by altering the hydrogen bond structure and intermolecular cohesion of the global solvent, as manifested by increased solvent viscosity [*The Journal of Chemical Physics*, 2016-07-28;145(4):041101]. Moreover, studies showed that the application of hygroscopic glycerol in the solvent of double-network hydrogel could enhance the water retaining ability of hydrogels, generating the water-Gly binary hydrogel with boosted sensitivity to NO_2 and significantly enhanced stability [*ACS Applied Materials and Interfaces*. 2019-01-16;11(2):2364-2373]. Most importantly, in this study we applied

dipalmitoylphosphatidylcholine (DPPC) in the Gly-CESTs preparation, constituting approximately 88.9% of the molar ratio of surfactants (lipids). Harvey et al. examined the effects of bulk glycerol (0-30% w/w) on dipalmitoylphosphatidylcholine (DPPC) monolayers structures and dynamics using complementary biophysical measurements and molecular dynamics (MD) simulations, in which DPPC monolayers and liposomes were used as model pulmonary interfaces. Glycerol was found to preferentially interact with the carbonyl groups in the interfacial region of DPPC and with phosphate and choline in the headgroup, thus causing an increase in the size of the headgroup solvation shell, as evidenced by an expansion of DPPC monolayers (molecular area increased from 52 to 68 Å²) and bilayers seen in both Langmuir isotherms and MD simulations. They illustrated that both small angle neutron scattering, and MD simulations indicated a reduction in gel phase DPPC bilayer thickness by ~3 Å in 30% w/w glycerol, a phenomenon consistent with the observation from FTIR data, that glycerol caused the lipid headgroup to remain oriented parallel to the membrane plane in contrast to its more perpendicular conformation adopted in pure water. Furthermore, in their study, the FTIR measurements suggested that the terminal methyl groups of the DPPC acyl chains were constrained in the presence of glycerol. This observation was supported by MD simulations, which predict bridging between adjacent DPPC headgroups by glycerol as a possible source of its putative membrane stiffening effect. Collectively, these data indicate that glycerol preferentially solvates DPPC headgroups and localizes in specific areas of the interfacial region, resulting in structural changes to DPPC bilayers [*Langmuir*. 2018-06-12;34(23):6941-6954].

In summary, the PFC nanoparticles were composed of a hydrophobic PFC core and a surrounding lipid with a hydrophobic inner layer and hydrophilic outer layers. The preferential interaction of glycerol with the carbonyl groups in the interfacial region of DPPC, as well as with phosphate and choline in the headgroup, along with other established mechanisms, indicates that glycerol locates and retains at the surface of DPPC-lipid nanoprobe. Furthermore, we have confirmed that the Gly-CESTs exhibit an enhanced ability to retain glycerol, thereby impeding its rapid diffusion from the nanoprobe into the surrounding environment, both *in vitro* and *in vivo*. These features finally enable PFC-based Gly-CESTs to realize the high sensitivity, multi-function, multi-modal imaging approaches that successfully employed for sensitive ¹⁹F-CEST

imaging-guided lung cancer precision radiotherapy, illustrating the potential of multi-functional imaging to noninvasively monitor and enhance RT-integrated effectiveness.

For the question about “PG” and biosafety issue:

The components of Gly-PFOBs were 20% (v/v) PFOB, 76% (v/v) ultra-pure water, 2% (v/v) glycerol, and 2% (w/v) of a surfactant. The surfactant (lipids) included 88.9 mol% dipalmitoyl phosphatidylcholine (DPPC), 1 mol% 1,2-dipalmitoyl-sn-glycero-3-phosphoethanolamine (DPPE), 10 mol% cholesterol, and 0.1 mol% lissaminerhodamine B sulfonyl (16:0 LissRhod PE). Therefore, phosphatidyl glyceride (PG) was not considered in our study since it may make the particle Zeta potential more negative as mentioned by reviewer.

In 1996, perfluorooctylbromide(PFOB), also known as perflubron, partial liquid ventilation was shown to alleviate neonatal respiratory distress syndrome and improve pulmonary function, due to its high oxygen dissolving and releasing capability, low surface tension, and excellent physical chemical properties [*New England Journal of Medicine*. 1996-09-12;335(11):761-7]. PFOB used in this study stands out among PFCs for medical use as it, non-toxic, high stability, inertness, possess the unique property of being both hydrophobic and lipophobic, and an acceptable excretion profile [*Pharmaceutics*. 2022-07-19;14(7)] [*Biomaterials*. 2018-05-01;165:1-13]. The Gly-CESTs applied in this research was the normal PFOB based nano-emulsions. It can be well dispersed in water, due to the specific surfactant commixture with high safety profile, such as dipalmitoyl phosphatidylcholine, cholesterol, and less likely to have chemical reactions and intermolecular interactions. Therefore, it has been widely used in the clinic for various purposes, including artificial blood substitution, ultrasound, and ¹⁹F magnetic resonance imaging [*Journal of Experimental & Clinical Cancer Research*, 2021-06-21;40(1):197] [*Journal of Pharmaceutical Investigation*, 2023-01-01;53(1):153-190]. Moreover, there have been PFC nano-emulsions related clinical researches in progress. [*Journal for Immunotherapy of Cancer*. 2023-06-01;11(6)] [*Magnetic Resonance in Medicine*, 2023-07-01;90(1):79-89]. Undoubtedly, glycerol is safe for medical applications. The biosafety and biocompatibility of PFOB based Gly-CESTs also confirmed in this study and results revealed that Gly-CESTs did not induce apparent pathological changes, including cytoplasm loss, cell atrophy, or inflammation, suggesting excellent biocompatibility (Supplementary Fig. 18a-d). Therefore, Gly-CESTs as a normal PFOB based nano-

emulsions, it exhibited a promising clinical translational potential with excellent safety profile. We would like to express our gratitude to the reviewer for their reminder. Additionally, we have incorporated relevant content pertaining to biosafety into the Discussion section (Page 24, line 761-766).

Supplementary figure 18. Gly-CESTs cytotoxicity and biocompatibility assessment. a. Cell viabilities of NCI-H460 cells after co-incubation with Gly-CESTs for 24 h and 48 h. Data are presented as mean \pm standard deviation (SD) (n = 3). b. Body weight changes. Data are presented as mean \pm standard deviation (SD) (n = 3). c. H&E staining of main organ sections from BALB/c

nude mice after Gly-CESTs injection, scale bar: 100 μm . (d) Corresponding hematological analysis of mice treated with saline or Gly-CESTs. Data are presented as mean \pm standard deviation (SD) (n = 3). WBC, white blood cells; RBC, red blood cells; HGB, hemoglobin; HCT, hematocrit; MCV, mean corpuscular volume; MCH, mean corpuscular hemoglobin; MCHC, mean corpuscular hemoglobin concentration; PLT, platelets; MPV, mean platelet volume; PDW, platelet distribution width. Blood samples were collected for hematological analysis on day 3 and day 30 after injection.

Once again, we would like to express our heartfelt thanks to the reviewer#4 for your continued interest in our manuscript and the insightful comments that helped to strengthen and improve our research results. Additionally, we appreciate the editorial recommendations provided for enhancing the abstract and introduction, which have significantly contributed to the enhancement and refinement of our manuscript. Furthermore, apart from the above-mentioned insightful remarks, we have also obtained additional feedback from other reviewers concerning the nomenclature of the dual-mode imaging technique as " ^{19}F -CEST" and the designation of the probes as "Gly-CESTs". Consequently, in the revised manuscript and supplementary information, we have opted to employ the terminology " $^{19}\text{F}/^1\text{H}$ -CEST" and "Gly-PFOBs" instead. We hereby notify Reviewer #4 of this alteration. We eagerly anticipate your response regarding our manuscript and remain at your disposal to address any further inquiries or comments you may have.

REVIEWERS' COMMENTS

Reviewer #1 (Remarks to the Author):

After revision, the manuscript was significantly improved. Additional experiments were performed to demonstrate the retention of glycerol on the surface of the nanoparticles. It is good to see that a quantitative analysis of the SN has been added. In the reference section, the references are still not presented in Nature Communications style. Some of the references include all author names, and some include only one author name. This should be revised. A similar color bar should be used when presenting the same samples. The full raw data should be presented in the Supplementary Information to avoid confusion. Overall, most of my concerns have been addressed. With some minor revisions, the manuscript may be suitable for publication in Nature Communications.

Reviewer #2 (Remarks to the Author):

In this revised manuscript, authors add experiments to evaluate the stability of glycerol on the developed nanoplatform, which is a great improvement. However, the language issues are not addressed well, which will distract readers from the valuable data. For example, but not limited to, weak logic in the abstract:

1. the overall goal of this study is development dual O₂/pH-sensitive Gly-PFOBs for MRI guided ORTW-RT to overcome the radio-resistance of tumor hypoxia. Thus, authors should first explain why ORTW is crucial, what is ORTW, why the hypoxia is dynamically distributed, what the enhancement ratio ORTW-RT will get, etc.
2. Clarify how ¹⁹F/H-CEST quantify O₂.
3. The Gly-PFOB will be a dual O₂/pH-sensitive CEST-MRI imaging probe. But, in the abstract, "oxygenated Gly-PFOB" is mentioned for oxygen-enhanced RT. Does that mean Gly-PFOB will be a theranostic platform?

To sum up, weak logic and confused abbreviation (e.g. PFC, PFOB, oxygenated Gly-PFOB, PFOB-based Gly-PFOBs) in the whole manuscript hinders the readability.

Reviewer #3 (Remarks to the Author):

All comments have been addressed.

Reviewer #4 (Remarks to the Author):

Thank you for the excellent f/u of new data and explanation. I think this proof of concept paper is compelling. Whether it is benefiting substantially from the higher than clinical field strength is not important at this juncture.

REVIEWER COMMENTS

Reviewer #1 (Remarks to the Author):

After revision, the manuscript was significantly improved. Additional experiments were performed to demonstrate the retention of glycerol on the surface of the nanoparticles. It is good to see that a quantitative analysis of the SN has been added. In the reference section, the references are still not presented in Nature Communications style. Some of the references include all author names, and some include only one author name. This should be revised. A similar color bar should be used when presenting the same samples. The full raw data should be presented in the Supplementary Information to avoid confusion. Overall, most of my concerns have been addressed. With some minor revisions, the manuscript may be suitable for publication in Nature Communications.

Thanks for the reviewer's highly positive comments about our work. In addition, we really appreciate the reviewer for carefully reading our manuscript and pointing out the improper reference style. We have modified the reference format to align with the conventions of *Nature Communications*: If 6 or more authors, list the first authors; If less than 6 authors, list all authors. In addition, we ensured that same color bars were used for each experiment. The complete raw data from the bio-distribution experiments depicted in Supplementary Figure 9 and Supplementary Figure 16 were presented in the Source Data file.

Once again, we would like to express our heartfelt thanks to the reviewer#1 for your continued interest in our manuscript and the insightful comments that helped to strengthen and improve our research results. We eagerly anticipate your response regarding our manuscript and remain at your disposal to address any further inquiries or comments you may have.

Reviewer #2 (Remarks to the Author):

In this revised manuscript, authors add experiments to evaluate the stability of glycerol on the developed nanoplatform, which is a great improvement. However, the language issues are not addressed well, which will distract readers from the valuable data. For example, but not limited to, weak logic in the abstract:

1. the overall goal of this study is development dual O₂/pH-sensitive Gly-PFOBs for MRI guided ORTW-RT to overcome the radio-resistance of tumor hypoxia. Thus, authors should first explain why ORTW is crucial, what is ORTW, why the hypoxia is dynamically distributed, what the enhancement ratio ORTW-RT will get, etc.

Thanks for the reviewer's highly positive comments about our work. In addition, we really appreciate the reviewer for carefully reading our manuscript and providing editorial suggestions. In accordance with the reviewer's suggestions, we have intentionally incorporated and emphasized the relevant information in the introduction section, taking into account the prescribed 150-word limit

for abstracts in *Nature Communications*.

About ORTW (Optimized radiotherapy time window) in Introduction section:

Another critical issue that needs to be addressed while administering oxygen-based RT sensitizers is the optimal timing for RT. In other words, to obtain ideal synergistic therapeutic effects, it is essential to figure out whether the oxygen is effectively supplied to tumor hypoxia microenvironment, and what is the real-time status of tumor hypoxia and acidic microenvironment, and which time is optimal for the implementation of radiotherapy.

About acidic and hypoxic tumor microenvironment (TME) in Introduction section:

Further clinical challenges include quantitative determination and visualization of TME dynamic hypoxic and acidic changes. Moreover, tumor hypoxia is often associated with the excessive accumulation of H⁺ ions in the tumor microenvironment (TME), in turn, acidic TME further facilitates the development of hypoxic tumor regions^{9,10}. This vicious cycle eventually leads to the exacerbation of tumor hypoxia, RT resistance, and even worsen therapeutic efficacy.

Enhancement ratio in NSCLC mouse model therapy:

we used five groups of NCI-H460 tumor-bearing BALB/c nude mice at different times after the probe injection: Group I: PBS control; Group II: RT+ Gly-PFOBs (O₂) 0 h; Group III: RT+ Gly-PFOBs (O₂) 1 h~2 h; Group IV: RT+ Gly-PFOBs (O₂) 3 h; Group V: RT alone. As displayed in Figure 5f, RT alone resulted in moderate tumor growth inhibition (44.72%, Group V versus I). Notably, Group II and Group IV effectively inhibited tumor growth by 51.1% (Group II versus Group I, $P<0.01$) and 50.04% (Group IV versus Group I, $P<0.01$). The highest effect of 81.31 % inhibition of tumor growth was found in the RT+ Gly-PFOBs (O₂) 1 h~2 h Group III compared with Group I ($P<0.0001$).

2. Clarify how 19F/H-CEST quantify O₂.

We express our gratitude for the valuable suggestions provided by the reviewer. The Z-spectrum and magnetization transfer ratio asymmetry (MTR_{asym}) curve were used to assess CEST signal contrast ($MTR_{asym} = (S^{\Delta\omega} - S^{-\Delta\omega})/S_0$, where $S^{-\Delta\omega}$ and $S^{\Delta\omega}$ are reference and label signals of RF saturation at $-\Delta\omega$ and $\Delta\omega$, respectively, and $\Delta\omega$ is the labile proton frequency shift from the water resonance. S_0 is the intensity of the bulk water CEST MR signal after irradiation at $-\Delta\omega$). The signal contrast varied as a function of the applied radio frequency (RF) saturation pulse parameters or pulse durations.

Therefore, CEST is not a way to directly provide the precise concentrations or pH values of metabolite indicators in the tumor microenvironment. In this research, we assessed the dynamic trends of acidic and hypoxic microenvironment changes in tumors through the analysis of the dynamic and amplified ¹H-CEST MR water signal trend.

The purpose of this study is to accurately provide ORTW (Optimized radiotherapy time window) through obtained ¹H-CEST MR signal trend. In this process, pH-sensitive chemical optical pH-1 microsensor was applied to verify our obtained ¹H-CEST MR imaging results, especially, the signal trend. The above statements are partially added to the discussion section to provide a comprehensive description of the methodologies employed.

Updated content added to Discussion section (Page 15, line 483-494) in the revised manuscript.

CEST constitutes a powerful sensitivity enhancement mechanism in which the signal of low concentration solutes can be amplified and further visualized through the bulk water signal. In this study, different from the single pH sensitivity agents, we have developed Gly-PFOBs with $^{19}\text{F}/^1\text{H}$ -CEST dual-modality imaging to dynamically provide important metabolic changes of the malignant tumor microenvironment, enabling synchronous pH and oxygen molecular imaging on a single MR machine. This complementary approach was also efficacious in illustrating the complexity and spatiotemporal heterogeneity of the oxygenation status and acidic microenvironment in solid tumors, providing integrated imaging information of precise ORTW. In this process, pH-sensitive chemical optical pH-1 microsensor was applied to verify the accuracy of the ^1H -CEST MR imaging results, especially the dynamic CEST signal trend.

3. The Gly-PFOB will be a dual O₂/pH-sensitive CEST-MRI imaging probe. But, in the abstract, "oxygenated Gly-PFOB" is mentioned for oxygen-enhanced RT. Does that mean Gly-PFOB will be a theranostic platform?

To sum up, weak logic and confused abbreviation (e.g. PFC, PFOB, oxygenated Gly-PFOB, PFOB-based Gly-PFOBs) in the whole manuscript hinders the readability.

We appreciate the helpful comments from the reviewer. In this context, perfluorinated compounds (PFCs) are inert organic compounds with excellent biocompatibility that are previously used in the clinic as "artificial blood" to improve tissue oxygenation due to their high affinity for O₂ via van der Waals interaction, thus Gly-PFOBs, $^{19}\text{F}/^1\text{H}$ -CEST dual-modality MR imaging with pH and O₂ dual-sensitive properties, can also be applied for oxygen enhanced RT. PFOB stands out among PFCs for medical use, thus prompting its utilization in the synthesis of Gly-PFOBs as investigated in this study. In order to mitigate any potential confusion, a thorough reassessment of our description of Gly-PFOBs has been conducted across the entirety of the manuscript.

Once again, we would like to express our heartfelt thanks to the reviewer#2 for your continued interest in our manuscript and the insightful comments that helped to strengthen our research results. We look forward to hearing from you regarding our manuscript. We would be glad to respond to any further questions and comments that you may have.

Reviewer #3 (Remarks to the Author):

All comments have been addressed.

We are very grateful to the reviewers for the positive feedback. Once again, we would like to express our heartfelt thanks to the reviewer#3 for your time and efforts to strengthen our manuscript.

Reviewer #4 (Remarks to the Author):

Thank you for the excellent f/u of new data and explanation. I this this proof of

concept paper is compelling. Whether it is benefiting substantially from the higher than clinical field strength is not important at this juncture.

Thanks for the reviewer's highly positive comments about our work. Once again, we would like to express our heartfelt thanks to the reviewer#4 for your continued interest in our manuscript and the insightful comments that helped to strengthen and improve our research results. Thanks very much!